# Maximum Likelihood Estimation is All You Need for Well-Specified Covariate Shift

**Jiawei Ge**[*†]     **Shange Tang** [*†]     **Jianqing Fan**[†]     **Cong Ma**[‡]     **Chi Jin**[§]

## Abstract

A key challenge of modern machine learning systems is to achieve Out-of-Distribution (OOD) generalization—generalizing to target data whose distribution differs from that of source data. Despite its significant importance, the fundamental question of "what are the most effective algorithms for OOD generalization" remains open even under the standard setting of covariate shift. This paper addresses this fundamental question by proving that, surprisingly, classical Maximum Likelihood Estimation (MLE) purely using source data (without any modification) achieves the *minimax* optimality for covariate shift under the *well-specified* setting. That is, *no* algorithm performs better than MLE in this setting (up to a constant factor), justifying MLE is all you need. Our result holds for a very rich class of parametric models, and does not require any boundedness condition on the density ratio. We illustrate the wide applicability of our framework by instantiating it to three concrete examples—linear regression, logistic regression, and phase retrieval. This paper further complement the study by proving that, under the *misspecified setting*, MLE is no longer the optimal choice, whereas Maximum Weighted Likelihood Estimator (MWLE) emerges as minimax optimal in certain scenarios.

## 1 Introduction

Distribution shift, where the distribution of test data (target data) significantly differs from the distribution of training data (source data), is commonly encountered in practical machine learning scenarios (Zou et al., 2018; Ramponi & Plank, 2020; Guan & Liu, 2021). A central challenge of modern machine learning is to achieve Out-of-Distribution (OOD) generalization, where learned models maintain good performance in the target domain despite the presence of distribution shifts. To address this challenge, a variety of algorithms and techniques have been proposed, including vanilla empirical risk minimization (ERM) (Vapnik, 1999; Gulrajani & Lopez-Paz, 2020), importance weighting (Shimodaira, 2000; Huang et al., 2006; Cortes et al., 2010b; Cortes & Mohri, 2014), learning invariant representations (Ganin et al., 2016; Arjovsky et al., 2019; Wu et al., 2019; Rosenfeld et al., 2020), distributionally robust optimization (DRO) (Sagawa et al., 2019), etc. See the recent survey (Shen et al., 2021) for more details. These results claim the effectiveness of the corresponding proposed algorithms in different regimes. This leads to a natural fundamental question:

*What are the most effective algorithms for OOD generalization?*

This paper consider a widely-studied formulation of OOD-generalization—covariate shift. Under covariate shift, the marginal distributions of the input covariates $X$ vary between the source and target domains, while the conditional distribution of output given covariates $Y \mid X$ remains the same across domains. We consider learning a model from a *known parametric model class* under *well-specified* setting, where well-specification refers to the problems where the true conditional distribution of $Y \mid X$ lies in the given parametric model class. We argue that well-specified setting

---

[*]equal contribution

[†]Department of Operations Research and Financial Engineering, Princeton University; {`jg5300,shangetang,jqfan`}`@princeton.edu`

[‡]Department of Statistics, University of Chicago; `congm@uchicago.edu`

[§]Department of Electrical and Computer Engineering, Princeton University; `chij@princeton.edu`

becomes increasingly more relevant in modern learning applications, because these applications typically use large-scale models with an enormous number of parameters, which are highly expressive and thus make the settings "approximately" well-specified.

Unfortunately, even under the basic setup of well-specified covariate shift, the aforementioned highlighted problem remains elusive — while the seminar work Shimodaira (2000) provides the first asymptotic guarantees for classical Maximum Likelihood Estimation (MLE) algorithm under this setup, and proves its optimality among a specific class of weighted likelihood estimators, his results leave two critical questions open: (1) Does MLE remain effective in the practical non-asymptotic scenario when the number of data is limited? (2) Do there exist smart algorithms beyond the class of weighted likelihood estimators that outperform MLE? This paper precisely addresses both critical questions and thus resolving the highlighted problem under well-specified covariate shift.

**Our contributions.** Concretely, this paper makes following contributions:

1. We prove that, for a large set of well-specified covariate shift problems, the classical Maximum Likelihood Estimation (MLE) — which is computed purely based on source data without using any target data — finds the optimal predictor on the target domain with prediction loss decreases as $\tilde{O}(\mathsf{Tr}(\mathcal{I}_T \mathcal{I}_S^{-1})/n)$. Here $\mathsf{Tr}(\cdot)$ standards for trace, $\mathcal{I}_S, \mathcal{I}_T$ are the fisher information under source and target data distribution respectively, and $n$ is the number of source data. Our result does not require any boundedness condition on the density ratio, and is, to our best knowledge, the *first* general, non-asymptotic, sharp result for MLE on a rich class of covariate shift problems.

2. We provide the *first* minimax lower bound under well-specified covariate shift for *any algorithm*, matching the error rate of MLE. This implies that MLE is minimax optimal, and no algorithm is better than MLE in this setting (up to a constant factor), justifying "MLE is all you need".

3. We instantiate our generic results by considering three representative examples with distinct problem structures: linear regression, logistic regression and phase retrieval. We verify preconditions, compute key quantities, and directly give covariate shift guarantees for these applications.

4. We further complement the study of this paper by considering the *mis-specfied* setting where MLE ceases to work. We establish the *first* general, non-asymptotic upper bound for the Maximum Weighted Likelihood Estimator (MWLE) provided bounded likelihood ratio. We prove that MWLE is minimax optimal under certain worst-case mis-specification.

**MLE versus MWLE.** This paper shows that importance weighting should not always be the go-to algorithm for covariate shift problems. Despite MWLE works under more general mis-specified setting given bounded density ratio, in the well-specified regime, MLE does not require bounded density ratio, and is provably more efficient than MWLE in terms of sample complexity. MLE is all you need for well-specified covariate shift problem.

## 1.1 RELATED WORK

**Parametric covariate shift.** The statistical study of covariate shift under parametric models can be dated back to Shimodaira (2000), which established the asymptotic normality of MWLE and pointed out that vanilla MLE is asymptotically optimal among all the weighted likelihood estimators when the model is well-specified. However, no finite sample guarantees were provided, and the optimality of MLE is only proved within the restricted class of weighted likelihood estimators. In contrast, this paper establishes non-asymptotic results and proves the optimality of MLE among all possible estimators under well-specified models. Cortes et al. (2010a) studied the importance weighting under the statistical learning framework and gave a non-asymptotic upper bound for the generalization error of the weighted estimator. However, their rate scales as $\mathcal{O}(1/\sqrt{n})$ compared to our rate $\mathcal{O}(1/n)$, where $n$ is the sample size. A recent line of work also provide non-asymptotic analyses for covariate shift under well-specified setting, however they focus on linear regression or a few specific models which are more restrictive than our setting: Mousavi Kalan et al. (2020) introduces a statistical minimax framework and provides lower bounds for OOD generalization in the context of linear and one-hidden layer neural network regression models. When applied to covariate shift, their lower bounds are loose and no longer minimax optimal. Lei et al. (2021) considers the minimax optimal estimator for linear regression under fixed design, the estimator they proposed is not MLE and is much more complicated in certain regimes. Finally, Zhang et al. (2022) considers covariate shift in linear regression where the learner can have access to a small number of

target labels, this is beyond the scope of this paper, where we focus on the classical covariate shift setup in which target labels are not known.

**Nonparametric covariate shift.** Another line of work focuses on well-specified nonparametric models under covariate shift. Kpotufe & Martinet (2018) presented minimax results for nonparametric classification problem, which was controlled by a transfer-exponent that measures the discrepancy between source and target. Hanneke & Kpotufe (2019) delves deeper into the advantages of unlabeled target data and demonstrates that typical importance sampling methods do not offer any improvements over the minimax rate already achieved by ERM. Inspired by the aforementioned work, Pathak et al. (2022) studied nonparametric regression problem over the class of Hölder continuous functions with a more fine-grained similarity measure. When considering reproducing kernel Hilbert space (RKHS), Ma et al. (2023) showed kernel ridge regression (KRR) estimator with a properly chosen penalty is minimax optimal for a large family of RKHS when the likelihood ratio is uniformly bounded, and a reweighted KRR using truncated likelihood ratios is minimax optimal when the likelihood ratio has a finite second moment. Later, Wang (2023) proposed a learning strategy based on pseudo-labels. When the likelihood ratio is bounded, their estimator enjoyed the optimality guarantees without prior knowledge about the amount of covariate shift. Although these works focused on covariate shift problems, they considered nonparametric setting, and hence are not directly comparable to our work. As an example, Ma et al. (2023) showed that MLE (empirical risk minimization in their language) is provably suboptimal for addressing covariate shift under nonparametric RKHS assumptions. In contrast, we show that MLE is optimal for covariate shift for a well-specified parametric model. We also highlight that our lower bound is instance dependent in the sense that it depends on the source and target distributions. This is in contrast to prior work (e.g. Ma et al. (2023); Kpotufe & Martinet (2018); Hanneke & Kpotufe (2019)) that consider the worst-case scenario over certain classes of source-target pairs (e.g., bounded density ratios).

**Maximum likelihood estimation.** A crucial part of this work is analyzing MLE, which is a dominant approach in statistical inference. There exists a variety of work studying the behavior of MLE under the standard no-distribution-shift setting. It is well known that MLE is asymptotically normal (Casella & Berger, 2021) with the inverse of Fisher information as the asymptotic variance. Cramér (1946); Rao (1992) established the famous Cramer-Rao bound for unbiased estimators, which also showed that no consistent estimator has lower asymptotic mean squared error than the MLE. White (1982) gave the asymptotic distribution of MLE under the mis-specified setting. More recently, non-asymptotic behaviours of MLE are studied under certain models. Bach (2010); Ostrovskii & Bach (2021) established the non-asymptotic error bound for MLE in logistic regression using self-concordance. This line of work does not consider covariate shift, which is an indispensable part of this paper.

**Importance reweighting algorithms.** Lastly, importance reweighting (or importance sampling) is a classical method to use independent samples from a proposal distribution to approximate expectations w.r.t. a target measure (Agapiou et al., 2017). Chatterjee & Diaconis (2018) studied the sample size (depending on the KL divergence between two distributions) required for importance sampling to approximate a single function. Sanz-Alonso (2018) extended analysis to the case with general $f$-divergences. Zhai et al. (2022) studied overparametrized linear models and showed that Generalized Reweighting (GRW) algorithms, upon achieving zero training error, yield the same results as ERM, thus offering no advantage over it. In addition to correcting covariate shift, importance reweighting has been central in offline reinforcement learning. For instance, Ma et al. (2022) showed a truncated version of importance reweighting is minimax optimal for estimation the value of a target policy using data from a behavior policy. For learning the optimal policy from the behavior data, Swaminathan & Joachims (2015) presented upper bounds of an importance-reweighted estimator. This spurs a long line of work of using importance weighting in offline RL. See the recent work Gabbianelli et al. (2023) and the references therein.

## 2 BACKGROUND AND PROBLEM FORMULATION

In this section, we provide background on the problem of learning under covariate shift. We also review two widely adopted estimators: maximum likelihood estimator and maximum weighted likelihood estimator.

**Notations.** Throughout the paper, we use $c$ to denote universal constants, which may vary from line to line.

## 2.1 COVARIATE SHIFT AND EXCESS RISK

Let $X \in \mathcal{X}$ be the covariates and $Y \in \mathcal{Y}$ be the response variable that we aim to predict. In a general out-of-distribution (OOD) generalization problem, we have two domains of interest, namely a source domain $S$ and a target domain $T$. Each domain is associated with a data generating distribution over $(X, Y)$: $\mathbb{P}_S(X, Y)$ for the source domain and $\mathbb{P}_T(X, Y)$ for the target domain. Given $n$ i.i.d. labeled samples $\{(x_i, y_i)\}_{i=1}^n \sim \mathbb{P}_S(X, Y)$ from the source domain, the goal of OOD generalization is to learn a prediction rule $X \to Y$ that performs well in the target domain. In this paper, we focus on the covariate shift version of the OOD generalization problem, in which the marginal distributions $\mathbb{P}_S(X)$ and $\mathbb{P}_T(X)$ of the covariates could differ between the source and target domains, while the conditional distribution $Y \mid X$ is assumed to be the same on both domains.

More precisely, we adopt the notion of excess risk to measure of the performance of an estimator under covariate shift. Let $\mathcal{F} := \{f(y \mid x; \beta) \mid \beta \in \mathbb{R}^d\}$ be a parameterized function class to model the conditional density function $p(y \mid x)$ of $Y \mid X$. A typical loss function is defined using the negative log-likelihood function $\ell(x, y, \beta) := -\log f(y \mid x; \beta)$. The excess risk at $\beta$ is then defined as

$$R(\beta) := \mathbb{E}_T\left[\ell(x, y, \beta)\right] - \inf_\beta \mathbb{E}_T\left[\ell(x, y, \beta)\right], \tag{1}$$

where the expectation $\mathbb{E}_T$ is taken over $\mathbb{P}_T(X, Y)$. When the model is well-specified, i.e., when the true density $p(y \mid x) = f(y \mid x; \beta^\star)$ for some $\beta^\star$, we have $\inf_\beta \mathbb{E}_T[\ell(x, y, \beta)] = \mathbb{E}_T[\ell(x, y, \beta^\star)]$. As a result, we evaluate the loss at $\beta$ against the loss at the true parameter $\beta^\star$. In contrast, in the case of mis-specification, i.e., when $p(y \mid x) \notin \mathcal{F}$, the loss at $\beta$ is compared against the loss of the best fit in the model class.

## 2.2 MAXIMUM LIKELIHOOD ESTIMATION AND ITS WEIGHTED VERSION

In the no-covariate-shift case, maximum likelihood estimation (MLE) is arguably the most popular approach. Let

$$\ell_n(\beta) := \tfrac{1}{n} \sum_{i=1}^n \ell(x_i, y_i, \beta) \tag{2}$$

be the empirical negative log-likelihood using the samples $\{(x_i, y_i)\}_{i=1}^n$ from the source domain. The vanilla MLE is defined as $\beta_{\mathsf{MLE}} := \arg\min_{\beta \in \mathbb{R}^d} \ell_n(\beta)$.

One potential "criticism" against MLE in the covariate shift setting is that the empirical negative log-likelihood is not a faithful estimate of the out-of-distribution generalization performance, i.e., $\mathbb{E}_T\left[\ell(x, y, \beta)\right]$. In light of this, a weighted version of MLE is proposed. Let $w(x) := d\mathbb{P}_T(x)/d\mathbb{P}_S(x)$ be the density ratio function and

$$\ell_n^w(\beta) := \tfrac{1}{n} \sum_{i=1}^n w(x_i)\ell(x_i, y_i, \beta). \tag{3}$$

be the weighed loss. Then the maximum weighted likelihood estimator is defined as $\beta_{\mathsf{MWLE}} := \arg\min_{\beta \in \mathbb{R}^d} \ell_n^w(\beta)$. It is easy to see that the weighted loss is an unbiased estimate of $\mathbb{E}_T\left[\ell(x, y, \beta)\right]$.

To ease presentations later, we would also recall the classical notion of Fisher information—an important quantity to measure the difficulty of parameter estimation. The Fisher information evaluated at $\beta$ on source and target is defined as

$$\mathcal{I}_S(\beta) := \mathbb{E}_{x \sim \mathbb{P}_S(X), y \sim f(y \mid x; \beta)}\left[\nabla^2 \ell(x, y, \beta)\right], \quad \mathcal{I}_T(\beta) := \mathbb{E}_{x \sim \mathbb{P}_T(X), y \sim f(y \mid x; \beta)}\left[\nabla^2 \ell(x, y, \beta)\right].$$

Here, the gradient and Hessian are taken with respect to the parameter $\beta$.

## 3 WELL-SPECIFIED PARAMETRIC MODEL UNDER COVARIATE SHIFT

In this section, we focus on covariate shift with a well-specified model, that is, the true conditional distribution falls in our parametric function class. This setting aligns with the practice, since in modern machine learning we often deploy large models whose representation ability are so strong that every possible true data distribution almost falls in the function class. We assume there exists

some $\beta^\star$ such that $p(y \mid x) = f(y \mid x; \beta^\star)$, and denote the excess risk evaluated at $\beta$ under true model parameter $\beta^\star$ as $R_{\beta^\star}(\beta)$, i.e.,

$$R_{\beta^\star}(\beta) := \mathbb{E}_{\substack{x \sim \mathbb{P}_T(X) \\ y|x \sim f(y|x;\beta^\star)}} [\ell(x, y, \beta)] - \mathbb{E}_{\substack{x \sim \mathbb{P}_T(X) \\ y|x \sim f(y|x;\beta^\star)}} [\ell(x, y, \beta^\star)]. \tag{4}$$

While the objective of MLE is not an unbiased estimate of the risk under the target domain, we will show in this section that MLE is in fact optimal for addressing covariate shift under well-specified models.

More specifically, in Section 3.1, we provide the performance upper bound for MLE under generic assumptions on the parametric model. Then in Section 3.2, we characterize the performance limit of any estimator in the presence of covariate shift. As we will see, MLE is minimax optimal as it matches the performance limit.

## 3.1 UPPER BOUND FOR MLE

In this subsection, we establish a non-asymptotic upper bound for MLE under generic assumptions on the model class.

**Assumption A.** *We make the following assumptions on the model class $\mathcal{F}$:*

*A.1 There exist $B_1, B_2, N(\delta)$, and absolute constants $c, \gamma$ such that for any fixed matrix $A \in \mathbb{R}^{d \times d}$, any $\delta \in (0, 1)$, and any $n > N(\delta)$, with probability at least $1 - \delta$:*

$$\|A (\nabla \ell_n(\beta^\star) - \mathbb{E}[\nabla \ell_n(\beta^\star)])\|_2 \le c \sqrt{\frac{V \log \frac{d}{\delta}}{n}} + B_1 \|A\|_2 \log^\gamma \left( \frac{B_1 \|A\|_2}{\sqrt{V}} \right) \frac{\log \frac{d}{\delta}}{n}, \tag{5}$$

$$\left\|\nabla^2 \ell_n(\beta^\star) - \mathbb{E}[\nabla^2 \ell_n(\beta^\star)]\right\|_2 \le B_2 \sqrt{\frac{\log \frac{d}{\delta}}{n}}, \tag{6}$$

*where $V = n \cdot \mathbb{E}\|A(\nabla \ell_n(\beta^\star) - \mathbb{E}[\nabla \ell_n(\beta^\star)])\|_2^2$ is the variance.*
*A.2 There exists some constant $B_3 \ge 0$ such that $\|\nabla^3 \ell(x, y, \beta)\|_2 \le B_3$ for all $x \in \mathcal{X}_S \cup \mathcal{X}_T, y \in \mathcal{Y}, \beta \in \mathbb{R}^d$, where $\mathcal{X}_S$ (resp. $\mathcal{X}_T$) is the support of $\mathbb{P}_S(X)$ (resp. $\mathbb{P}_T(X)$).*
*A.3 The empirical loss $\ell_n(\cdot)$ defined in (2) has a unique local minimum in $\mathbb{R}^d$, which is also the global minimum.*

Several remarks on Assumption A are in order. Assumption A.1 is a general version of Bernstein inequality (when $\gamma = 0$ it reduces to classical Bernstein inequality), which gives concentration on gradient and Hessian. This assumption is naturally satisfied when the gradient and Hessian are bounded (see Proposition D.2 for details). Assumption A.2 requires the third order derivative of log-likelihood to be bounded, which is easy to satisfy (e.g., linear regression satisfies this assumption with $B_3 = 0$). Assumption A.3 ensures the MLE is unique, which is standard in the study of the behaviour of MLE. We can see that it naturally applies to traditional convex losses. It is worth noting that our general theorem can also be applied under a relaxed version of Assumption A.3, which will be shown in Theorem 4.5. In Section 4, we will see that Assumption A is mild and easily satisfied for a wide range of models.

Now we are ready to present the performance upper bound for MLE under covariate shift.

**Theorem 3.1.** *Suppose that the model class $\mathcal{F}$ satisfies Assumption A. Let $\mathcal{I}_T := \mathcal{I}_T(\beta^\star)$ and $\mathcal{I}_S := \mathcal{I}_S(\beta^\star)$. For any $\delta \in (0, 1)$, if $n \ge c \max\{N^\star \log(d/\delta), N(\delta)\}$, then with probability at least $1 - 2\delta$, we have $R_{\beta^\star}(\beta_{\mathsf{MLE}}) \le c \frac{\mathsf{Tr}(\mathcal{I}_T \mathcal{I}_S^{-1}) \log \frac{d}{\delta}}{n}$ for an absolute constant $c$. Here $N^\star := Poly(d, B_1, B_2, B_3, \|\mathcal{I}_S^{-1}\|_2, \|\mathcal{I}_T^{\frac{1}{2}} \mathcal{I}_S^{-1} \mathcal{I}_T^{\frac{1}{2}}\|_2^{-1}).$*

For an exact characterization of the threshold $N^\star$, one can refer to Theorem A.1 in the appendix.

Theorem 3.1 gives a non-asymptotic upper bound for the excess risk of MLE: when the sample size exceeds a certain threshold of $\max\{N^\star \log(d/\delta), N(\delta)\}$, MLE achieves an instance dependent risk bound $\mathsf{Tr}(\mathcal{I}_T \mathcal{I}_S^{-1})/n$. It is worth noting that our analysis does not require boundedness on the density ratios between the target and source distributions (as have been assumed in prior art (Ma et al., 2023)), which yields broader applicability. In Section 4, we will instantiate our generic analysis on three different examples: linear regression, logistic regression and phase retrieval.

## 3.2 Minimax lower bound

In the previous section, we have established the upper bound for the vanilla MLE. Now we turn to the complementary question regarding the fundamental limit of covariate shift under well-specified models. To establish the lower bound, we will need the following Assumption B that is a slight variant of Assumption A. Different from the upper bound, the lower bound is algorithm independent and involve a model class rather than a fixed ground truth. Hence, Assumption B focuses on population properties of our model as opposed to Assumption A, which is on the sample level.

**Assumption B.** *Let $\beta_0 \in \mathbb{R}^d$ and $B > 0$. We make the following assumptions on the model class $\mathcal{F}$:*

*B.1 Assumption A.2 holds.*

*B.2 There exist some constants $L_S, L_T \geq 0$ such that for any $\beta_1, \beta_2 \in \mathbb{B}_{\beta_0}(B)$:*

$$\|\mathcal{I}_S(\beta_1) - \mathcal{I}_S(\beta_2)\|_2 \leq L_S \|\beta_1 - \beta_2\|_2, \quad \|\mathcal{I}_T(\beta_1) - \mathcal{I}_T(\beta_2)\|_2 \leq L_T \|\beta_1 - \beta_2\|_2.$$

*B.3 For any $\beta^\star \in \mathbb{B}_{\beta_0}(B)$, the excess risk $R_{\beta^\star}(\beta)$ defined in (4) is convex in $\beta \in \mathbb{R}^d$.*

*B.4 We assume $\mathcal{I}_S(\beta)$ and $\mathcal{I}_T(\beta)$ are positive definite for all $\beta \in \mathbb{B}_{\beta_0}(B)$.*

Assumption B.2 essentially requires the Fisher information will not vary drastically in a small neighbourhood of $\beta_0$. This assumption is easy to hold when the fisher information has certain smoothness (e.g., in linear regression, the fisher information does not change when $\beta$ varies). Since Assumption B is a slight variant of Assumption A, both assumptions are often satisfied simultaneously for a wide range of models, as we will show in Section 4.

**Theorem 3.2.** *Suppose the model class $\mathcal{F}$ satisfies Assumption B. As long as $n \geq N_0$, we have*

$$\inf_{\hat{\beta}} \sup_{\beta^\star \in \mathbb{B}_{\beta_0}(B)} \mathsf{Tr}\left(\mathcal{I}_T(\beta^\star)\mathcal{I}_S^{-1}(\beta^\star)\right)^{-1} \mathbb{E}_{\substack{x_i \sim \mathbb{P}_S(X) \\ y_i|x_i \sim f(y|x;\beta^\star)}}\left[R_{\beta^\star}(\hat{\beta})\right] \geq \frac{1}{50n},$$

*where $N_0 := Poly(d, B^{-1}, B_3, L_S, L_T, \|\mathcal{I}_S(\beta_0)\|_2, \|\mathcal{I}_T(\beta_0)\|_2, \|\mathcal{I}_S(\beta_0)^{-1}\|_2, \|\mathcal{I}_T(\beta_0)^{-1}\|_2)$.*

For an exact characterization of the threshold $N_0$, one can refer to Theorem A.4 in the appendix.

Comparing Theorem 3.1 and 3.2, we can see that, under[1] Assumptions A and B, then for large enough sample size $n$, $\mathsf{Tr}\left(\mathcal{I}_T(\beta^\star)\mathcal{I}_S^{-1}(\beta^\star)\right)/n$ exactly characterizes the fundamental hardness of covariate shift under well-specified parametric models. It also reveals that vanilla MLE is minimax optimal under this scenario. To gain some intuitions, $\mathcal{I}_S^{-1}$ captures the variance of the parameter estimation, and $\mathcal{I}_T$ measures how the excess risk on the target depends on the estimation accuracy of the parameter. Therefore what really affects the excess risk (on target) is the accuracy of estimating the parameter, and vanilla MLE is naturally the most efficient choice.

## 4 Applications

In this section, we illustrate the broad applicability of our framework by delving into three distinct statistical models, namely linear regression, logistic regression and phase retrieval. For each model, we will demonstrate the validity of the assumptions, and give the explicit non-asymptotic upper bound on the vanilla MLE obtained by our framework as well as the threshold of sample size needed to obtain the upper bound.

### 4.1 Linear regression

In linear regression, we have $Y = X^T \beta^\star + \varepsilon$, where $\varepsilon \sim \mathcal{N}(0,1)$ and $\varepsilon \perp\!\!\!\perp X$. The corresponding negative log-likelihood function (i.e. the loss function) is given by $\ell(x, y, \beta) := \frac{1}{2}(y - x^T\beta)^2$. We assume $X \sim \mathcal{N}(0, I_d)$ on the source domain and $X \sim \mathcal{N}(\alpha, \sigma^2 I_d)$ on the target domain.

**Proposition 4.1.** *The aforementioned linear regression model satisfies Assumption A and B with $\gamma = 1$, $N(\delta) = d\log(1/\delta)$, $B_1 = c\sqrt{d}$, $B_2 = c\sqrt{d}$, $B_3 = 0$ and $L_S = L_T = 0$. Moreover, we have $\mathsf{Tr}(\mathcal{I}_T\mathcal{I}_S^{-1}) = \|\alpha\|_2^2 + \sigma^2 d$.*

---

[1]It is worthy to point out that, it is not hard for Assumptions A and B to be satisfied simultaneously. These assumptions will hold naturally when the domain is bounded and the log-likelihood is of certain convexity and smoothness, as we will show in the next section by several concrete examples.

By Theorem 3.1 and Theorem 3.2, since Assumption A and B are satisfied, we immediately demonstrate the optimality of MLE under linear regression. The following theorem gives the explicit form of excess risk bound by applying Theorem 3.1:

**Theorem 4.2.** *For any $\delta \in (0,1)$, if $n \geq \mathcal{O}(N \log \frac{d}{\delta})$, then with probability at least $1 - 2\delta$, we have $R_{\beta^\star}(\beta_{\mathsf{MLE}}) \leq c\frac{(\|\alpha\|_2^2 + \sigma^2 d)\log\frac{d}{\delta}}{n}$, where $N := d\left(1 + \frac{\|\alpha\|_2^2 d + \sigma^2 d}{\|\alpha\|_2^2 + \sigma^2 d}\right)^2$.*

*Remark* (Excess risk). Regarding the upper bound of the excess risk, we categorize it into two scenarios: large shift and small shift. In the small shift scenarios (i.e., $\|\alpha\|_2^2 \leq \sigma^2 d$), the result is the same as that in scenarios without any mean shift, with a rate of $\sigma^2 d/n$. On the other hand, in the large shift scenarios (i.e., $\|\alpha\|_2^2 \geq \sigma^2 d$), the upper bound of the excess risk increases with the mean shift at a rate of $\|\alpha\|_2^2/n$.

*Remark* (Threshold $N$). For a minor mean shift, specifically when $\|\alpha\|_2 = c\sigma$ for a given constant $c$, the threshold is $N = d$. This aligns with the results from linear regression without any covariate shift. On the other hand, as the mean shift increases (i.e., $|\alpha|_2 = \sigma d^k$ for some $0 < k < 1/2$), the threshold becomes $N = d^{4k+1}$, increasing with the growth of $k$. In scenarios where the mean shift significantly surpasses the scaling shift, denoted as $\alpha \geq \sigma\sqrt{d}$, the threshold reaches $N = d^3$.

## 4.2 LOGISTIC REGRESSION

In the logistic regression, the response variable $Y \in \{0,1\}$ obeys $\mathbb{P}(Y = 1 \mid X = x) = 1/(1 + e^{x^T\beta^\star})$, $\mathbb{P}(Y = 0 \mid X = x) = 1/(1 + e^{-x^T\beta^\star})$. The corresponding negative log-likelihood function (i.e. the loss function) is given by $\ell(x, y, \beta) := \log(1 + e^{x^T\beta}) - y(x^T\beta)$. We assume $X \sim \mathsf{Uniform}(\mathcal{S}^{d-1}(\sqrt{d}))$ on the source domain and $X \sim \mathsf{Uniform}(\mathcal{S}^{d-1}(\sqrt{d})) + v$ on the target domain, where $\mathcal{S}^{d-1}(\sqrt{d}) := \{x \in \mathbb{R}^d \mid \|x\|_2 = \sqrt{d}\}$. In the following, we will give the upper bound of the excess risk for MLE when $v = r\beta_\perp^\star$, where $\beta_\perp^\star$ represents a vector perpendicular to $\beta^\star$ (i.e., $\beta_\perp^{\star T}\beta^\star$=0). Without loss of generality, we assume $\|\beta^\star\|_2 = \|\beta_\perp^\star\|_2 = 1$.

**Proposition 4.3.** *The aforementioned logistic regression model satisfies Assumption A and B with $\gamma = 0$, $N(\delta) = 0$, $B_1 = c\sqrt{d}$, $B_2 = cd$, $B_3 = (\sqrt{d} + r)^3$, $L_S = d^{1.5}$ and $L_T = (\sqrt{d} + r)^3$. Moreover, we have $\mathsf{Tr}(\mathcal{I}_T\mathcal{I}_S^{-1}) \asymp d + r^2$.*

By Theorem 3.1 and Theorem 3.2, since Assumption A and B are satisfied, we immediately demonstrate the optimality of MLE under logistic regression. The following theorem gives the explicit form of excess risk bound by applying Theorem 3.1:

**Theorem 4.4.** *For any $\delta \in (0,1)$, if $n \geq \mathcal{O}(N \log \frac{d}{\delta})$, then with probability at least $1 - 2\delta$, we have $R_{\beta^\star}(\beta_{\mathsf{MLE}}) \leq c\frac{(d+r^2)\log\frac{d}{\delta}}{n}$, where $N := d^4(1 + r^6)$.*

*Remark* (Excess risk). The bound on the excess risk incorporates a $r^2$ term, which is a measurement of the mean shift. This is due to the fact that the MLE does not utilize the information that $v^T\beta^\star = 0$. Therefore, $v^T\beta_{\mathsf{MLE}}$ is not necessarily zero, which will lead to an additional bias. Similar to linear regression, we can categorize the upper bound of the excess risk into two scenarios: large shift ($r \geq \sqrt{d}$) and small shift ($r \leq \sqrt{d}$).

*Remark* (Threshold $N$). We admit that the $N$ here may not be tight, as we lean on a general framework designed for a variety of models rather than a specific one.

## 4.3 PHASE RETRIEVAL

As we have mentioned, our generic framework can also be applied to the scenarios where some of the assumptions are relaxed. In this subsection, we will further illustrate this point by delving into the phase retrieval model. In the phase retrieval, the response variable $Y = (X^T\beta^\star)^2 + \varepsilon$, where $\varepsilon \sim \mathcal{N}(0,1)$ and $\varepsilon \perp\!\!\!\perp X$. We assume $\mathbb{P}_S(X)$ and $\mathbb{P}_T(X)$ follow the same distribution as that in the logistic regression model (i.e., Section 4.2). Note that both the phase retrieval model and the logistic regression model belong to generalized linear model (GLM), thus they are expected to have similar properties. However, given the loss function $\ell(x, y, \beta) := \frac{1}{2}\left(y - (x^T\beta)^2\right)^2$, it is obvious that Assumption A.3 is not satisfied, since if $\beta$ is a global minimum of $\ell_n$, $-\beta$ is also a global minimum. The following theorem shows that we can still obtain results similar to logistic regression though Assumption A.3 fails to hold.

**Theorem 4.5.** *For any $\delta \in (0,1)$, if $n \geq \mathcal{O}(N \log \frac{d}{\delta})$, then with probability at least $1 - 2\delta$, we have $R_{\beta^\star}(\beta_{\mathsf{MLE}}) \leq c \frac{(d+r^2) \log \frac{d}{\delta}}{n}$, where $N := d^8(1 + r^8)$.*

# 5 MIS-SPECIFIED PARAMETRIC MODEL UNDER COVARIATE SHIFT

In the case of model mis-specification, we still employ a parameterized function class $\mathcal{F} := \{f(y \,|\, x; \beta) \,|\, \beta \in \mathbb{R}^d\}$ to model the conditional density function of $Y \,|\, X$. However, the true density $p(y \,|\, x)$ might not be in $\mathcal{F}$. As we previously showed, under a well-specified parametric model, the vanilla MLE is minimax optimal up to constants. However, when the model is mis-specified, the classical MLE may not necessarily provide a good estimator.

**Proposition 5.1.** *There exist certain mis-specified scenarios such that classical MLE is not consistent, whereas MWLE is.*

Proposition 5.1 illustrates the necessity of adaptation under model mis-specification since the classical MLE asymptotically gives the wrong estimator. In this section, we study the non-asymptotic property of MWLE. Let $\mathcal{M}$ be the model class of the ground truth $Y \,|\, X$, and $M \in \mathcal{M}$ be the ground truth model for $Y \,|\, X$.

We denote the optimal fit on target as $\beta^\star(M) := \arg\min_\beta \mathbb{E}_{\substack{x \sim \mathbb{P}_T(X) \\ y|x \sim M}}[\ell(x, y, \beta)]$. The excess risk evaluated at $\beta$ is then given by $R_M(\beta) = \mathbb{E}_{\substack{x \sim \mathbb{P}_T(X) \\ y|x \sim M}} [\ell(x, y, \beta)] - \mathbb{E}_{\substack{x \sim \mathbb{P}_T(X) \\ y|x \sim M}} [\ell(x, y, \beta^\star(M))]$.

## 5.1 UPPER BOUND FOR MWLE

In this subsection, we establish the non-asymptotic upper bound for MWLE, as an analog to Theorem 3.1. We make the following assumption which is a modification of Assumption A.

**Assumption C.** *We assume the function class $\mathcal{F}$ satisfies the follows:*

*C.1 There exists some constant $W > 1$ such that the density ratio $w(x) \leq W$ for all $x \in \mathcal{X}_S \cup \mathcal{X}_T$.*

*C.2 There exist $B_1, B_2$ and $N(\delta)$, and absolute constants $c, \gamma$ such that for any fixed matrix $A \in \mathbb{R}^{d \times d}$, any $\delta \in (0,1)$, and any $n > N(\delta)$, with probability at least $1 - \delta$:*

$$\left\| A \left( \nabla \ell_n^w(\beta^\star(M)) - \mathbb{E}[\nabla \ell_n^w(\beta^\star(M))] \right) \right\|_2 \leq c \sqrt{\frac{V \log \frac{d}{\delta}}{n}} + W B_1 \|A\|_2 \log^\gamma \left( \frac{W B_1 \|A\|_2}{\sqrt{V}} \right) \frac{\log \frac{d}{\delta}}{n},$$

$$\left\| \nabla^2 \ell_n^w(\beta^\star(M)) - \mathbb{E}[\nabla^2 \ell_n^w(\beta^\star(M))] \right\|_2 \leq W B_2 \sqrt{\frac{\log \frac{d}{\delta}}{n}},$$

*where $V = n \cdot \mathbb{E}\|A(\nabla \ell_n^w(\beta^\star(M)) - \mathbb{E}[\nabla \ell_n^w(\beta^\star(M))])\|_2^2$ is the variance.*

*C.3 Assumption A.2 holds.*

*C.4 There exists $N'(\delta)$ such that for any $\delta \in (0,1)$ and any $n \geq N'(\delta)$, with probability at least $1 - \delta$, the empirical loss $\ell_n^w(\cdot)$ defined in (3) has a unique local minimum in $\mathbb{R}^d$, which is also the global minimum.*

Assumption C.1 is a density ratio upper bound (not required for analyzing MLE), which is essential for the analysis of MWLE. Assumption C.2 is an analog of Assumption A.1, in the sense that the empirical loss $\ell_n$ is replaced by its weighted version $\ell_n^w$. Assumption C.4 is a weaker version of Assumption A.3 in the sense that it only requires $\ell_n^w$ has a unique local minimum with high probability. This is due to the nature of reweighting: when applying MWLE, $w(x_i)$ can sometimes be zero, which lead to the degeneration of $\ell_n^w$ (with a small probability). Therefore we only require the uniqueness of local minimum holds with high probability.

To state our non-asymptotic upper bound for MWLE, we define the following "weighted version" of Fisher information:

$$G_w(M) := \mathbb{E}_{\substack{x \sim \mathbb{P}_S(X) \\ y|x \sim M}} \left[ w(x)^2 \nabla \ell(x, y, \beta^\star(M)) \nabla \ell(x, y, \beta^\star(M))^T \right],$$

$$H_w(M) := \mathbb{E}_{\substack{x \sim \mathbb{P}_S(X) \\ y|x \sim M}} \left[ w(x) \nabla^2 \ell(x, y, \beta^\star(M)) \right] = \mathbb{E}_{\substack{x \sim \mathbb{P}_T(X) \\ y|x \sim M}} \left[ \nabla^2 \ell(x, y, \beta^\star(M)) \right].$$

**Theorem 5.2.** *Suppose the function class $\mathcal{F}$ satisfies Assumption C. Let $G_w := G_w(M)$ and $H_w := H_w(M)$. For any $\delta \in (0,1)$, if $n \geq c \max\{N^\star \log(d/\delta), N(\delta), N'(\delta)\}$, then with probability at least $1 - 3\delta$, we have $R_M(\beta_{\mathsf{MWLE}}) \leq c \frac{\mathsf{Tr}(G_w H_w^{-1}) \log \frac{d}{\delta}}{n}$ for an absolute constant c. Here $N^\star := Poly(W, B_1, B_2, B_3, \|H_w^{-1}\|_2, \mathsf{Tr}(G_w H_w^{-2}), \mathsf{Tr}(G_w H_w^{-2})^{-1})$.*

For an exact characterization of the threshold $N^\star$, one can refer to Theorem C.1 in the appendix.

Compared with Theorem 3.1, Theorem 5.2 does not require well-specification of the model, demonstrating the wide applicability of MWLE. The excess risk upper bound can be explained as follows: note that $\mathsf{Tr}(G_w H_w^{-1})$ can be expanded as $\mathsf{Tr}(H_w H_w^{-1} G_w H_w^{-1})$. As shown by Shimodaira (2000), the term $\sqrt{n}(\beta_{\mathsf{MWLE}} - \beta^\star)$ converges asymptotically to a normal distribution, denoted as $\mathcal{N}(0, H_w^{-1} G_w H_w^{-1})$. Thus, the component $H_w^{-1} G_w H_w^{-1}$ characterizes the variance of the estimator, corresponding to the $\mathcal{I}_S^{-1}$ term in Theorem 3.1. Additionally, the excess risk's dependence on the parameter estimation is captured by $H_w$ as a counterpart of $\mathcal{I}_T$ in Theorem 3.1.

However, to establish Theorem 5.2, it is necessary to assume the bounded density ratio, which does not appear in Theorem 3.1. Moreover, when the model is well-specified, by Cauchy-Schwarz inequality, we have $\mathsf{Tr}(G_w H_w^{-1}) \geq \mathsf{Tr}(\mathcal{I}_T \mathcal{I}_S^{-1})$, which implies the upper bound for MWLE is larger than the vanilla MLE. This observation aligns with the results presented in Shimodaira (2000), which point out that when the model is well specified, MLE is more efficient than MWLE in terms of the asymptotic variance.

## 5.2 Optimality of MWLE

To understand the optimality of MWLE, it is necessary to establish a matching lower bound. However, deriving a lower bound similar to Theorem 3.2, which holds for any model classes that satisfies certain mild conditions, is challenging due to hardness of capturing the difference between $\mathcal{M}$ and $\mathcal{F}$. As a solution, we present a lower bound tailored for certain model classes and data distributions in the following.

**Theorem 5.3.** *There exist $\mathbb{P}_S(X) \neq \mathbb{P}_T(X)$, a model class $\mathcal{M}$ and a prediction class $\mathcal{F}$ satisfying Assumption C such that when $n$ is sufficiently large, we have*

$$\inf_{\hat{\beta}} \sup_{M \in \mathcal{M}} \mathsf{Tr}\left(G_w(M) H_w^{-1}(M)\right)^{-1} \mathbb{E}_{\substack{x_i \sim \mathbb{P}_S(X) \\ y_i | x_i \sim M}} \left[R_M(\hat{\beta})\right] \gtrsim \tfrac{1}{n}. \tag{7}$$

By Theorem 5.2, the excess risk of MWLE is upper bounded by $\mathsf{Tr}(G_w H_w^{-1})/n$. Therefore, Theorem 5.3 shows that there exists a non-trivial scenario where MWLE is minimax optimal.

Notice that Theorem 5.3 presents a weaker lower bound compared to Theorem 3.2. The lower bound presented in Theorem 5.3 holds only for certain meticulously chosen $\mathbb{P}_S(X), \mathbb{P}_T(X)$, model class $\mathcal{M}$ and prediction class $\mathcal{F}$. In contrast, the lower bound in Theorem 3.2 applies to any $\mathbb{P}_S(X), \mathbb{P}_T(X)$, and class $\mathcal{F}$ that meet the required assumptions.

## 6 Conclusion and Discussion

To conclude, we prove that MLE achieves the minimax optimality for covariate shift under a well-specified parametric model. Along the way, we demonstrate that the term $\mathsf{Tr}(\mathcal{I}_T \mathcal{I}_S^{-1})$ characterizes the foundamental hardness of covariate shift, where $\mathcal{I}_S$ and $\mathcal{I}_T$ are the Fisher information on the source domain and the target domain, respectively. To complement the study, we also consider the misspecified setting and show that Maximum Weighted Likelihood Estimator (MWLE) emerges as minimax optimal in specific scenarios, outperforming MLE.

Our work opens up several interesting avenues for future study. First, it is of great interest to extend our analysis to other types of OOD generalization problems, e.g., imbalanced data, posterior shift, etc. Second, our analyses relies on standard regularity assumptions, such as the positive definiteness of the Fisher information (which implies certain identifiability of the parameter) and the uniqueness of the minimum of the loss function. Addressing covariate shift without these assumptions is also important future directions.

ACKNOWLEDGEMENT

Cong Ma is partially supported by the National Science Foundation via grant DMS-2311127.

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
