# A    PROOFS FOR SECTION 3

## A.1    PROOFS FOR THEOREM 3.1

The detailed version of Theorem 3.1 is stated as the following.

**Theorem A.1.** *Suppose that the model class $\mathcal{F}$ satisfies Assumption A. Let $\mathcal{I}_T := \mathcal{I}_T(\beta^\star)$ and $\mathcal{I}_S := \mathcal{I}_S(\beta^\star)$. For any $\delta \in (0,1)$, if $n \geq c \max\{N^\star \log(d/\delta), N(\delta)\}$, then with probability at least $1 - 2\delta$, we have*

$$R_{\beta^\star}(\beta_{\mathsf{MLE}}) \leq c \frac{\mathsf{Tr}\left(\mathcal{I}_T \mathcal{I}_S^{-1}\right) \log \frac{d}{\delta}}{n}$$

*for an absolute constant $c$. Here*

$$N^\star := (1 + \tilde{\kappa}/\kappa)^2 \cdot \max\left\{\tilde{\kappa}^{-1}\alpha_1^2 \log^{2\gamma}\left((1 + \tilde{\kappa}/\kappa)\tilde{\kappa}^{-1}\alpha_1^2\right),\ \alpha_2^2,\ \tilde{\kappa}(1 + \|\mathcal{I}_T^{\frac{1}{2}}\mathcal{I}_S^{-1}\mathcal{I}_T^{\frac{1}{2}}\|_2^{-2})\alpha_3^2\right\},$$

*where $\alpha_1 := B_1 \|\mathcal{I}_S^{-1}\|_2^{1/2}$, $\alpha_2 := B_2 \|\mathcal{I}_S^{-1}\|_2$, $\alpha_3 := B_3 \|\mathcal{I}_S^{-1}\|_2^{3/2}$,*

$$\kappa := \frac{\mathsf{Tr}(\mathcal{I}_T \mathcal{I}_S^{-1})}{\|\mathcal{I}_T^{\frac{1}{2}}\mathcal{I}_S^{-1}\mathcal{I}_T^{\frac{1}{2}}\|_2}, \quad \tilde{\kappa} := \frac{\mathsf{Tr}(\mathcal{I}_S^{-1})}{\|\mathcal{I}_S^{-1}\|_2}.$$

For proving Theorem A.1, we first state two main lemmas. Informally speaking, Lemma A.2 and Lemma A.3 capture the distance between $\beta_{\mathsf{MLE}}$ and $\beta^\star$ under different measurements.

**Lemma A.2.** *Suppose Assumption A holds. For any $\delta \in (0,1)$ and any $n \geq c \max\{N_1 \log(d/\delta), N(\delta)\}$, with probability at least $1 - \delta$, we have $\beta_{\mathsf{MLE}} \in \mathbb{B}_{\beta^\star}(c\sqrt{\frac{\mathsf{Tr}(\mathcal{I}_S^{-1}) \log \frac{d}{\delta}}{n}})$ for some absolute constant $c$. Here*

$$N_1 := \max\left\{B_2^2 \|\mathcal{I}_S^{-1}\|_2^2, B_3^2 \|\mathcal{I}_S^{-1}\|_2^2 \mathsf{Tr}(\mathcal{I}_S^{-1}), \left(\frac{B_1^2 B_2 \|\mathcal{I}_S^{-1}\|_2^3 \log^{2\gamma}(\tilde{\kappa}^{-1/2}\alpha_1)}{\mathsf{Tr}(\mathcal{I}_S^{-1})}\right)^{\frac{2}{3}},\right.$$

$$\left.\left(\frac{B_1^3 B_3 \|\mathcal{I}_S^{-1}\|_2^4 \log^{3\gamma}(\tilde{\kappa}^{-1/2}\alpha_1)}{\mathsf{Tr}(\mathcal{I}_S^{-1})}\right)^{\frac{1}{2}}, \frac{B_1^2 \|\mathcal{I}_S^{-1}\|_2^2 \log^{2\gamma}(\tilde{\kappa}^{-1/2}\alpha_1)}{\mathsf{Tr}(\mathcal{I}_S^{-1})}\right\}.$$

**Lemma A.3.** *Suppose Assumption A holds. For any $\delta \in (0,1)$ and any $n \geq c \max\{N_1 \log(d/\delta), N_2 \log(d/\delta), N(\delta)\}$, with probability at least $1 - 2\delta$, we have*

$$\|\mathcal{I}_T^{\frac{1}{2}}(\beta_{\mathsf{MLE}} - \beta^\star)\|_2^2 \leq c \frac{\mathsf{Tr}(\mathcal{I}_T \mathcal{I}_S^{-1}) \log \frac{d}{\delta}}{n}$$

*for some absolute constant $c$. Here $N_1$ is defined in Lemma A.2 and*

$$N_2 := \max\left\{\left(\frac{B_2 \|\mathcal{I}_T^{\frac{1}{2}}\mathcal{I}_S^{-\frac{1}{2}}\|_2^2 \mathsf{Tr}(\mathcal{I}_S^{-1})}{\mathsf{Tr}(\mathcal{I}_T \mathcal{I}_S^{-1})}\right)^2, \left(\frac{B_3 \|\mathcal{I}_T^{\frac{1}{2}}\mathcal{I}_S^{-\frac{1}{2}}\|_2^2 \mathsf{Tr}(\mathcal{I}_S^{-1})^{1.5}}{\mathsf{Tr}(\mathcal{I}_T \mathcal{I}_S^{-1})}\right)^2,\right.$$

$$\left(\frac{B_1^2 B_2 \|\mathcal{I}_T^{\frac{1}{2}}\mathcal{I}_S^{-\frac{1}{2}}\|_2^2 \|\mathcal{I}_S^{-1}\|_2^2 \log^{2\gamma}(\tilde{\kappa}^{-1/2}\alpha_1)}{\mathsf{Tr}(\mathcal{I}_T \mathcal{I}_S^{-1})}\right)^{\frac{2}{3}}, \left(\frac{B_1^3 B_3 \|\mathcal{I}_T^{\frac{1}{2}}\mathcal{I}_S^{-\frac{1}{2}}\|_2^2 \|\mathcal{I}_S^{-1}\|_2^3 \log^{3\gamma}(\tilde{\kappa}^{-1/2}\alpha_1)}{\mathsf{Tr}(\mathcal{I}_T \mathcal{I}_S^{-1})}\right)^{\frac{1}{2}},$$

$$\frac{B_1^2 \|\mathcal{I}_T^{\frac{1}{2}}\mathcal{I}_S^{-\frac{1}{2}}\|_2^2 \|\mathcal{I}_S^{-1}\|_2 \log^{2\gamma}(\kappa^{-1/2}\alpha_1)}{\mathsf{Tr}(\mathcal{I}_T \mathcal{I}_S^{-1})}\left.\right\}.$$

The proofs for Lemma A.2 and A.3 are delayed to the end of this subsection. With these two lemmas, we can now state the proof for Theorem A.1.

*Proof of Theorem A.1.* By Assumption A.2, we can do Taylor expansion w.r.t. $\beta$ as the following:

$$R_{\beta^\star}(\beta_{\mathsf{MLE}}) = \mathop{\mathbb{E}}_{\substack{x \sim \mathbb{P}_T(X) \\ y|x \sim f(y|x;\beta^\star)}} [\ell(x,y,\beta_{\mathsf{MLE}}) - \ell(x,y,\beta^\star)]$$

$$\leq \mathop{\mathbb{E}}_{\substack{x \sim \mathbb{P}_T(X) \\ y|x \sim f(y|x;\beta^\star)}} [\nabla \ell(x,y,\beta^\star)]^T (\beta_{\mathsf{MLE}} - \beta^\star)$$

$$+ \frac{1}{2}(\beta_{\mathsf{MLE}} - \beta^\star)^T \mathcal{I}_T (\beta_{\mathsf{MLE}} - \beta^\star) + \frac{B_3}{6}\|\beta_{\mathsf{MLE}} - \beta^\star\|_2^3.$$

Applying Lemma A.2 and A.3, we know for any $\delta$ and any $n \geq c \max\{N_1 \log(d/\delta), N_2 \log(d/\delta), N(\delta)\}$, with probability at least $1 - 2\delta$, we have

$$(\beta_{\mathsf{MLE}} - \beta^\star)^T \mathcal{I}_T (\beta_{\mathsf{MLE}} - \beta^\star) \leq c \frac{\mathsf{Tr}(\mathcal{I}_T \mathcal{I}_S^{-1}) \log \frac{d}{\delta}}{n}$$

and

$$\|\beta_{\mathsf{MLE}} - \beta^\star\|_2 \leq c \sqrt{\frac{\mathsf{Tr}(\mathcal{I}_S^{-1}) \log \frac{d}{\delta}}{n}}.$$

Also notice that, $\mathbb{E}_{\substack{x \sim \mathbb{P}_T(X) \\ y|x \sim f(y|x;\beta^\star)}} [\nabla \ell(x, y, \beta^\star)] = 0$. Therefore, with probability at least $1 - 2\delta$, we have

$$R_{\beta^\star}(\beta_{\mathsf{MLE}}) \leq \frac{c}{2} \frac{\mathsf{Tr}(\mathcal{I}_T \mathcal{I}_S^{-1}) \log \frac{d}{\delta}}{n} + \frac{c^3}{6} B_3 \mathsf{Tr}(\mathcal{I}_S^{-1})^{1.5} (\frac{\log \frac{d}{\delta}}{n})^{1.5}$$

for any $\delta$ and any $n \geq c \max\{N_1 \log(d/\delta), N_2 \log(d/\delta), N(\delta)\}$. If we further assume $n \geq c (\frac{B_3 \mathsf{Tr}(\mathcal{I}_S^{-1})^{1.5}}{\mathsf{Tr}(\mathcal{I}_T \mathcal{I}_S^{-1})})^2 \log(d/\delta)$, it then holds that

$$R_{\beta^\star}(\beta_{\mathsf{MLE}}) \leq c \frac{\mathsf{Tr}(\mathcal{I}_T \mathcal{I}_S^{-1}) \log \frac{d}{\delta}}{n}.$$

Note that

$$\max \left\{ N_1, N_2, \left( \frac{B_3 \mathsf{Tr}(\mathcal{I}_S^{-1})^{1.5}}{\mathsf{Tr}(\mathcal{I}_T \mathcal{I}_S^{-1})} \right)^2 \right\}$$

$$= \max \left\{ B_2^2 \|\mathcal{I}_S^{-1}\|_2^2, B_3^2 \|\mathcal{I}_S^{-1}\|_2^2 \mathsf{Tr}(\mathcal{I}_S^{-1}), \left( \frac{B_1^2 B_2 \|\mathcal{I}_S^{-1}\|_2^3 \log^{2\gamma}(\tilde{\kappa}^{-1/2}\alpha_1)}{\mathsf{Tr}(\mathcal{I}_S^{-1})} \right)^{\frac{2}{3}}, \left( \frac{B_1^3 B_3 \|\mathcal{I}_S^{-1}\|_2^4 \log^{3\gamma}(\tilde{\kappa}^{-1/2}\alpha_1)}{\mathsf{Tr}(\mathcal{I}_S^{-1})} \right)^{\frac{1}{2}}, \right.$$

$$\frac{B_1^2 \|\mathcal{I}_S^{-1}\|_2^2 \log^{2\gamma}(\tilde{\kappa}^{-1/2}\alpha_1)}{\mathsf{Tr}(\mathcal{I}_S^{-1})}, \left( \frac{B_2 \|\mathcal{I}_T^{\frac{1}{2}} \mathcal{I}_S^{-\frac{1}{2}}\|_2^2 \mathsf{Tr}(\mathcal{I}_S^{-1})}{\mathsf{Tr}(\mathcal{I}_T \mathcal{I}_S^{-1})} \right)^2, \left( \frac{B_3 \|\mathcal{I}_T^{\frac{1}{2}} \mathcal{I}_S^{-\frac{1}{2}}\|_2^2 \mathsf{Tr}(\mathcal{I}_S^{-1})^{1.5}}{\mathsf{Tr}(\mathcal{I}_T \mathcal{I}_S^{-1})} \right)^2,$$

$$\left( \frac{B_1^2 B_2 \|\mathcal{I}_T^{\frac{1}{2}} \mathcal{I}_S^{-\frac{1}{2}}\|_2^2 \|\mathcal{I}_S^{-1}\|_2^2 \log^{2\gamma}(\tilde{\kappa}^{-1/2}\alpha_1)}{\mathsf{Tr}(\mathcal{I}_T \mathcal{I}_S^{-1})} \right)^{\frac{2}{3}}, \left( \frac{B_1^3 B_3 \|\mathcal{I}_T^{\frac{1}{2}} \mathcal{I}_S^{-\frac{1}{2}}\|_2^2 \|\mathcal{I}_S^{-1}\|_2^3 \log^{3\gamma}(\tilde{\kappa}^{-1/2}\alpha_1)}{\mathsf{Tr}(\mathcal{I}_T \mathcal{I}_S^{-1})} \right)^{\frac{1}{2}},$$

$$\left. \frac{B_1^2 \|\mathcal{I}_T^{\frac{1}{2}} \mathcal{I}_S^{-\frac{1}{2}}\|_2^2 \|\mathcal{I}_S^{-1}\|_2 \log^{2\gamma}(\kappa^{-1/2}\alpha_1)}{\mathsf{Tr}(\mathcal{I}_T \mathcal{I}_S^{-1})}, \left( \frac{B_3 \mathsf{Tr}(\mathcal{I}_S^{-1})^{1.5}}{\mathsf{Tr}(\mathcal{I}_T \mathcal{I}_S^{-1})} \right)^2 \right\}$$

$$= \max \left\{ \alpha_2^2, \tilde{\kappa}\alpha_3^2, \alpha_1^{4/3}\alpha_2^{2/3}\tilde{\kappa}^{-2/3} \log^{4\gamma/3}(\tilde{\kappa}^{-1/2}\alpha_1), \alpha_1^{3/2}\alpha_3^{1/2}\tilde{\kappa}^{-1/2} \log^{3\gamma/2}(\tilde{\kappa}^{-1/2}\alpha_1), \alpha_1^2\tilde{\kappa}^{-1} \log^{2\gamma}(\tilde{\kappa}^{-1/2}\alpha_1), \right.$$

$$\alpha_2^2(\tilde{\kappa}/\kappa)^2, \alpha_3^2\tilde{\kappa}^3/\kappa^2, \alpha_1^{4/3}\alpha_2^{2/3}\kappa^{-2/3} \log^{4\gamma/3}(\tilde{\kappa}^{-1/2}\alpha_1), \alpha_1^{3/2}\alpha_3^{1/2}\kappa^{-1/2} \log^{3\gamma/2}(\tilde{\kappa}^{-1/2}\alpha_1),$$

$$\left. \alpha_1^2\kappa^{-1} \log^{2\gamma}(\kappa^{-1/2}\alpha_1), \alpha_3^2\tilde{\kappa}^3\kappa^{-2} \|\mathcal{I}_T^{\frac{1}{2}} \mathcal{I}_S^{-1} \mathcal{I}_T^{\frac{1}{2}}\|_2^{-2} \right\}$$

$$\leq \max \left\{ \tilde{\kappa}^{-1}\alpha_1^2 \log^{2\gamma} \left( (1 + \tilde{\kappa}/\kappa)\tilde{\kappa}^{-1}\alpha_1^2 \right), \kappa^{-1}\alpha_1^2 \log^{2\gamma} \left( (1 + \tilde{\kappa}/\kappa)\tilde{\kappa}^{-1}\alpha_1^2 \right), \alpha_2^2, (\tilde{\kappa}/\kappa)^2\alpha_2^2, \right.$$

$$\left. \tilde{\kappa}\alpha_3^2, (\tilde{\kappa}^3/\kappa^2)\alpha_3^2, \tilde{\kappa}^3\kappa^{-2} \|\mathcal{I}_T^{\frac{1}{2}} \mathcal{I}_S^{-1} \mathcal{I}_T^{\frac{1}{2}}\|_2^{-2}\alpha_3^2 \right\}$$

$$\leq (1 + \tilde{\kappa}/\kappa)^2 \cdot \max\{\tilde{\kappa}^{-1}\alpha_1^2 \log^{2\gamma} \left( (1 + \tilde{\kappa}/\kappa)\tilde{\kappa}^{-1}\alpha_1^2 \right), \alpha_2^2, \tilde{\kappa}(1 + \|\mathcal{I}_T^{\frac{1}{2}} \mathcal{I}_S^{-1} \mathcal{I}_T^{\frac{1}{2}}\|_2^{-2})\alpha_3^2\}$$

$$=: N^\star.$$

To summarize, for any $\delta$, any $n \geq c \max\{N^\star \log(d/\delta), N(\delta)\}$, with probability at least $1 - 2\delta$, we have

$$R_{\beta^\star}(\beta_{\mathsf{MLE}}) \leq c \frac{\mathsf{Tr}(\mathcal{I}_T \mathcal{I}_S^{-1}) \log \frac{d}{\delta}}{n}.$$

$\square$

In the following, we prove Lemma A.2 and A.3.

**Proof of Lemma A.2**

*Proof of Lemma A.2.* For notation simplicity, we denote $g := \nabla \ell_n(\beta^\star) - \mathbb{E}[\nabla \ell_n(\beta^\star)]$. Note that

$$
\begin{aligned}
V &= n \cdot \mathbb{E}[\|A(\nabla \ell_n(\beta^\star) - \mathbb{E}[\nabla \ell_n(\beta^\star)])\|_2^2] \\
&= n \cdot \mathbb{E}[\nabla \ell_n(\beta^\star)^T A^T A \nabla \ell_n(\beta^\star)] \\
&= n \cdot \mathbb{E}[\mathsf{Tr}(A \nabla \ell_n(\beta^\star) \nabla \ell_n(\beta^\star)^T A^T)] \\
&= \mathsf{Tr}(A \mathcal{I}_S A^T).
\end{aligned}
$$

By taking $A = \mathcal{I}_S^{-1}$ in Assumption A.1, for any $\delta$, any $n > N(\delta)$, we have with probability at least $1 - \delta$:

$$
\begin{aligned}
\|\mathcal{I}_S^{-1} g\|_2 &\leq c \sqrt{\frac{\mathsf{Tr}(\mathcal{I}_S^{-1}) \log \frac{d}{\delta}}{n}} + B_1 \|\mathcal{I}_S^{-1}\|_2 \log^\gamma \left( \frac{B_1 \|\mathcal{I}_S^{-1}\|_2}{\sqrt{\mathsf{Tr}(\mathcal{I}_S^{-1})}} \right) \frac{\log \frac{d}{\delta}}{n} \\
&= c \sqrt{\frac{\mathsf{Tr}(\mathcal{I}_S^{-1}) \log \frac{d}{\delta}}{n}} + B_1 \|\mathcal{I}_S^{-1}\|_2 \log^\gamma(\tilde{\kappa}^{-1/2} \alpha_1) \frac{\log \frac{d}{\delta}}{n}, \quad (8)
\end{aligned}
$$

$$
\left\| \nabla^2 \ell_n(\beta^\star) - \mathbb{E}[\nabla^2 \ell_n(\beta^\star)] \right\|_2 \leq B_2 \sqrt{\frac{\log \frac{d}{\delta}}{n}}. \quad (9)
$$

Let event $A := \{(8), (9) \text{ holds}\}$. Under the event $A$, we have the following Taylor expansion:

$$
\begin{aligned}
\ell_n(\beta) - \ell_n(\beta^\star) &\overset{\text{by Assumption A.2}}{\leq} (\beta - \beta^\star)^T \nabla \ell_n(\beta^\star) + \frac{1}{2}(\beta - \beta^\star)^T \nabla^2 \ell_n(\beta^\star)(\beta - \beta^\star) + \frac{B_3}{6}\|\beta - \beta^\star\|_2^3 \\
&\overset{\nabla \ell(\beta^\star)=0}{=} (\beta - \beta^\star)^T g + \frac{1}{2}(\beta - \beta^\star)^T \nabla^2 \ell_n(\beta^\star)(\beta - \beta^\star) + \frac{B_3}{6}\|\beta - \beta^\star\|_2^3 \\
&\overset{\text{by (9)}}{\leq} (\beta - \beta^\star)^T g + \frac{1}{2}(\beta - \beta^\star)^T \mathcal{I}_S(\beta - \beta^\star) + B_2 \sqrt{\frac{\log \frac{d}{\delta}}{n}} \|\beta - \beta^\star\|_2^2 + \frac{B_3}{6}\|\beta - \beta^\star\|_2^3 \\
&\overset{\Delta_\beta := \beta - \beta^\star}{=} \Delta_\beta^T g + \frac{1}{2}\Delta_\beta^T \mathcal{I}_S \Delta_\beta + B_2 \sqrt{\frac{\log \frac{d}{\delta}}{n}} \|\Delta_\beta\|_2^2 + \frac{B_3}{6}\|\Delta_\beta\|_2^3 \\
&= \frac{1}{2}(\Delta_\beta - z)^T \mathcal{I}_S (\Delta_\beta - z) - \frac{1}{2} z^T \mathcal{I}_S z + B_2 \sqrt{\frac{\log \frac{d}{\delta}}{n}} \|\Delta_\beta\|_2^2 + \frac{B_3}{6}\|\Delta_\beta\|_2^3 \quad (10)
\end{aligned}
$$

where $z := -\mathcal{I}_S^{-1} g$. Similarly

$$
\ell_n(\beta) - \ell_n(\beta^\star) \geq \frac{1}{2}(\Delta_\beta - z)^T \mathcal{I}_S (\Delta_\beta - z) - \frac{1}{2} z^T \mathcal{I}_S z - B_2 \sqrt{\frac{\log \frac{d}{\delta}}{n}} \|\Delta_\beta\|_2^2 - \frac{B_3}{6}\|\Delta_\beta\|_2^3. \tag{11}
$$

Notice that $\Delta_{\beta^\star + z} = z$, by (8) and (10), we have

$$
\begin{aligned}
\ell_n(\beta^\star + z) - \ell_n(\beta^\star) &\leq -\frac{1}{2} z^T \mathcal{I}_S z + B_2 \sqrt{\frac{\log \frac{d}{\delta}}{n}} \left( c \sqrt{\frac{\mathsf{Tr}(\mathcal{I}_S^{-1}) \log \frac{d}{\delta}}{n}} + B_1 \|\mathcal{I}_S^{-1}\|_2 \log^\gamma(\tilde{\kappa}^{-1/2} \alpha_1) \frac{\log \frac{d}{\delta}}{n} \right)^2 \\
&\quad + \frac{B_3}{6} \left( c \sqrt{\frac{\mathsf{Tr}(\mathcal{I}_S^{-1}) \log \frac{d}{\delta}}{n}} + B_1 \|\mathcal{I}_S^{-1}\|_2 \log^\gamma(\tilde{\kappa}^{-1/2} \alpha_1) \frac{\log \frac{d}{\delta}}{n} \right)^3 \\
&\leq -\frac{1}{2} z^T \mathcal{I}_S z + 2 c^2 B_2 \mathsf{Tr}(\mathcal{I}_S^{-1}) (\frac{\log \frac{d}{\delta}}{n})^{1.5} + 2 B_1^2 B_2 \|\mathcal{I}_S^{-1}\|_2^2 \log^{2\gamma}(\tilde{\kappa}^{-1/2} \alpha_1)(\frac{\log \frac{d}{\delta}}{n})^{2.5} \\
&\quad + \frac{2}{3} c^3 B_3 \mathsf{Tr}(\mathcal{I}_S^{-1})^{1.5}(\frac{\log \frac{d}{\delta}}{n})^{1.5} + \frac{2}{3} B_1^3 B_3 \|\mathcal{I}_S^{-1}\|_2^3 \log^{3\gamma}(\tilde{\kappa}^{-1/2} \alpha_1)(\frac{\log \frac{d}{\delta}}{n})^3, \\
&\hspace{12cm} (12)
\end{aligned}
$$

where we use the fact that $(a + b)^n \leq 2^{n-1}(a^n + b^n)$ in the last inequality. For any $\beta \in \mathbb{B}_{\beta^\star}(3c\sqrt{\frac{\mathsf{Tr}(\mathcal{I}_S^{-1})\log\frac{d}{\delta}}{n}})$, by (11), we have

$$\ell_n(\beta) - \ell_n(\beta^\star) \geq \frac{1}{2}(\Delta_\beta - z)^T\mathcal{I}_S(\Delta_\beta - z) - \frac{1}{2}z^T\mathcal{I}_S z$$
$$- 9c^2 B_2 \mathsf{Tr}(\mathcal{I}_S^{-1})(\frac{\log\frac{d}{\delta}}{n})^{1.5} - \frac{9}{2}c^3 B_3 \mathsf{Tr}(\mathcal{I}_S^{-1})^{1.5}(\frac{\log\frac{d}{\delta}}{n})^{1.5}. \tag{13}$$

(13) - (12) gives

$$\ell_n(\beta) - \ell_n(\beta^\star + z) \geq \frac{1}{2}(\Delta_\beta - z)^T\mathcal{I}_S(\Delta_\beta - z)$$
$$- \Big(9c^2 B_2 \mathsf{Tr}(\mathcal{I}_S^{-1})(\frac{\log\frac{d}{\delta}}{n})^{1.5} + \frac{9}{2}c^3 B_3 \mathsf{Tr}(\mathcal{I}_S^{-1})^{1.5}(\frac{\log\frac{d}{\delta}}{n})^{1.5}$$
$$+ 2c^2 B_2 \mathsf{Tr}(\mathcal{I}_S^{-1})(\frac{\log\frac{d}{\delta}}{n})^{1.5} + 2B_1^2 B_2\|\mathcal{I}_S^{-1}\|_2^2 \log^{2\gamma}(\tilde{\kappa}^{-1/2}\alpha_1)(\frac{\log\frac{d}{\delta}}{n})^{2.5}$$
$$+ \frac{2}{3}c^3 B_3 \mathsf{Tr}(\mathcal{I}_S^{-1})^{1.5}(\frac{\log\frac{d}{\delta}}{n})^{1.5} + \frac{2}{3}B_1^3 B_3\|\mathcal{I}_S^{-1}\|_2^3 \log^{3\gamma}(\tilde{\kappa}^{-1/2}\alpha_1)(\frac{\log\frac{d}{\delta}}{n})^3\Big)$$
$$= \frac{1}{2}(\Delta_\beta - z)^T\mathcal{I}_S(\Delta_\beta - z)$$
$$- \Big(11c^2 B_2 \mathsf{Tr}(\mathcal{I}_S^{-1})(\frac{\log\frac{d}{\delta}}{n})^{1.5} + \frac{31}{6}c^3 B_3 \mathsf{Tr}(\mathcal{I}_S^{-1})^{1.5}(\frac{\log\frac{d}{\delta}}{n})^{1.5}$$
$$+ 2B_1^2 B_2\|\mathcal{I}_S^{-1}\|_2^2 \log^{2\gamma}(\tilde{\kappa}^{-1/2}\alpha_1)(\frac{\log\frac{d}{\delta}}{n})^{2.5} + \frac{2}{3}B_1^3 B_3\|\mathcal{I}_S^{-1}\|_2^3 \log^{3\gamma}(\tilde{\kappa}^{-1/2}\alpha_1)(\frac{\log\frac{d}{\delta}}{n})^3\Big) \tag{14}$$

Consider the ellipsoid

$$\mathcal{D} := \Big\{\beta \in \mathbb{R}^d \Big| \frac{1}{2}(\Delta_\beta - z)^T\mathcal{I}_S(\Delta_\beta - z)$$
$$\leq 11c^2 B_2 \mathsf{Tr}(\mathcal{I}_S^{-1})(\frac{\log\frac{d}{\delta}}{n})^{1.5} + \frac{31}{6}c^3 B_3 \mathsf{Tr}(\mathcal{I}_S^{-1})^{1.5}(\frac{\log\frac{d}{\delta}}{n})^{1.5}$$
$$+ 2B_1^2 B_2\|\mathcal{I}_S^{-1}\|_2^2 \log^{2\gamma}(\tilde{\kappa}^{-1/2}\alpha_1)(\frac{\log\frac{d}{\delta}}{n})^{2.5} + \frac{2}{3}B_1^3 B_3\|\mathcal{I}_S^{-1}\|_2^3 \log^{3\gamma}(\tilde{\kappa}^{-1/2}\alpha_1)(\frac{\log\frac{d}{\delta}}{n})^3\Big\}.$$

Then by (14), for any $\beta \in \mathbb{B}_{\beta^\star}(3c\sqrt{\frac{\mathsf{Tr}(\mathcal{I}_S^{-1})\log\frac{d}{\delta}}{n}}) \cap \mathcal{D}^C$,

$$\ell_n(\beta) - \ell_n(\beta^\star + z) > 0. \tag{15}$$

Notice that by the definition of $\mathcal{D}$, using $\lambda_{\min}^{-1}(\mathcal{I}_S) = \|\mathcal{I}_S^{-1}\|_2$, we have for any $\beta \in \mathcal{D}$,

$$\|\Delta_\beta - z\|_2^2 \leq 22c^2 B_2\|\mathcal{I}_S^{-1}\|_2 \mathsf{Tr}(\mathcal{I}_S^{-1})(\frac{\log\frac{d}{\delta}}{n})^{1.5} + \frac{31}{3}c^3 B_3\|\mathcal{I}_S^{-1}\|_2 \mathsf{Tr}(\mathcal{I}_S^{-1})^{1.5}(\frac{\log\frac{d}{\delta}}{n})^{1.5}$$
$$+ 4B_1^2 B_2\|\mathcal{I}_S^{-1}\|_2^3 \log^{2\gamma}(\tilde{\kappa}^{-1/2}\alpha_1)(\frac{\log\frac{d}{\delta}}{n})^{2.5} + \frac{4}{3}B_1^3 B_3\|\mathcal{I}_S^{-1}\|_2^4 \log^{3\gamma}(\tilde{\kappa}^{-1/2}\alpha_1)(\frac{\log\frac{d}{\delta}}{n})^3.$$

Thus for any $\beta \in \mathcal{D}$, we have

$$\|\Delta_\beta\|_2^2 \leq 2(\|\Delta_\beta - z\|_2^2 + \|z\|_2^2)$$
$$\overset{\text{by}(8)}{\leq} 44c^2 B_2\|\mathcal{I}_S^{-1}\|_2 \mathsf{Tr}(\mathcal{I}_S^{-1})(\frac{\log\frac{d}{\delta}}{n})^{1.5} + \frac{62}{3}c^3 B_3\|\mathcal{I}_S^{-1}\|_2 \mathsf{Tr}(\mathcal{I}_S^{-1})^{1.5}(\frac{\log\frac{d}{\delta}}{n})^{1.5}$$
$$+ 8B_1^2 B_2\|\mathcal{I}_S^{-1}\|_2^3 \log^{2\gamma}(\tilde{\kappa}^{-1/2}\alpha_1)(\frac{\log\frac{d}{\delta}}{n})^{2.5} + \frac{8}{3}B_1^3 B_3\|\mathcal{I}_S^{-1}\|_2^4 \log^{3\gamma}(\tilde{\kappa}^{-1/2}\alpha_1)(\frac{\log\frac{d}{\delta}}{n})^3$$
$$+ 4c^2 \frac{\mathsf{Tr}(\mathcal{I}_S^{-1})\log\frac{d}{\delta}}{n} + 4B_1^2\|\mathcal{I}_S^{-1}\|_2^2 \log^{2\gamma}(\tilde{\kappa}^{-1/2}\alpha_1)(\frac{\log\frac{d}{\delta}}{n})^2.$$

To guarantee $\frac{\mathsf{Tr}(\mathcal{I}_S^{-1})\log\frac{d}{\delta}}{n}$ is the leading term, we only need $\frac{\mathsf{Tr}(\mathcal{I}_S^{-1})\log\frac{d}{\delta}}{n}$ to dominate the rest of the terms. Hence, if we further have $n \geq cN_1\log(d/\delta)$, it then holds that

$$\|\Delta_\beta\|_2^2 \leq 9c^2\frac{\mathsf{Tr}(\mathcal{I}_S^{-1})\log\frac{d}{\delta}}{n},$$

i.e., $\beta \in \mathbb{B}_{\beta^\star}(3c\sqrt{\frac{\mathsf{Tr}(\mathcal{I}_S^{-1})\log\frac{d}{\delta}}{n}})$. Here

$$N_1 := \max\left\{ B_2^2\|\mathcal{I}_S^{-1}\|_2^2, B_3^2\|\mathcal{I}_S^{-1}\|_2^2\mathsf{Tr}(\mathcal{I}_S^{-1}), \left(\frac{B_1^2 B_2\|\mathcal{I}_S^{-1}\|_2^3\log^{2\gamma}(\tilde{\kappa}^{-1/2}\alpha_1)}{\mathsf{Tr}(\mathcal{I}_S^{-1})}\right)^{\frac{2}{3}}, \right.$$
$$\left. \left(\frac{B_1^3 B_3\|\mathcal{I}_S^{-1}\|_2^4\log^{3\gamma}(\tilde{\kappa}^{-1/2}\alpha_1)}{\mathsf{Tr}(\mathcal{I}_S^{-1})}\right)^{\frac{1}{2}}, \frac{B_1^2\|\mathcal{I}_S^{-1}\|_2^2\log^{2\gamma}(\tilde{\kappa}^{-1/2}\alpha_1)}{\mathsf{Tr}(\mathcal{I}_S^{-1})}\right\}.$$

In other words, we show that $\mathcal{D} \subset \mathbb{B}_{\beta^\star}(3c\sqrt{\frac{\mathsf{Tr}(\mathcal{I}_S^{-1})\log\frac{d}{\delta}}{n}})$ when $n \geq c\max\{N_1\log(d/\delta), N(\delta)\}$. Recall that by (15), we know that for any $\beta \in \mathbb{B}_{\beta^\star}(3c\sqrt{\frac{\mathsf{Tr}(\mathcal{I}_S^{-1})\log\frac{d}{\delta}}{n}}) \cap \mathcal{D}^C$,

$$\ell_n(\beta) - \ell_n(\beta^\star + z) > 0.$$

Note that $\beta^\star + z \in \mathcal{D}$. Hence there is a local minimum of $\ell_n(\beta)$ in $\mathcal{D}$. By Assumption A.3, we know that the global minimum of $\ell_n(\beta)$ is in $\mathcal{D}$, i.e.,

$$\beta_{\mathsf{MLE}} \in \mathcal{D} \subset \mathbb{B}_{\beta^\star}(3c\sqrt{\frac{\mathsf{Tr}(\mathcal{I}_S^{-1})\log\frac{d}{\delta}}{n}}).$$

$\square$

**Proof of Lemma A.3**

*Proof of Lemma A.3.* Let $E := \{\beta_{\mathsf{MLE}} \in \mathcal{D} \subset \mathbb{B}_{\beta^\star}(c\sqrt{\frac{\mathsf{Tr}(\mathcal{I}_S^{-1})\log\frac{d}{\delta}}{n}})\}$. For any $\delta$ and any $n \geq c\max\{N_1\log(d/\delta), N(\delta)\}$, by the proof of Lemma A.2, we have $\mathbb{P}(E) \geq 1 - \delta$.

By taking $A = \mathcal{I}_T^{\frac{1}{2}}\mathcal{I}_S^{-1}$ in Assumption A.1, for any $\delta$, any $n > N(\delta)$, we have with probability at least $1 - \delta$:

$$\|\mathcal{I}_T^{\frac{1}{2}}\mathcal{I}_S^{-1}g\|_2 \leq c\sqrt{\frac{\mathsf{Tr}(\mathcal{I}_S^{-1}\mathcal{I}_T)\log\frac{d}{\delta}}{n}} + B_1\|\mathcal{I}_T^{\frac{1}{2}}\mathcal{I}_S^{-1}\|_2\log^\gamma\left(\frac{B_1\|\mathcal{I}_T^{\frac{1}{2}}\mathcal{I}_S^{-1}\|_2}{\sqrt{\mathsf{Tr}(\mathcal{I}_S^{-1}\mathcal{I}_T)}}\right)\frac{\log\frac{d}{\delta}}{n}$$

$$\leq c\sqrt{\frac{\mathsf{Tr}(\mathcal{I}_S^{-1}\mathcal{I}_T)\log\frac{d}{\delta}}{n}} + B_1\|\mathcal{I}_T^{\frac{1}{2}}\mathcal{I}_S^{-1}\|_2\log^\gamma(\kappa^{-1/2}\alpha_1)\frac{\log\frac{d}{\delta}}{n}. \quad (16)$$

We denote $E' := \{(16)$ holds$\}$. For any $\delta$ and any $n \geq c\max\{N_1\log(d/\delta), N(\delta)\}$, we have $\mathbb{P}(E \cap E') \geq 1 - 2\delta$.

Under $E \cap E'$, $\beta_{\mathsf{MLE}} \in \mathcal{D}$, i.e.,

$$\frac{1}{2}(\Delta_{\beta_{\mathsf{MLE}}} - z)^T\mathcal{I}_S(\Delta_{\beta_{\mathsf{MLE}}} - z)$$

$$\leq 11c^2 B_2\mathsf{Tr}(\mathcal{I}_S^{-1})(\frac{\log\frac{d}{\delta}}{n})^{1.5} + \frac{31}{6}c^3 B_3\mathsf{Tr}(\mathcal{I}_S^{-1})^{1.5}(\frac{\log\frac{d}{\delta}}{n})^{1.5}$$

$$+ 2B_1^2 B_2\|\mathcal{I}_S^{-1}\|_2^2\log^{2\gamma}(\tilde{\kappa}^{-1/2}\alpha_1)(\frac{\log\frac{d}{\delta}}{n})^{2.5} + \frac{2}{3}B_1^3 B_3\|\mathcal{I}_S^{-1}\|_2^3\log^{3\gamma}(\tilde{\kappa}^{-1/2}\alpha_1)(\frac{\log\frac{d}{\delta}}{n})^3.$$

In other words,

$$\|\mathcal{I}_S^{\frac{1}{2}}(\Delta_{\beta_{\mathsf{MLE}}} - z)\|_2^2$$

$$\leq 22c^2 B_2\mathsf{Tr}(\mathcal{I}_S^{-1})(\frac{\log\frac{d}{\delta}}{n})^{1.5} + \frac{31}{3}c^3 B_3\mathsf{Tr}(\mathcal{I}_S^{-1})^{1.5}(\frac{\log\frac{d}{\delta}}{n})^{1.5}$$

$$+ 4B_1^2 B_2\|\mathcal{I}_S^{-1}\|_2^2\log^{2\gamma}(\tilde{\kappa}^{-1/2}\alpha_1)(\frac{\log\frac{d}{\delta}}{n})^{2.5} + \frac{4}{3}B_1^3 B_3\|\mathcal{I}_S^{-1}\|_2^3\log^{3\gamma}(\tilde{\kappa}^{-1/2}\alpha_1)(\frac{\log\frac{d}{\delta}}{n})^3 \quad (17)$$

Thus we have

$$\|\mathcal{I}_T^{\frac{1}{2}}(\beta_{\mathsf{MLE}} - \beta^\star)\|_2^2$$

$$= \|\mathcal{I}_T^{\frac{1}{2}}\Delta_{\beta_{\mathsf{MLE}}}\|_2^2$$

$$= \|\mathcal{I}_T^{\frac{1}{2}}(\Delta_{\beta_{\mathsf{MLE}}} - z) + \mathcal{I}_T^{\frac{1}{2}}z\|_2^2$$

$$\leq 2\|\mathcal{I}_T^{\frac{1}{2}}(\Delta_{\beta_{\mathsf{MLE}}} - z)\|_2^2 + 2\|\mathcal{I}_T^{\frac{1}{2}}z\|_2^2$$

$$= 2\|\mathcal{I}_T^{\frac{1}{2}}\mathcal{I}_S^{-\frac{1}{2}}(\mathcal{I}_S^{\frac{1}{2}}(\Delta_{\beta_{\mathsf{MLE}}} - z))\|_2^2 + 2\|\mathcal{I}_T^{\frac{1}{2}}\mathcal{I}_S^{-1}g\|_2^2$$

$$\leq 2\|\mathcal{I}_T^{\frac{1}{2}}\mathcal{I}_S^{-\frac{1}{2}}\|_2^2\|\mathcal{I}_S^{\frac{1}{2}}(\Delta_{\beta_{\mathsf{MLE}}} - z)\|_2^2 + 2\|\mathcal{I}_T^{\frac{1}{2}}\mathcal{I}_S^{-1}g\|_2^2$$

$$\overset{\text{by}(17)\text{and}(16)}{\leq} 4c^2\frac{\mathsf{Tr}(\mathcal{I}_T\mathcal{I}_S^{-1})\log\frac{d}{\delta}}{n}$$

$$+ 44c^2 B_2\|\mathcal{I}_T^{\frac{1}{2}}\mathcal{I}_S^{-\frac{1}{2}}\|_2^2\mathsf{Tr}(\mathcal{I}_S^{-1})(\frac{\log\frac{d}{\delta}}{n})^{1.5} + \frac{62}{3}c^3 B_3\|\mathcal{I}_T^{\frac{1}{2}}\mathcal{I}_S^{-\frac{1}{2}}\|_2^2\mathsf{Tr}(\mathcal{I}_S^{-1})^{1.5}(\frac{\log\frac{d}{\delta}}{n})^{1.5}$$

$$+ 8B_1^2 B_2\|\mathcal{I}_T^{\frac{1}{2}}\mathcal{I}_S^{-\frac{1}{2}}\|_2^2\|\mathcal{I}_S^{-1}\|_2^2\log^{2\gamma}(\tilde{\kappa}^{-1/2}\alpha_1)(\frac{\log\frac{d}{\delta}}{n})^{2.5} + \frac{8}{3}B_1^3 B_3\|\mathcal{I}_T^{\frac{1}{2}}\mathcal{I}_S^{-\frac{1}{2}}\|_2^2\|\mathcal{I}_S^{-1}\|_2^3\log^{3\gamma}(\tilde{\kappa}^{-1/2}\alpha_1)(\frac{\log\frac{d}{\delta}}{n})^3$$

$$+ 4B_1^2\|\mathcal{I}_T^{\frac{1}{2}}\mathcal{I}_S^{-\frac{1}{2}}\|_2^2\|\mathcal{I}_S^{-1}\|_2\log^{2\gamma}(\kappa^{-1/2}\alpha_1)(\frac{\log\frac{d}{\delta}}{n})^2$$

To guarantee $\frac{\mathsf{Tr}(\mathcal{I}_T\mathcal{I}_S^{-1})\log\frac{d}{\delta}}{n}$ is the leading term, we only need $\frac{\mathsf{Tr}(\mathcal{I}_T\mathcal{I}_S^{-1})\log\frac{d}{\delta}}{n}$ to dominate the rest of the terms. Hence, if we further have $n \geq cN_2\log(d/\delta)$, we have

$$\|\mathcal{I}_T^{\frac{1}{2}}(\beta_{\mathsf{MLE}} - \beta^\star)\|_2^2 \leq 9c^2\frac{\mathsf{Tr}(\mathcal{I}_T\mathcal{I}_S^{-1})\log\frac{d}{\delta}}{n}.$$

Here

$$N_2 := \max\left\{ \left(\frac{B_2\|\mathcal{I}_T^{\frac{1}{2}}\mathcal{I}_S^{-\frac{1}{2}}\|_2^2\mathsf{Tr}(\mathcal{I}_S^{-1})}{\mathsf{Tr}(\mathcal{I}_T\mathcal{I}_S^{-1})}\right)^2, \left(\frac{B_3\|\mathcal{I}_T^{\frac{1}{2}}\mathcal{I}_S^{-\frac{1}{2}}\|_2^2\mathsf{Tr}(\mathcal{I}_S^{-1})^{1.5}}{\mathsf{Tr}(\mathcal{I}_T\mathcal{I}_S^{-1})}\right)^2, \right.$$

$$\left(\frac{B_1^2 B_2\|\mathcal{I}_T^{\frac{1}{2}}\mathcal{I}_S^{-\frac{1}{2}}\|_2^2\|\mathcal{I}_S^{-1}\|_2^2\log^{2\gamma}(\tilde{\kappa}^{-1/2}\alpha_1)}{\mathsf{Tr}(\mathcal{I}_T\mathcal{I}_S^{-1})}\right)^{\frac{2}{3}}, \left(\frac{B_1^3 B_3\|\mathcal{I}_T^{\frac{1}{2}}\mathcal{I}_S^{-\frac{1}{2}}\|_2^2\|\mathcal{I}_S^{-1}\|_2^3\log^{3\gamma}(\tilde{\kappa}^{-1/2}\alpha_1)}{\mathsf{Tr}(\mathcal{I}_T\mathcal{I}_S^{-1})}\right)^{\frac{1}{2}},$$

$$\left. \frac{B_1^2\|\mathcal{I}_T^{\frac{1}{2}}\mathcal{I}_S^{-\frac{1}{2}}\|_2^2\|\mathcal{I}_S^{-1}\|_2\log^{2\gamma}(\kappa^{-1/2}\alpha_1)}{\mathsf{Tr}(\mathcal{I}_T\mathcal{I}_S^{-1})}\right\}.$$

To summarize, we show that for any $\delta \in (0,1)$ and any $n \geq c\max\{N_1\log(d/\delta), N_2\log(d/\delta), N(\delta)\}$, with probability at least $1 - 2\delta$, we have

$$\|\mathcal{I}_T^{\frac{1}{2}}(\beta_{\mathsf{MLE}} - \beta^\star)\|_2^2 \leq 9c^2\frac{\mathsf{Tr}(\mathcal{I}_T\mathcal{I}_S^{-1})\log\frac{d}{\delta}}{n}.$$

$\square$

## A.2 PROOFS FOR THEOREM 3.2

The detailed version of Theorem 3.2 is stated as the following.

**Theorem A.4.** *Suppose the model class $\mathcal{F}$ satisfies Assumption B. Then we have*

$$\inf_{\hat{\beta}} \sup_{\beta^\star \in \mathbb{B}_{\beta_0}(B)} \mathsf{Tr}\left(\mathcal{I}_T(\beta^\star)\mathcal{I}_S^{-1}(\beta^\star)\right)^{-1} \mathbb{E}_{\substack{x_i\sim\mathbb{P}_S(X) \\ y_i|x_i\sim f(y|x;\beta^\star)}}\left[R_{\beta^\star}(\hat{\beta})\right]$$

$$\geq \frac{1}{16}\cdot\frac{1}{2n + \frac{\pi^2 d}{R_1^2}\mathsf{Tr}\left(\mathcal{I}_T(\beta_0)\mathcal{I}_S^{-2}(\beta_0)\right)\mathsf{Tr}\left(\mathcal{I}_T(\beta_0)\mathcal{I}_S^{-1}(\beta_0)\right)^{-1}},$$

*where*

$$R_1 := \frac{1}{4}\sqrt{\frac{\lambda_{\min}(\mathcal{I}_T(\beta_0))}{\lambda_{\max}(\mathcal{I}_T(\beta_0))}}\cdot\min\left\{\frac{\lambda_{\min}^2(\mathcal{I}_S(\beta_0))}{4L_S\lambda_{\max}(\mathcal{I}_S(\beta_0))}, \frac{\lambda_{\min}(\mathcal{I}_T(\beta_0))}{4B_3 + 2L_T}, B\right\}.$$

We first present some useful lemmas that will be used in the proof of Theorem A.4.

**Lemma A.5.** *Under Assumptions A.2, B.2 and B.3, we can choose $R_0 \leq B$ such that for any $\beta, \beta^\star \in \mathbb{B}_{\beta_0}(R_0)$:*

$$\frac{1}{2} \cdot \mathcal{I}_S^{-1}(\beta_0) \preceq \mathcal{I}_S^{-1}(\beta) \preceq 2 \cdot \mathcal{I}_S^{-1}(\beta_0), \tag{18}$$

$$\frac{1}{2} \cdot \mathcal{I}_T(\beta_0) \preceq \mathbb{E}_{\substack{x \sim \mathbb{P}_T(X) \\ y|x \sim f(y|x;\beta^\star)}} \left[ \nabla^2 \ell(x,y,\beta) \right] \preceq 2 \cdot \mathcal{I}_T(\beta_0). \tag{19}$$

*We can further choose $R_1 \leq R_0$ such that for any $\beta^\star \in \mathbb{B}_{\beta_0}(R_1), \beta \notin \mathbb{B}_{\beta_0}(R_0)$: $R_{\beta^\star}(\beta) \geq R_{\beta^\star}(\beta_0)$.*

Taking $\beta^\star = \beta$, Lemma A.5 (19) implies for any $\beta \in \mathbb{B}_{\beta_0}(R_0)$:

$$\frac{1}{2} \cdot \mathcal{I}_T(\beta_0) \preceq \mathcal{I}_T(\beta) \preceq 2 \cdot \mathcal{I}_T(\beta_0). \tag{20}$$

**Lemma A.6.** *Let $C_{\beta_0}(B) := \{\beta \in \mathbb{R}^d \mid \beta - \beta_0 \in [-B, B]^d\}$ be a cube around $\beta_0$. For any $\beta_0 \in \mathbb{R}^d$ and $B > 0$, there exists a prior density $\lambda(\beta)$ supported on $C_{\beta_0}(B)$ such that for any estimator $\hat{\beta}$, we have*

$$\mathbb{E}_{\beta^\star \sim \lambda(\beta)} \mathbb{E}_{\substack{x_i \sim \mathbb{P}_S(X) \\ y_i|x_i \sim f(y|x;\beta^\star)}} \left[ (\hat{\beta} - \beta^\star)^T \mathcal{I}_T(\beta_0)(\hat{\beta} - \beta^\star) \right]$$

$$\geq \frac{\mathsf{Tr}\left( \mathcal{I}_T(\beta_0)\mathcal{I}_S^{-1}(\beta_0) \right)^2}{n \mathbb{E}_{\beta^\star \sim \lambda(\beta)} \left[ \mathsf{Tr}\left( \mathcal{I}_S^{-1}(\beta_0)\mathcal{I}_S(\beta^\star)\mathcal{I}_S^{-1}(\beta_0)\mathcal{I}_T(\beta_0) \right) \right] + \frac{\pi^2}{B^2} \mathsf{Tr}\left( \mathcal{I}_T(\beta_0)\mathcal{I}_S^{-2}(\beta_0) \right)}$$

The proofs for the above lemmas are delivered to the end of this subsection. With Lemma A.5 and Lemma A.6 in hand, we are now ready to prove Theorem A.4.

*Proof of Theorem A.4.* For any estimator $\hat{\beta}$, we define

$$\hat{\beta}^p := \begin{cases} \hat{\beta} & \hat{\beta} \in \mathbb{B}_{\beta_0}(R_0) \\ \beta_0 & \hat{\beta} \notin \mathbb{B}_{\beta_0}(R_0). \end{cases}$$

By Lemma A.5, for any $\beta^\star \in \mathbb{B}_{\beta_0}(R_1)$, we have $R_{\beta^\star}(\hat{\beta}) \geq R_{\beta^\star}(\hat{\beta}^p)$. We then have

$$\inf_{\hat{\beta}} \sup_{\beta^\star \in \mathbb{B}_{\beta_0}(B)} \mathsf{Tr}\left( \mathcal{I}_T(\beta^\star)\mathcal{I}_S^{-1}(\beta^\star) \right)^{-1} \mathbb{E}_{\substack{x_i \sim \mathbb{P}_S(X) \\ y_i|x_i \sim f(y|x;\beta^\star)}} \left[ R_{\beta^\star}(\hat{\beta}) \right]$$

$$\geq \inf_{\hat{\beta}} \sup_{\beta^\star \in \mathbb{B}_{\beta_0}(R_1)} \mathsf{Tr}\left( \mathcal{I}_T(\beta^\star)\mathcal{I}_S^{-1}(\beta^\star) \right)^{-1} \mathbb{E}_{\substack{x_i \sim \mathbb{P}_S(X) \\ y_i|x_i \sim f(y|x;\beta^\star)}} \left[ R_{\beta^\star}(\hat{\beta}) \right]$$

$$\geq \inf_{\hat{\beta}^p} \sup_{\beta^\star \in \mathbb{B}_{\beta_0}(R_1)} \mathsf{Tr}\left( \mathcal{I}_T(\beta^\star)\mathcal{I}_S^{-1}(\beta^\star) \right)^{-1} \mathbb{E}_{\substack{x_i \sim \mathbb{P}_S(X) \\ y_i|x_i \sim f(y|x;\beta^\star)}} \left[ R_{\beta^\star}(\hat{\beta}^p) \right]$$

$$\geq \inf_{\hat{\beta} \in \mathbb{B}_{\beta_0}(R_0)} \sup_{\beta^\star \in \mathbb{B}_{\beta_0}(R_1)} \mathsf{Tr}\left( \mathcal{I}_T(\beta^\star)\mathcal{I}_S^{-1}(\beta^\star) \right)^{-1} \mathbb{E}_{\substack{x_i \sim \mathbb{P}_S(X) \\ y_i|x_i \sim f(y|x;\beta^\star)}} \left[ R_{\beta^\star}(\hat{\beta}) \right], \tag{21}$$

where the first inequality follows from the fact that $R_1 \leq R_0 \leq B$, the second inequality follows from $R_{\beta^\star}(\hat{\beta}) \geq R_{\beta^\star}(\hat{\beta}^p)$, and the third inequality follows from $\hat{\beta}^p \in \mathbb{B}_{\beta_0}(R_0)$. For any $\beta^\star \in \mathbb{B}_{\beta_0}(R_1) \subseteq \mathbb{B}_{\beta_0}(R_0)$, by (18) and (20), we have

$$\mathcal{I}_T(\beta^\star) \preceq 2\mathcal{I}_T(\beta_0), \quad \mathcal{I}_S^{-1}(\beta^\star) \preceq 2\mathcal{I}_S^{-1}(\beta_0),$$

which implies

$$\mathsf{Tr}\left( \mathcal{I}_T(\beta^\star)\mathcal{I}_S^{-1}(\beta^\star) \right)^{-1} \geq \frac{1}{4}\mathsf{Tr}\left( \mathcal{I}_T(\beta_0)\mathcal{I}_S^{-1}(\beta_0) \right)^{-1}. \tag{22}$$

Combine (21) and (22), we have

$$\inf_{\hat{\beta}} \sup_{\beta^\star \in \mathbb{B}_{\beta_0}(B)} \mathsf{Tr}\left( \mathcal{I}_T(\beta^\star)\mathcal{I}_S^{-1}(\beta^\star) \right)^{-1} \mathbb{E}_{\substack{x_i \sim \mathbb{P}_S(X) \\ y_i|x_i \sim f(y|x;\beta^\star)}} \left[ R_{\beta^\star}(\hat{\beta}) \right]$$

$$\geq \frac{1}{4}\mathsf{Tr}\left( \mathcal{I}_T(\beta_0)\mathcal{I}_S^{-1}(\beta_0) \right)^{-1} \inf_{\hat{\beta} \in \mathbb{B}_{\beta_0}(R_0)} \sup_{\beta^\star \in \mathbb{B}_{\beta_0}(R_1)} \mathbb{E}_{\substack{x_i \sim \mathbb{P}_S(X) \\ y_i|x_i \sim f(y|x;\beta^\star)}} \left[ R_{\beta^\star}(\hat{\beta}) \right]. \tag{23}$$

By Taylor expansion, for any $\hat{\beta} \in \mathbb{B}_{\beta_0}(R_0), \beta^\star \in \mathbb{B}_{\beta_0}(R_1)$, we have

$$
\begin{aligned}
R_{\beta^\star}(\hat{\beta}) &= R_{\beta^\star}(\beta^\star) + (\hat{\beta} - \beta^\star)^T \mathbb{E}_{\substack{x \sim \mathbb{P}_T(X) \\ y|x \sim f(y|x;\beta^\star)}} \left[ \nabla \ell(x, y, \beta^\star) \right] \\
&\quad + \frac{1}{2} (\hat{\beta} - \beta^\star)^T \mathbb{E}_{\substack{x \sim \mathbb{P}_T(X) \\ y|x \sim f(y|x;\beta^\star)}} \left[ \nabla^2 \ell(x, y, \tilde{\beta}) \right] (\hat{\beta} - \beta^\star) \\
&= \frac{1}{2} (\hat{\beta} - \beta^\star)^T \mathbb{E}_{\substack{x \sim \mathbb{P}_T(X) \\ y|x \sim f(y|x;\beta^\star)}} \left[ \nabla^2 \ell(x, y, \tilde{\beta}) \right] (\hat{\beta} - \beta^\star)
\end{aligned}
$$

for some $\tilde{\beta} \in \mathbb{B}_{\beta_0}(R_0)$. By Lemma A.5 (19), it then holds that

$$
R_{\beta^\star}(\hat{\beta}) \geq \frac{1}{4} (\hat{\beta} - \beta^\star)^T \mathcal{I}_T(\beta_0)(\hat{\beta} - \beta^\star). \tag{24}
$$

By (23) and (24), we then have

$$
\begin{aligned}
&\inf_{\hat{\beta}} \sup_{\beta^\star \in \mathbb{B}_{\beta_0}(B)} \mathsf{Tr} \left( \mathcal{I}_T(\beta^\star) \mathcal{I}_S^{-1}(\beta^\star) \right)^{-1} \mathbb{E}_{\substack{x_i \sim \mathbb{P}_S(X) \\ y_i|x_i \sim f(y|x;\beta^\star)}} \left[ R_{\beta^\star}(\hat{\beta}) \right] \\
&\geq \frac{1}{16} \mathsf{Tr} \left( \mathcal{I}_T(\beta_0) \mathcal{I}_S^{-1}(\beta_0) \right)^{-1} \inf_{\hat{\beta} \in \mathbb{B}_{\beta_0}(R_0)} \sup_{\beta^\star \in \mathbb{B}_{\beta_0}(R_1)} \mathbb{E}_{\substack{x_i \sim \mathbb{P}_S(X) \\ y_i|x_i \sim f(y|x;\beta^\star)}} \left[ (\hat{\beta} - \beta^\star)^T \mathcal{I}_T(\beta_0)(\hat{\beta} - \beta^\star) \right] \\
&\geq \frac{1}{16} \mathsf{Tr} \left( \mathcal{I}_T(\beta_0) \mathcal{I}_S^{-1}(\beta_0) \right)^{-1} \inf_{\hat{\beta} \in \mathbb{B}_{\beta_0}(R_0)} \sup_{\beta^\star \in C_{\beta_0}(\frac{R_1}{\sqrt{d}})} \mathbb{E}_{\substack{x_i \sim \mathbb{P}_S(X) \\ y_i|x_i \sim f(y|x;\beta^\star)}} \left[ (\hat{\beta} - \beta^\star)^T \mathcal{I}_T(\beta_0)(\hat{\beta} - \beta^\star) \right],
\end{aligned}
\tag{25}
$$

where the last inequality follows from the fact that $C_{\beta_0}(\frac{R_1}{\sqrt{d}}) \subseteq \mathbb{B}_{\beta_0}(R_1)$. By Lemma A.6, there exists a prior density $\lambda(\beta)$ supported on $C_{\beta_0}(\frac{R_1}{\sqrt{d}})$ such that for any estimator $\hat{\beta}$, we have

$$
\begin{aligned}
&\mathbb{E}_{\beta^\star \sim \lambda(\beta)} \mathbb{E}_{\substack{x_i \sim \mathbb{P}_S(X) \\ y_i|x_i \sim f(y|x;\beta^\star)}} \left[ (\hat{\beta} - \beta^\star)^T \mathcal{I}_T(\beta_0)(\hat{\beta} - \beta^\star) \right] \\
&\geq \frac{\mathsf{Tr} \left( \mathcal{I}_T(\beta_0) \mathcal{I}_S^{-1}(\beta_0) \right)^2}{n \mathbb{E}_{\beta^\star \sim \lambda(\beta)} \left[ \mathsf{Tr} \left( \mathcal{I}_S^{-1}(\beta_0) \mathcal{I}_S(\beta^\star) \mathcal{I}_S^{-1}(\beta_0) \mathcal{I}_T(\beta_0) \right) \right] + \frac{\pi^2 d}{R_1^2} \mathsf{Tr} \left( \mathcal{I}_T(\beta_0) \mathcal{I}_S^{-2}(\beta_0) \right)} \\
&\geq \frac{\mathsf{Tr} \left( \mathcal{I}_T(\beta_0) \mathcal{I}_S^{-1}(\beta_0) \right)^2}{2n \mathsf{Tr} \left( \mathcal{I}_T(\beta_0) \mathcal{I}_S^{-1}(\beta_0) \right) + \frac{\pi^2 d}{R_1^2} \mathsf{Tr} \left( \mathcal{I}_T(\beta_0) \mathcal{I}_S^{-2}(\beta_0) \right)}.
\end{aligned}
$$

Here the last inequality uses the fact that for any $\beta^\star \in C_{\beta_0}(\frac{R_1}{\sqrt{d}}) \subseteq \mathbb{B}_{\beta_0}(R_0)$, by Lemma A.5 (18), we have $\mathcal{I}_S^{-1}(\beta_0) \preceq 2 \mathcal{I}_S^{-1}(\beta^\star)$, which implies

$$
\begin{aligned}
\mathbb{E}_{\beta^\star \sim \lambda(\beta)} \left[ \mathsf{Tr} \left( \mathcal{I}_S^{-1}(\beta_0) \mathcal{I}_S(\beta^\star) \mathcal{I}_S^{-1}(\beta_0) \mathcal{I}_T(\beta_0) \right) \right] &\leq \mathbb{E}_{\beta^\star \sim \lambda(\beta)} \left[ \mathsf{Tr} \left( 2 \mathcal{I}_S^{-1}(\beta^\star) \mathcal{I}_S(\beta^\star) \mathcal{I}_S^{-1}(\beta_0) \mathcal{I}_T(\beta_0) \right) \right] \\
&= 2 \mathsf{Tr} \left( \mathcal{I}_T(\beta_0) \mathcal{I}_S^{-1}(\beta_0) \right).
\end{aligned}
$$

We then conclude for any estimator $\hat{\beta}$

$$
\begin{aligned}
&\sup_{\beta^\star \in C_{\beta_0}(\frac{R_1}{\sqrt{d}})} \mathbb{E}_{\substack{x_i \sim \mathbb{P}_S(X) \\ y_i|x_i \sim f(y|x;\beta^\star)}} \left[ (\hat{\beta} - \beta^\star)^T \mathcal{I}_T(\beta_0)(\hat{\beta} - \beta^\star) \right] \\
&\geq \mathbb{E}_{\beta^\star \sim \lambda(\beta)} \mathbb{E}_{\substack{x_i \sim \mathbb{P}_S(X) \\ y_i|x_i \sim f(y|x;\beta^\star)}} \left[ (\hat{\beta} - \beta^\star)^T \mathcal{I}_T(\beta_0)(\hat{\beta} - \beta^\star) \right] \\
&\geq \frac{\mathsf{Tr} \left( \mathcal{I}_T(\beta_0) \mathcal{I}_S^{-1}(\beta_0) \right)^2}{2n \mathsf{Tr} \left( \mathcal{I}_T(\beta_0) \mathcal{I}_S^{-1}(\beta_0) \right) + \frac{\pi^2 d}{R_1^2} \mathsf{Tr} \left( \mathcal{I}_T(\beta_0) \mathcal{I}_S^{-2}(\beta_0) \right)}.
\end{aligned}
\tag{26}
$$

Combine (25) and (26), we have

$$
\begin{aligned}
\inf_{\hat{\beta}} \sup_{\beta^{\star} \in \mathbb{B}_{\beta_0}(B)} &\mathsf{Tr}\left(\mathcal{I}_T(\beta^{\star})\mathcal{I}_S^{-1}(\beta^{\star})\right)^{-1} \mathbb{E}_{\substack{x_i \sim \mathbb{P}_S(X) \\ y_i | x_i \sim f(y|x;\beta^{\star})}} \left[R_{\beta^{\star}}(\hat{\beta})\right] \\
&\geq \frac{1}{16}\mathsf{Tr}\left(\mathcal{I}_T(\beta_0)\mathcal{I}_S^{-1}(\beta_0)\right)^{-1} \cdot \frac{\mathsf{Tr}\left(\mathcal{I}_T(\beta_0)\mathcal{I}_S^{-1}(\beta_0)\right)^2}{2n\mathsf{Tr}\left(\mathcal{I}_T(\beta_0)\mathcal{I}_S^{-1}(\beta_0)\right) + \frac{\pi^2 d}{R_1^2}\mathsf{Tr}\left(\mathcal{I}_T(\beta_0)\mathcal{I}_S^{-2}(\beta_0)\right)} \\
&= \frac{1}{16} \cdot \frac{1}{2n + \frac{\pi^2 d}{R_1^2}\mathsf{Tr}\left(\mathcal{I}_T(\beta_0)\mathcal{I}_S^{-2}(\beta_0)\right) \mathsf{Tr}\left(\mathcal{I}_T(\beta_0)\mathcal{I}_S^{-1}(\beta_0)\right)^{-1}}.
\end{aligned}
$$

Thus we prove Theorem A.4. $\qquad\square$

In the following, we prove Lemma A.5 and Lemma A.6.

**Proofs for Lemma A.5**

*Proof of Lemma A.5.* We choose

$$
R_0 := \min\left\{\frac{\lambda_{\min}^2(\mathcal{I}_S(\beta_0))}{4L_S\lambda_{\max}(\mathcal{I}_S(\beta_0))}, \frac{\lambda_{\min}(\mathcal{I}_T(\beta_0))}{4B_3 + 2L_T}, B\right\}, \quad R_1 := \frac{1}{4}\sqrt{\frac{\lambda_{\min}(\mathcal{I}_T(\beta_0))}{\lambda_{\max}(\mathcal{I}_T(\beta_0))}} \cdot R_0.
$$

In the sequel, we will show the aforementioned choices of $R_0$ and $R_1$ satisfy the conditions outlined in Lemma A.5.

First of all, we show (18) holds. Fix any $\beta \in \mathbb{B}_{\beta_0}(R_0)$. By Assumption B.2, we have

$$
\|\mathcal{I}_S(\beta) - \mathcal{I}_S(\beta_0)\|_2 \leq L_S\|\beta - \beta_0\|_2 \leq L_S R_0,
$$

which implies

$$
\|\mathcal{I}_S^{-1}(\beta) - \mathcal{I}_S^{-1}(\beta_0)\|_2 \leq \|\mathcal{I}_S^{-1}(\beta_0)\|_2 \cdot \|\mathcal{I}_S(\beta) - \mathcal{I}_S(\beta_0)\|_2 \cdot \|\mathcal{I}_S^{-1}(\beta)\|_2 \leq \frac{L_S R_0}{\lambda_{\min}(\mathcal{I}_S(\beta_0))\lambda_{\min}(\mathcal{I}_S(\beta))}.
$$

By Weyl's inequality (Lemma 2.2 in Chen et al. (2021)), we have

$$
|\lambda_{\min}(\mathcal{I}_S(\beta)) - \lambda_{\min}(\mathcal{I}_S(\beta_0))| \leq \|\mathcal{I}_S(\beta) - \mathcal{I}_S(\beta_0)\|_2 \leq L_S R_0.
$$

Note that

$$
R_0 \leq \frac{\lambda_{\min}^2(\mathcal{I}_S(\beta_0))}{4L_S\lambda_{\max}(\mathcal{I}_S(\beta_0))} \leq \frac{\lambda_{\min}(\mathcal{I}_S(\beta_0))}{2L_S}.
$$

Thus we have

$$
\lambda_{\min}(\mathcal{I}_S(\beta)) \geq \lambda_{\min}(\mathcal{I}_S(\beta_0)) - L_S R_0 \geq \frac{1}{2}\lambda_{\min}(\mathcal{I}_S(\beta_0)),
$$

which implies

$$
\|\mathcal{I}_S^{-1}(\beta) - \mathcal{I}_S^{-1}(\beta_0)\|_2 \leq \frac{L_S R_0}{\lambda_{\min}(\mathcal{I}_S(\beta_0))\lambda_{\min}(\mathcal{I}_S(\beta))} \leq \frac{2L_S R_0}{\lambda_{\min}^2(\mathcal{I}_S(\beta_0))} \leq \frac{1}{2\lambda_{\max}(\mathcal{I}_S(\beta_0))}. \tag{27}
$$

Then for any $x \in \mathbb{R}^d$, we have

$$
\begin{aligned}
x^T\left(\mathcal{I}_S^{-1}(\beta) - \frac{1}{2}\mathcal{I}_S^{-1}(\beta_0)\right)x &= \frac{1}{2}x^T\mathcal{I}_S^{-1}(\beta_0)x + x^T\left(\mathcal{I}_S^{-1}(\beta) - \mathcal{I}_S^{-1}(\beta_0)\right)x \\
&\geq \frac{\|x\|_2^2}{2\lambda_{\max}(\mathcal{I}_S(\beta_0))} - \|x\|_2^2 \cdot \|\mathcal{I}_S^{-1}(\beta) - \mathcal{I}_S^{-1}(\beta_0)\|_2 \\
&= \|x\|_2^2\left(\frac{1}{2\lambda_{\max}(\mathcal{I}_S(\beta_0))} - \|\mathcal{I}_S^{-1}(\beta) - \mathcal{I}_S^{-1}(\beta_0)\|_2\right) \\
&\geq 0,
\end{aligned}
$$

where the last inequality follows from (27). Thus we conclude $\mathcal{I}_S^{-1}(\beta) \succeq \frac{1}{2}\mathcal{I}_S^{-1}(\beta_0)$. Similarly, we can show that $\mathcal{I}_S^{-1}(\beta) \preceq 2\mathcal{I}_S^{-1}(\beta_0)$. As a result, we show that (18) holds.

Next, we show (19) holds. Fix any $\beta^\star, \beta \in \mathbb{B}_{\beta_0}(R_0)$. By Assumption A.2, for any $x \in \mathcal{X}, y \in \mathcal{Y}$, we have

$$\|\nabla^2 \ell(x, y, \beta) - \nabla^2 \ell(x, y, \beta^\star)\|_2 \leq B_3 \|\beta - \beta^\star\|_2 \leq 2B_3 R_0,$$

which implies

$$
\begin{aligned}
& \left\| \mathbb{E}_{\substack{x \sim \mathbb{P}_T(X) \\ y|x \sim f(y|x;\beta^\star)}} [\nabla^2 \ell(x, y, \beta)] - \mathbb{E}_{\substack{x \sim \mathbb{P}_T(X) \\ y|x \sim f(y|x;\beta^\star)}} [\nabla^2 \ell(x, y, \beta^\star)] \right\|_2 \\
& \leq \mathbb{E}_{\substack{x \sim \mathbb{P}_T(X) \\ y|x \sim f(y|x;\beta^\star)}} [\|\nabla^2 \ell(x, y, \beta) - \nabla^2 \ell(x, y, \beta^\star)\|_2] \leq 2B_3 R_0.
\end{aligned}
\tag{28}
$$

By Assumption B.2, we have

$$\|\mathcal{I}_T(\beta^\star) - \mathcal{I}_T(\beta_0)\|_2 \leq L_T \|\beta^\star - \beta_0\|_2 \leq L_T R_0 \tag{29}$$

Thus, by (28) and (29), we have

$$
\begin{aligned}
& \left\| \mathbb{E}_{\substack{x \sim \mathbb{P}_T(X) \\ y|x \sim f(y|x;\beta^\star)}} [\nabla^2 \ell(x, y, \beta)] - \mathcal{I}_T(\beta_0) \right\|_2 \\
& \leq \left\| \mathbb{E}_{\substack{x \sim \mathbb{P}_T(X) \\ y|x \sim f(y|x;\beta^\star)}} [\nabla^2 \ell(x, y, \beta)] - \mathbb{E}_{\substack{x \sim \mathbb{P}_T(X) \\ y|x \sim f(y|x;\beta^\star)}} [\nabla^2 \ell(x, y, \beta^\star)] \right\|_2 + \|\mathcal{I}_T(\beta^\star) - \mathcal{I}_T(\beta_0)\|_2 \\
& \leq (2B_3 + L_T) R_0 \\
& \leq \frac{1}{2} \lambda_{\min}(\mathcal{I}_T(\beta_0)),
\end{aligned}
$$

where the last inequality follows from the choice of $R_0$. Consequently, for any $x \in \mathbb{R}^d$, we have

$$
\begin{aligned}
& x^T \left( \mathbb{E}_{\substack{x \sim \mathbb{P}_T(X) \\ y|x \sim f(y|x;\beta^\star)}} [\nabla^2 \ell(x, y, \beta)] - \frac{1}{2}\mathcal{I}_T(\beta_0) \right) x \\
& = \frac{1}{2} x^T \mathcal{I}_T(\beta_0) x + x^T \left( \mathbb{E}_{\substack{x \sim \mathbb{P}_T(X) \\ y|x \sim f(y|x;\beta^\star)}} [\nabla^2 \ell(x, y, \beta)] - \mathcal{I}_T(\beta_0) \right) x \\
& \geq \frac{1}{2} \|x\|_2^2 \lambda_{\min}(\mathcal{I}_T(\beta_0)) - \|x\|_2^2 \left\| \mathbb{E}_{\substack{x \sim \mathbb{P}_T(X) \\ y|x \sim f(y|x;\beta^\star)}} [\nabla^2 \ell(x, y, \beta)] - \mathcal{I}_T(\beta_0) \right\|_2 \\
& \geq \frac{1}{2} \|x\|_2^2 \lambda_{\min}(\mathcal{I}_T(\beta_0)) - \frac{1}{2} \|x\|_2^2 \lambda_{\min}(\mathcal{I}_T(\beta_0)) = 0.
\end{aligned}
$$

We then conclude $\mathbb{E}_{\substack{x \sim \mathbb{P}_T(X) \\ y|x \sim f(y|x;\beta^\star)}} [\nabla^2 \ell(x, y, \beta)] \succeq \frac{1}{2}\mathcal{I}_T(\beta_0)$. Similarly, we can show that $\mathbb{E}_{\substack{x \sim \mathbb{P}_T(X) \\ y|x \sim f(y|x;\beta^\star)}} [\nabla^2 \ell(x, y, \beta)] \preceq 2\mathcal{I}_T(\beta_0)$. Thus we show that (19) holds.

Finally, we need to show that for any $\beta^\star \in \mathbb{B}_{\beta_0}(R_1), \beta \notin \mathbb{B}_{\beta_0}(R_0)$: $R_{\beta^\star}(\beta) \geq R_{\beta^\star}(\beta_0)$. Fix any $\beta^\star \in \mathbb{B}_{\beta_0}(R_1), \beta \notin \mathbb{B}_{\beta_0}(R_0)$. We denote

$$\beta' := \{\lambda \beta + (1 - \lambda)\beta^\star \mid \lambda \in [0, 1]\} \cap \{\beta' \mid \|\beta' - \beta_0\|_2 = R_0\}.$$

By the choice of $R_1$, we know that $R_1 \leq R_0/2$, which implies

$$\|\beta' - \beta^\star\|_2 \geq \|\beta' - \beta_0\|_2 - \|\beta_0 - \beta^\star\|_2 \geq R_0 - R_1 \geq \frac{R_0}{2}. \tag{30}$$

By convexity of $R_{\beta^\star}(\cdot)$ assumed in Assumption B.3 and $R_{\beta^\star}(\beta) \geq R_{\beta^\star}(\beta^\star)$, we have $R_{\beta^\star}(\beta) \geq R_{\beta^\star}(\beta')$. Thus, we obtain

$$
\begin{aligned}
R_{\beta^\star}(\beta) - R_{\beta^\star}(\beta^\star) &\geq R_{\beta^\star}(\beta') - R_{\beta^\star}(\beta^\star) \\
&\overset{\text{Taylor}}{=} \frac{1}{2}(\beta' - \beta^\star)^T \mathbb{E}_{\substack{x \sim \mathbb{P}_T(X) \\ y|x \sim f(y|x;\beta^\star)}} [\nabla^2 \ell(x, y, \tilde{\beta})](\beta' - \beta^\star) \\
&\overset{\text{by (19)}}{\geq} \frac{1}{4}(\beta' - \beta^\star)^T \mathcal{I}_T(\beta_0)(\beta' - \beta^\star) \\
&\geq \frac{1}{4}\lambda_{\min}(\mathcal{I}_T(\beta_0))\|\beta' - \beta^\star\|_2^2 \\
&\overset{\text{by (30)}}{\geq} \frac{R_0^2}{16}\lambda_{\min}(\mathcal{I}_T(\beta_0)).
\end{aligned}
\tag{31}
$$

Note that

$$
\begin{aligned}
R_{\beta^\star}(\beta_0) - R_{\beta^\star}(\beta^\star) &\overset{\text{Taylor}}{=} \frac{1}{2}(\beta_0 - \beta^\star)^T \mathbb{E}_{\substack{x \sim \mathbb{P}_T(X) \\ y|x \sim f(y|x;\beta^\star)}} [\nabla^2 \ell(x, y, \tilde{\beta})](\beta_0 - \beta^\star) \\
&\overset{\text{by (19)}}{\leq} (\beta_0 - \beta^\star)^T \mathcal{I}_T(\beta_0)(\beta_0 - \beta^\star) \\
&\leq \lambda_{\max}(\mathcal{I}_T(\beta_0))\|\beta_0 - \beta^\star\|_2^2 \\
&\leq R_1^2 \lambda_{\max}(\mathcal{I}_T(\beta_0)) \\
&= \frac{R_0^2}{16}\lambda_{\min}(\mathcal{I}_T(\beta_0)),
\end{aligned}
\tag{32}
$$

where the last equation follows from the choice of $R_1$. By (31) and (32), we obtain $R_{\beta^\star}(\beta) \geq R_{\beta^\star}(\beta_0)$. Thus, we finish the proof of Lemma A.5. □

**Proofs for Lemma A.6**

*Proof of Lemma A.6.* Let $\beta_0 = [\beta_{0,1}, \ldots, \beta_{0,d}]^T$, $\beta = [\beta_1, \ldots, \beta_d]^T$ and

$$
f_i(x) := \frac{\pi}{4B} \cos\left(\frac{\pi}{2B}(x - \beta_{0,i})\right), \ i = 1, \ldots, d.
$$

We define the prior density as

$$
\lambda(\beta) := \begin{cases} \Pi_{i=1}^d f_i(\beta_i) & \beta \in C_{\beta_0}(B) \\ 0 & \beta \notin C_{\beta_0}(B) \end{cases},
$$

which is supported on $C_{\beta_0}(B)$. In the sequel, we will show this prior density satisfies the condition outlined in Lemma A.6.

For notation simplicity, we denote

$$
A = (A_{ij}) := \mathcal{I}_T^{-1}(\beta_0), \ C = (C_{ij}) := \mathcal{I}_T(\beta_0)\mathcal{I}_S^{-1}(\beta_0).
$$

By multivariate van Trees inequality (Theorem 1 in Gill & Levit (1995)), for any estimator $\hat{\beta}$, we have

$$
\begin{aligned}
&\mathbb{E}_{\beta^\star \sim \lambda(\beta)} \mathbb{E}_{\substack{x_i \sim \mathbb{P}_S(X) \\ y_i|x_i \sim f(y|x;\beta^\star)}} \left[(\hat{\beta} - \beta^\star)^T \mathcal{I}_T(\beta_0)(\hat{\beta} - \beta^\star)\right] \\
&\geq \frac{\text{Tr}\left(\mathcal{I}_T(\beta_0)\mathcal{I}_S^{-1}(\beta_0)\right)^2}{n\mathbb{E}_{\beta^\star \sim \lambda(\beta)}\left[\text{Tr}\left(\mathcal{I}_S^{-1}(\beta_0)\mathcal{I}_S(\beta^\star)\mathcal{I}_S^{-1}(\beta_0)\mathcal{I}_T(\beta_0)\right)\right] + \tilde{\mathcal{I}}(\lambda)},
\end{aligned}
\tag{33}
$$

where

$$
\tilde{\mathcal{I}}(\lambda) = \int_{C_{\beta_0}(B)} \left(\sum_{i,j,k,\ell} A_{ij}C_{ik}C_{j\ell}\frac{\partial}{\partial \beta_k}\lambda(\beta)\frac{\partial}{\partial \beta_\ell}\lambda(\beta)\right)\frac{1}{\lambda(\beta)}d\beta.
$$

By the choice of $\lambda(\beta)$, we have

$$\int_{C_{\beta_0}(B)} \left( \sum_{\substack{i,j,k,\ell \\ k \neq \ell}} A_{ij} C_{ik} C_{j\ell} \frac{\partial}{\partial \beta_k} \lambda(\beta) \frac{\partial}{\partial \beta_\ell} \lambda(\beta) \right) \frac{1}{\lambda(\beta)} d\beta$$

$$= \int_{C_{\beta_0}(B)} \sum_{\substack{i,j,k,\ell \\ k \neq \ell}} A_{ij} C_{ik} C_{j\ell} f_k'(\beta_k) f_\ell'(\beta_\ell) \Pi_{i \neq k, \ell} f_i(\beta_i) d\beta$$

$$= \sum_{\substack{i,j,k,\ell \\ k \neq \ell}} A_{ij} C_{ik} C_{j\ell} \int_{C_{\beta_0}(B)} f_k'(\beta_k) f_\ell'(\beta_\ell) \Pi_{i \neq k, \ell} f_i(\beta_i) d\beta$$

$$= 0.$$

Here the last equation follows from the fact

$$\int_{\beta_{0,k}-B}^{\beta_{0,k}+B} f_k'(\beta_k) d\beta_k = \int_{\beta_{0,\ell}-B}^{\beta_{0,\ell}+B} f_\ell'(\beta_\ell) d\beta_\ell = 0.$$

Note that

$$\int_{C_{\beta_0}(B)} \left( \sum_{\substack{i,j,k,\ell \\ k = \ell}} A_{ij} C_{ik} C_{j\ell} \frac{\partial}{\partial \beta_k} \lambda(\beta) \frac{\partial}{\partial \beta_\ell} \lambda(\beta) \right) \frac{1}{\lambda(\beta)} d\beta$$

$$= \sum_{i,j,k} A_{ij} C_{ik} C_{jk} \int_{C_{\beta_0}(B)} \frac{(f_k'(\beta_k))^2}{f_k(\beta_k)} \Pi_{i \neq k} f_i(\beta_i) d\beta$$

$$= \sum_{i,j,k} A_{ij} C_{ik} C_{jk} \int_{\beta_{0,k}-B}^{\beta_{0,k}+B} \frac{(f_k'(\beta_k))^2}{f_k(\beta_k)} d\beta_k$$

$$= \frac{\pi^2}{B^2} \sum_{i,j,k} A_{ij} C_{ik} C_{jk}$$

$$= \frac{\pi^2}{B^2} \mathsf{Tr}(ACC^T).$$

Thus, we have

$$\tilde{\mathcal{I}}(\lambda) = \int_{C_{\beta_0}(B)} \left( \sum_{\substack{i,j,k,\ell \\ k \neq \ell}} A_{ij} C_{ik} C_{j\ell} \frac{\partial}{\partial \beta_k} \lambda(\beta) \frac{\partial}{\partial \beta_\ell} \lambda(\beta) \right) \frac{1}{\lambda(\beta)} d\beta$$

$$+ \int_{C_{\beta_0}(B)} \left( \sum_{\substack{i,j,k,\ell \\ k = \ell}} A_{ij} C_{ik} C_{j\ell} \frac{\partial}{\partial \beta_k} \lambda(\beta) \frac{\partial}{\partial \beta_\ell} \lambda(\beta) \right) \frac{1}{\lambda(\beta)} d\beta$$

$$= \frac{\pi^2}{B^2} \mathsf{Tr}(ACC^T)$$

$$= \frac{\pi^2}{B^2} \mathsf{Tr}\left( \mathcal{I}_T(\beta_0) \mathcal{I}_S^{-2}(\beta_0) \right). \tag{34}$$

Combine (33) and (34), we prove Lemma A.6.

$$\square$$

# B  PROOFS FOR SECTION 4

## B.1  PROOFS FOR PROPOSITION 4.1 AND THEOREM 4.2

*Proof.* For our linear regression model,

$$\ell(x, y, \beta) = \frac{1}{2}\log(2\pi) + \frac{1}{2}(y - x^T\beta)^2.$$

The convexity of $\ell$ in $\beta$ immediately implies Assumption B.3. We then have

$$\nabla\ell(x, y, \beta) = -x(y - x^T\beta),$$
$$\nabla^2\ell(x, y, \beta) = xx^T,$$
$$\nabla^3\ell(x, y, \beta) = 0,$$
$$\mathcal{I}_S = \mathbb{E}_{x\sim\mathbb{P}_S(X)}[xx^T] = I_d,$$
$$\mathcal{I}_T = \mathbb{E}_{x\sim\mathbb{P}_T(X)}[xx^T] = \alpha\alpha^T + \sigma^2 I_d.$$

Therefore Assumption B.2 is satisfied with $L_S = L_T = 0$ and Assumption B.4 trivially holds. Note that $\nabla\ell(x_i, y_i, \beta^\star) = -x_i\varepsilon_i$. Since $\|x_i\|_2$ is $\sqrt{d}$-subgaussian and $|\varepsilon_i|$ is 1-subgaussian, by Lemma 2.7.7 in Vershynin (2018), it holds that $\|x_i\|_2|\varepsilon_i|$ is $\sqrt{d}$-subexponential random variable. Thus $\|A\nabla\ell(x_i, y_i, \beta^\star)\|_2$ is $\|A\|_2\sqrt{d}$-subexponential random variable.

Then, by Lemma D.1 with $u_i = A(\nabla\ell(x_i, y_i, \beta^\star) - \mathbb{E}[\nabla\ell(x_i, y_i, \beta^\star)]) = A\nabla\ell(x_i, y_i, \beta^\star)$, $V = \mathbb{E}[\|u_i\|_2^2] = n\cdot\mathbb{E}\|A(\nabla\ell_n(\beta^\star) - \mathbb{E}[\nabla\ell_n(\beta^\star)])\|_2^2$, $\alpha = 1$ and $B_u^{(\alpha)} = c\sqrt{d}\|A\|_2$, we have for any matrix $A \in \mathbb{R}^{d\times d}$, and any $\delta \in (0, 1)$, with probability at least $1 - \delta$:

$$\|A(\nabla\ell_n(\beta^\star) - \mathbb{E}[\nabla\ell_n(\beta^\star)])\|_2 \le c\left(\sqrt{\frac{V\log\frac{d}{\delta}}{n}} + \sqrt{d}\|A\|_2\log(\frac{\sqrt{d}\|A\|_2}{\sqrt{V}})\frac{\log\frac{d}{\delta}}{n}\right),$$

which satisfies the gradient concentration in Assumption A.1 with $B_1 = c\sqrt{d}$ and $\gamma = 1$.

Note that $x_i \sim \mathcal{N}(0, I_d)$. Thus, by Theorem 13.3 in Rinaldo (2018), for any $\delta \in (0, 1)$, with probability at least $1 - \delta$, we have

$$\|\nabla^2\ell_n(\beta^\star) - \mathbb{E}[\nabla^2\ell_n(\beta^\star)]\|_2 = \left\|\frac{1}{n}\sum_{i=1}^n x_i x_i^T - I_d\right\|_2$$
$$\le c\left(\sqrt{\frac{d\log(1/\delta)}{n}} + \frac{d\log(1/\delta)}{n}\right)$$
$$\le 2c\sqrt{\frac{d\log(1/\delta)}{n}},$$

where the last inequality holds if $n \ge \mathcal{O}(d\log\frac{1}{\delta})$. Hence linear regression model satisfies the matrix concentration in Assumption A.1 with $B_2 = c\sqrt{d}$, $N(\delta) = d\log\frac{1}{\delta}$. Since $\nabla^3\ell \equiv 0$, we know Assumption A.2 holds with $B_3 = 0$.

Note that

$$\nabla^2\ell_n(\beta) = \frac{1}{n}\sum_{i=1}^n x_i x_i^T = \frac{1}{n}X^T X,$$

where $X := [x_1, \ldots, x_n]^T$. Given that $\{x_i\}_{i=1}^n$ are i.i.d $\mathcal{N}(0, I_d)$, it follows that $X$ is almost surely full rank when $n \ge d$. Hence, when $n \ge d$, we have

$$\nabla^2\ell_n(\beta) = \frac{1}{n}\sum_{i=1}^n x_i x_i^T = \frac{1}{n}X^T X \succ 0.$$

Consequently, $\ell_n(\cdot)$ is strictly convex and thus satisfies Assumption A.3. Finally, Theorem 4.2 follows directly from Theorem 3.1 with $\gamma = 1$, $B_1 = c\sqrt{d}$, $B_2 = c\sqrt{d}$, $B_3 = 0$, $N(\delta) = d\log\frac{1}{\delta}$, $\mathcal{I}_S = I_d$ and $\mathcal{I}_T = \alpha\alpha^T + \sigma^2 I_d$.

$\square$

### B.2 PROOFS FOR PROPOSITION 4.3 AND THEOREM 4.4

*Proof.* In the following, we will show the logistic regression model satisfies Assumptions A and B. For logistic regression, the loss function is defined as

$$\ell(x, y, \beta) = \log(1 + e^{x^T\beta}) - y(x^T\beta).$$

We then have

$$\nabla\ell(x, y, \beta) = \frac{x}{1 + e^{-x^T\beta}} - xy,$$

$$\nabla^2\ell(x, y, \beta) = \frac{xx^T}{2 + e^{-x^T\beta} + e^{x^T\beta}},$$

$$\nabla^3\ell(x, y, \beta) = \frac{e^{-x^T\beta} - e^{x^T\beta}}{(2 + e^{-x^T\beta} + e^{x^T\beta})^2} \cdot x \otimes x \otimes x.$$

Here $\otimes$ represents the tensor product and $x \otimes x \otimes x \in \mathbb{R}^{d \times d \times d}$ with $(x \otimes x \otimes x)_{ijk} = x_i x_j x_k$. The convexity of $\ell$ in $\beta$ immediately implies Assumption B.3; Assumption B.4 trivially holds. Note that on source domain $\|x\|_2 = \sqrt{d}$ and $|y| \le 1$. Hence we have for any $(x, y)$ on source domain:

$$\|\nabla\ell(x, y, \beta^\star)\|_2 = \left\|\frac{x}{1 + e^{-x^T\beta^\star}} - xy\right\|_2 \le \left\|\frac{x}{1 + e^{-x^T\beta^\star}}\right\|_2 + \|xy\|_2 \le \|x\|_2 + \|x\|_2 = 2\sqrt{d},$$

$$\|\nabla^2\ell(x, y, \beta^\star)\|_2 = \left\|\frac{xx^T}{2 + e^{-x^T\beta^\star} + e^{x^T\beta^\star}}\right\|_2 \le \|xx^T\|_2 \le \|x\|_2^2 \le d.$$

By Lemma D.1 with $u_i = A(\nabla\ell(x_i, y_i, \beta^\star) - \mathbb{E}[\nabla\ell(x_i, y_i, \beta^\star)]) = A\nabla\ell(x_i, y_i, \beta^\star), V = \mathbb{E}[\|u_i\|_2^2], \alpha = +\infty, B_u^{(\alpha)} = 2\sqrt{d}\|A\|_2$, we have for any matrix $A \in \mathbb{R}^{d \times d}$, and any $\delta \in (0, 1)$, with probability at least $1 - \delta$:

$$\|A(\nabla\ell_n(\beta^\star) - \mathbb{E}[\nabla\ell_n(\beta^\star)])\|_2 \le c\left(\sqrt{\frac{V\log\frac{d}{\delta}}{n}} + \frac{\sqrt{d}\|A\|_2\log\frac{d}{\delta}}{n}\right),$$

which satisfies the gradient concentration in Assumption A.1 with $B_1 = c\sqrt{d}$ and $\gamma = 0$. By matrix Hoeffding inequality, logistic regression model satisfies the matrix concentration in Assumption A.1 with $B_2 = cd$. We conclude that logistic regression model satisfies Assumption A.1 with $N(\delta) = 0$, $B_1 = c\sqrt{d}, \gamma = 0, B_2 = cd$.

Note that for $x$ on source domain, we have $\|x\|_2 \le \sqrt{d}$; for $x$ on target domain, we have $\|x\|_2 \le \sqrt{d} + r$. Thus, it holds that

$$\|\nabla^3\ell(x, y, \beta)\|_2 = \left\|\frac{e^{-x^T\beta} - e^{x^T\beta}}{(2 + e^{-x^T\beta} + e^{x^T\beta})^2} \cdot x \otimes x \otimes x\right\|_2 \le_{(i)} \|x \otimes x \otimes x\|_2 \le \|x\|_2^3 \le (\sqrt{d} + r)^3.$$

Here $(i)$ uses the fact that

$$\left|\frac{e^{-x^T\beta} - e^{x^T\beta}}{(2 + e^{-x^T\beta} + e^{x^T\beta})^2}\right| \le \frac{e^{-x^T\beta} + e^{x^T\beta}}{(2 + e^{-x^T\beta} + e^{x^T\beta})^2} \le \frac{1}{2 + e^{-x^T\beta} + e^{x^T\beta}} \le 1.$$

Hence logistic regression satisfies Assumptions A.2 with $B_3 = (\sqrt{d} + r)^3$. Notice that this also implies Assumption B.2: By definition,

$$\mathcal{I}_S(\beta) := \mathbb{E}_{x \sim \mathbb{P}_S(X)}[\nabla^2\ell(x, y, \beta)],$$

therefore

$$\|\mathcal{I}_S(\beta_1) - \mathcal{I}_S(\beta_2)\| = \|\mathbb{E}_{x \sim \mathbb{P}_S(X)}[\nabla^2\ell(x, y, \beta_1) - \nabla^2\ell(x, y, \beta_2)]\|$$
$$\le \mathbb{E}_{x \sim \mathbb{P}_S(X)}[\|\nabla^2\ell(x, y, \beta_1) - \nabla^2\ell(x, y, \beta_2)\|]$$
$$\le (\sqrt{d})^3\|\beta_1 - \beta_2\|.$$

Similarly

$$\|\mathcal{I}_T(\beta_1) - \mathcal{I}_T(\beta_2)\| \leq (\sqrt{d} + r)^3 \|\beta_1 - \beta_2\|.$$

These inequities shows that logistic regression model satisfies Assumption B.2 with $L_S = d^{1.5}$ and $L_T = (\sqrt{d} + r)^3$. Note that

$$\nabla^2 \ell_n(\beta) = \frac{1}{n} \sum_{i=1}^n \nabla^2 \ell(x_i, y_i, \beta) = \frac{1}{n} \sum_{i=1}^n \frac{x_i x_i^T}{2 + e^{-x_i^T \beta} + e^{x_i^T \beta}} = \frac{1}{n} X^T A X,$$

where $X := [x_1, \ldots, x_n]^T \in \mathbb{R}^{n \times d}$ and $A := \mathsf{diag}(1/(2 + e^{-x_i^T \beta} + e^{x_i^T \beta})) \succ 0$. When $n \geq d$, $X$ is full rank (i.e., $\mathsf{rank}(X) = d$) almost surely, consequently, $\ell_n(\cdot)$ is strictly convex and thus satisfies Assumption A.3.

By Theorem 3.1, we have when $n \geq \mathcal{O}(N^\star \log \frac{d}{\delta})$,

$$R_{\beta^\star}(\beta_{\mathsf{MLE}}) \lesssim \frac{\mathsf{Tr}\left(\mathcal{I}_T \mathcal{I}_S^{-1}\right) \log \frac{d}{\delta}}{n}.$$

Here

$$N^\star := (1 + \tilde{\kappa}/\kappa)^2 \cdot \max\left\{ \tilde{\kappa}^{-1} \alpha_1^2 \log^{2\gamma}\left((1 + \tilde{\kappa}/\kappa)\tilde{\kappa}^{-1}\alpha_1^2\right), \, \alpha_2^2, \, \tilde{\kappa}(1 + \|\mathcal{I}_T^{\frac{1}{2}} \mathcal{I}_S^{-1} \mathcal{I}_T^{\frac{1}{2}}\|_2^{-2})\alpha_3^2 \right\},$$

where $\alpha_1 := B_1 \|\mathcal{I}_S^{-1}\|_2^{0.5}$, $\alpha_2 := B_2 \|\mathcal{I}_S^{-1}\|_2$, $\alpha_3 := B_3 \|\mathcal{I}_S^{-1}\|_2^{1.5}$,

$$\kappa := \frac{\mathsf{Tr}(\mathcal{I}_T \mathcal{I}_S^{-1})}{\|\mathcal{I}_T^{\frac{1}{2}} \mathcal{I}_S^{-1} \mathcal{I}_T^{\frac{1}{2}}\|_2}, \quad \tilde{\kappa} := \frac{\mathsf{Tr}(\mathcal{I}_S^{-1})}{\|\mathcal{I}_S^{-1}\|_2}.$$

Now it remains to calculate the quantities $N^\star$ and $\mathsf{Tr}\left(\mathcal{I}_T \mathcal{I}_S^{-1}\right)$ for this instance, where the crucial part is to identify what are $\mathcal{I}_S$ and $\mathcal{I}_T$. The following two lemmas give the characterization of $\mathcal{I}_S$ and $\mathcal{I}_T$.

**Lemma B.1.** *Under the conditions of Theorem 4.4, we have $\mathcal{I}_S = U\mathsf{diag}(\lambda_1, \lambda_2, \ldots, \lambda_2)U^T$ and $\mathcal{I}_T = U\mathsf{diag}(\lambda_1, \lambda_2 + r^2\lambda_3, \lambda_2, \ldots, \lambda_2)U^T$ for an orthonormal matrix $U$. Where*

$$\lambda_1 := \mathbb{E}_{x \sim \mathsf{Uniform}(\mathcal{S}^{d-1}(\sqrt{d}))}\left[\frac{(\beta^{\star T} x)^2}{2 + \exp(\beta^{\star T} x) + \exp(-\beta^{\star T} x)}\right],$$

$$\lambda_2 := \mathbb{E}_{x \sim \mathsf{Uniform}(\mathcal{S}^{d-1}(\sqrt{d}))}\left[\frac{(\beta_\perp^{\star T} x)^2}{2 + \exp(\beta^{\star T} x) + \exp(-\beta^{\star T} x)}\right],$$

$$\lambda_3 := \mathbb{E}_{x \sim \mathsf{Uniform}(\mathcal{S}^{d-1}(\sqrt{d}))}\left[\frac{1}{2 + \exp(\beta^{\star T} x) + \exp(-\beta^{\star T} x)}\right].$$

**Lemma B.2.** *Under the conditions of Theorem 4.4, there exist absolute constants $c, C, c' > 0$ such that $c < \lambda_1, \lambda_2, \lambda_3 < C$, for $d \geq c'$.*

The proofs for these two lemmas are in the next section. With Lemma B.1, we have $\mathcal{I}_T \mathcal{I}_S^{-1} = U\mathsf{diag}(1, 1 + r^2 \frac{\lambda_3}{\lambda_2}, \ldots, 1)U^T$, $\mathcal{I}_S^{-1} = U\mathsf{diag}(\frac{1}{\lambda_1}, \frac{1}{\lambda_2}, \ldots, \frac{1}{\lambda_2})U^T$. By Lemma B.2, since $\lambda_1, \lambda_2, \lambda_3 = O(1)$, we have $\mathsf{Tr}(\mathcal{I}_T \mathcal{I}_S^{-1}) = d + r^2 \frac{\lambda_3}{\lambda_2} \asymp d + r^2$, $\|\mathcal{I}_T \mathcal{I}_S^{-1}\|_2 = 1 + r^2 \frac{\lambda_3}{\lambda_2} \asymp 1 + r^2$. Similarly $\mathsf{Tr}(\mathcal{I}_S^{-1}) = \lambda_1^{-1} + (d-1)\lambda_2^{-1} \asymp d$, $\|\mathcal{I}_S^{-1}\|_2 = \max\{\lambda_1^{-1}, \lambda_2^{-1}\} \asymp 1$. Also recall that $B_1 = \sqrt{d}, B_2 = d, B_3 = (\sqrt{d} + r)^3$, plug in all those quantities we have $\kappa = \frac{\mathsf{Tr}(\mathcal{I}_T \mathcal{I}_S^{-1})}{\|\mathcal{I}_T \mathcal{I}_S^{-1}\|_2} \asymp \frac{d + r^2}{1 + r^2}$, $\tilde{\kappa} = \frac{\mathsf{Tr}(\mathcal{I}_S^{-1})}{\|\mathcal{I}_S^{-1}\|_2} \asymp d, \alpha_1 = B_1 \|\mathcal{I}_S^{-1}\|_2^{0.5} \asymp \sqrt{d}, \alpha_2 = B_2 \|\mathcal{I}_S^{-1}\|_2 \asymp d, \alpha_3 = B_3 \|\mathcal{I}_S^{-1}\|_2^{1.5} \asymp (\sqrt{d} + r)^3$. Therefore we have when $n \geq \mathcal{O}(N^\star \log \frac{d}{\delta})$,

$$R_{\beta^\star}(\beta_{\mathsf{MLE}}) \lesssim \frac{\mathsf{Tr}\left(\mathcal{I}_T \mathcal{I}_S^{-1}\right) \log \frac{d}{\delta}}{n} \asymp \frac{(d + r^2) \log \frac{d}{\delta}}{n},$$

where

$$N^\star = (1 + \tilde{\kappa}/\kappa)^2 \cdot \max\left\{\tilde{\kappa}^{-1}\alpha_1^2 \log^{2\gamma}\left((1 + \tilde{\kappa}/\kappa)\tilde{\kappa}^{-1}\alpha_1^2\right),\ \alpha_2^2,\ \tilde{\kappa}(1 + \|\mathcal{I}_T^{\frac{1}{2}}\mathcal{I}_S^{-1}\mathcal{I}_T^{\frac{1}{2}}\|_2^{-2})\alpha_3^2\right\}$$

$$\asymp \left(1 + \frac{d + r^2 d}{d + r^2}\right)^2 \cdot \max\left\{1, d^2, d(1 + (1 + r^2)^{-2})(\sqrt{d} + r)^6\right\}$$

$$= \left(1 + \frac{d + r^2 d}{d + r^2}\right)^2 \cdot d(\sqrt{d} + r)^6.$$

When $r \lesssim 1$, $N^\star \asymp d^4$. When $1 \lesssim r \lesssim \sqrt{d}$, $N^\star \asymp r^4 d^4$. When $\sqrt{d} \lesssim r$, $N^\star \asymp r^6 d^3$. $\qquad\square$

### B.2.1 PROOFS FOR LEMMA B.1 AND B.2

The intuition of proving Lemma B.1 and B.2 is that, when $d$ is large, distribution $\mathsf{Uniform}(\mathcal{S}^{d-1}(\sqrt{d}))$ behaves similar to distribution $\mathcal{N}(0, I_d)$ which has good properties (isotropic, independence of each entry, etc.)

*Proof of Lemma B.1.* By definition,

$$\mathcal{I}_S := \mathbb{E}_{x \sim \mathsf{Uniform}(\mathcal{S}^{d-1}(\sqrt{d}))}\left[\frac{xx^T}{2 + \exp(\beta^{\star T}x) + \exp(-\beta^{\star T}x)}\right]$$

Let $z \sim \mathcal{N}(0, I_d)$, then $x$ and $z\frac{\sqrt{d}}{\|z\|_2}$ have the same distribution. Therefore

$$\mathcal{I}_S = \mathbb{E}_{x \sim \mathsf{Uniform}(\mathcal{S}^{d-1}(\sqrt{d}))}\left[\frac{xx^T}{2 + \exp(\beta^{\star T}x) + \exp(-\beta^{\star T}x)}\right]$$

$$= \mathbb{E}_{z \sim \mathcal{N}(0, I_d)}\left[\frac{zz^T \frac{d}{\|z\|_2^2}}{2 + \exp(\beta^{\star T}z \cdot \frac{\sqrt{d}}{\|z\|_2}) + \exp(-\beta^{\star T}z \cdot \frac{\sqrt{d}}{\|z\|_2})}\right]$$

$$= \mathbb{E}_{z \sim \mathcal{N}(0, I_d)}\left[\frac{(\beta^\star \beta^{\star T} + U_\perp U_\perp^T)zz^T \frac{d}{\|z\|_2^2}}{2 + \exp(\beta^{\star T}z \cdot \frac{\sqrt{d}}{\|z\|_2}) + \exp(-\beta^{\star T}z \cdot \frac{\sqrt{d}}{\|z\|_2})}\right]$$

where $[\beta^\star, U_\perp] \in \mathbb{R}^{d \times d}$ is a orthogonal basis.

With this expression, we first prove $\beta^\star$ is an eigenvector of $\mathcal{I}_S$ with corresponding eigenvalue $\lambda_1$.

$$\mathcal{I}_S\beta^\star = \mathbb{E}_{z \sim \mathcal{N}(0, I_d)}\left[\frac{(\beta^\star \beta^{\star T} + U_\perp U_\perp^T)zz^T \frac{d}{\|z\|_2^2}}{2 + \exp(\beta^{\star T}z \cdot \frac{\sqrt{d}}{\|z\|_2}) + \exp(-\beta^{\star T}z \cdot \frac{\sqrt{d}}{\|z\|_2})}\right]\beta^\star$$

$$= \mathbb{E}_{z \sim \mathcal{N}(0, I_d)}\left[\frac{\beta^\star \beta^{\star T}zz^T \frac{d}{\|z\|_2^2}\beta^\star}{2 + \exp(\beta^{\star T}z \cdot \frac{\sqrt{d}}{\|z\|_2}) + \exp(-\beta^{\star T}z \cdot \frac{\sqrt{d}}{\|z\|_2})}\right]$$

$$+ \mathbb{E}_{z \sim \mathcal{N}(0, I_d)}\left[\frac{U_\perp U_\perp^T zz^T \frac{d}{\|z\|_2^2}\beta^\star}{2 + \exp(\beta^{\star T}z \cdot \frac{\sqrt{d}}{\|z\|_2}) + \exp(-\beta^{\star T}z \cdot \frac{\sqrt{d}}{\|z\|_2})}\right]$$

$$= \mathbb{E}_{z \sim \mathcal{N}(0, I_d)}\left[\frac{(\beta^{\star T}z)^2 \frac{d}{\|z\|_2^2}}{2 + \exp(\beta^{\star T}z \cdot \frac{\sqrt{d}}{\|z\|_2}) + \exp(-\beta^{\star T}z \cdot \frac{\sqrt{d}}{\|z\|_2})}\right]\beta^\star$$

$$+ \mathbb{E}_{z \sim \mathcal{N}(0, I_d)}\left[\frac{U_\perp U_\perp^T zz^T \frac{d}{\|z\|_2^2}\beta^\star}{2 + \exp(\beta^{\star T}z \cdot \frac{\sqrt{d}}{\|z\|_2}) + \exp(-\beta^{\star T}z \cdot \frac{\sqrt{d}}{\|z\|_2})}\right]$$

$$= \lambda_1\beta^\star + \mathbb{E}_{z \sim \mathcal{N}(0, I_d)}\left[\frac{U_\perp U_\perp^T zz^T \frac{d}{\|z\|_2^2}\beta^\star}{2 + \exp(\beta^{\star T}z \cdot \frac{\sqrt{d}}{\|z\|_2}) + \exp(-\beta^{\star T}z \cdot \frac{\sqrt{d}}{\|z\|_2})}\right].$$

Therefore we only need to prove

$$\mathbb{E}_{z\sim\mathcal{N}(0,I_d)}\Big[\frac{U_\perp U_\perp^T zz^T \frac{d}{\|z\|_2^2}\beta^\star}{2+\exp(\beta^{\star T}z\cdot\frac{\sqrt{d}}{\|z\|_2})+\exp(-\beta^{\star T}z\cdot\frac{\sqrt{d}}{\|z\|_2})}\Big]=0.$$

In fact,

$$\mathbb{E}_{z\sim\mathcal{N}(0,I_d)}\Big[\frac{U_\perp^T zz^T \frac{d}{\|z\|_2^2}\beta^\star}{2+\exp(\beta^{\star T}z\cdot\frac{\sqrt{d}}{\|z\|_2})+\exp(-\beta^{\star T}z\cdot\frac{\sqrt{d}}{\|z\|_2})}\Big]$$

$$=\mathbb{E}_{z\sim\mathcal{N}(0,I_d)}\Big[\frac{\frac{d}{\|z\|_2^2}(U_\perp^T z)(z^T\beta^\star)}{2+\exp(\beta^{\star T}z\cdot\frac{\sqrt{d}}{\|z\|_2})+\exp(-\beta^{\star T}z\cdot\frac{\sqrt{d}}{\|z\|_2})}\Big]$$

$$=\mathbb{E}_{z\sim\mathcal{N}(0,I_d)}\Big[\frac{\frac{d}{|A|^2+\|B\|^2}AB}{2+\exp(A\cdot\frac{\sqrt{d}}{\sqrt{|A|^2+\|B\|^2}})+\exp(-A\cdot\frac{\sqrt{d}}{|A|^2+\|B\|^2})}\Big]$$

where we let $A:=z^T\beta^\star$, $B:=U_\perp^T z$. Notice that by the property of $z\sim\mathcal{N}(0,I_d)$, $A$ and $B$ are independent. Also, $B$ is symmetric, i.e., $B$ and $-B$ have the same distribution. Therefore

$$\mathbb{E}_{z\sim\mathcal{N}(0,I_d)}\Big[\frac{\frac{d}{|A|^2+\|B\|^2}AB}{2+\exp(A\cdot\frac{\sqrt{d}}{\sqrt{|A|^2+\|B\|^2}})+\exp(-A\cdot\frac{\sqrt{d}}{|A|^2+\|B\|^2})}\Big]$$

$$\overset{\text{replace }B\text{ by }-B}{=}\mathbb{E}_{z\sim\mathcal{N}(0,I_d)}\Big[\frac{-\frac{d}{|A|^2+\|B\|^2}AB}{2+\exp(A\cdot\frac{\sqrt{d}}{\sqrt{|A|^2+\|B\|^2}})+\exp(-A\cdot\frac{\sqrt{d}}{|A|^2+\|B\|^2})}\Big]$$

$$=-\mathbb{E}_{z\sim\mathcal{N}(0,I_d)}\Big[\frac{\frac{d}{|A|^2+\|B\|^2}AB}{2+\exp(A\cdot\frac{\sqrt{d}}{\sqrt{|A|^2+\|B\|^2}})+\exp(-A\cdot\frac{\sqrt{d}}{|A|^2+\|B\|^2})}\Big],$$

which implies

$$\mathbb{E}_{z\sim\mathcal{N}(0,I_d)}\Big[\frac{U_\perp U_\perp^T zz^T \frac{d}{\|z\|_2^2}\beta^\star}{2+\exp(\beta^{\star T}z\cdot\frac{\sqrt{d}}{\|z\|_2})+\exp(-\beta^{\star T}z\cdot\frac{\sqrt{d}}{\|z\|_2})}\Big]=0.$$

Next we will prove that for any $\beta_\perp$ such that $\|\beta_\perp\|_2=1$, $\beta^{\star T}\beta_\perp=0$, $\beta_\perp$ is an eigenvector of $\mathcal{I}_S$ with corresponding eigenvalue $\lambda_2$. Let $[\beta_\perp,U]$ be an orthogonal basis ($\beta^\star$ is the first column of $U$).

$$\mathcal{I}_S\beta_\perp=\mathbb{E}_{z\sim\mathcal{N}(0,I_d)}\Big[\frac{(\beta_\perp\beta_\perp^T+UU^T)zz^T\frac{d}{\|z\|_2^2}}{2+\exp(\beta^{\star T}z\cdot\frac{\sqrt{d}}{\|z\|_2})+\exp(-\beta^{\star T}z\cdot\frac{\sqrt{d}}{\|z\|_2})}\Big]\beta_\perp$$

$$=\mathbb{E}_{z\sim\mathcal{N}(0,I_d)}\Big[\frac{\beta_\perp\beta_\perp^T zz^T\frac{d}{\|z\|_2^2}\beta_\perp}{2+\exp(\beta^{\star T}z\cdot\frac{\sqrt{d}}{\|z\|_2})+\exp(-\beta^{\star T}z\cdot\frac{\sqrt{d}}{\|z\|_2})}\Big]$$

$$+\mathbb{E}_{z\sim\mathcal{N}(0,I_d)}\Big[\frac{UU^T zz^T\frac{d}{\|z\|_2^2}\beta_\perp}{2+\exp(\beta^{\star T}z\cdot\frac{\sqrt{d}}{\|z\|_2})+\exp(-\beta^{\star T}z\cdot\frac{\sqrt{d}}{\|z\|_2})}\Big]$$

$$=\mathbb{E}_{z\sim\mathcal{N}(0,I_d)}\Big[\frac{(\beta_\perp^T z)^2\frac{d}{\|z\|_2^2}}{2+\exp(\beta^{\star T}z\cdot\frac{\sqrt{d}}{\|z\|_2})+\exp(-\beta^{\star T}z\cdot\frac{\sqrt{d}}{\|z\|_2})}\Big]\beta_\perp$$

$$+\mathbb{E}_{z\sim\mathcal{N}(0,I_d)}\Big[\frac{UU^T zz^T\frac{d}{\|z\|_2^2}\beta_\perp}{2+\exp(\beta^{\star T}z\cdot\frac{\sqrt{d}}{\|z\|_2})+\exp(-\beta^{\star T}z\cdot\frac{\sqrt{d}}{\|z\|_2})}\Big]$$

$$=\lambda_2\beta_\perp+0$$

$$=\lambda_2\beta_\perp$$

Here

$$\mathbb{E}_{z\sim\mathcal{N}(0,I_d)}\Big[\frac{UU^Tzz^T\frac{d}{\|z\|_2^2}\beta_\perp}{2+\exp(\beta^{\star T}z\cdot\frac{\sqrt{d}}{\|z\|_2})+\exp(-\beta^{\star T}z\cdot\frac{\sqrt{d}}{\|z\|_2})}\Big]=0$$

because of a similar reason as in the previous part.

For $\mathcal{I}_T$, the proving strategy is similar. For $x\sim\mathsf{Uniform}(\mathcal{S}^{d-1}(\sqrt{d}))+v$ on the target domain, where $v=r\beta_\perp^\star$, let $w=x-v=x-r\beta_\perp^\star$, then $w\sim\mathsf{Uniform}(\mathcal{S}^{d-1}(\sqrt{d}))$. Let $z\sim\mathcal{N}(0,I_d)$, then $w$ and $z\frac{\sqrt{d}}{\|z\|_2}$ have the same distribution. We have

$$\mathcal{I}_T=\mathbb{E}_{x\sim\mathsf{Uniform}(\mathcal{S}^{d-1}(\sqrt{d}))+v}\Big[\frac{xx^T}{2+\exp(\beta^{\star T}x)+\exp(-\beta^{\star T}x)}\Big]$$

$$=\mathbb{E}_{w\sim\mathsf{Uniform}(\mathcal{S}^{d-1}(\sqrt{d}))}\Big[\frac{(w+v)(w+v)^T}{2+\exp(\beta^{\star T}(w+v))+\exp(-\beta^{\star T}(w+v))}\Big]$$

$$\overset{v^T\beta^\star=0}{=}\mathbb{E}_{w\sim\mathsf{Uniform}(\mathcal{S}^{d-1}(\sqrt{d}))}\Big[\frac{ww^T+wv^T+vw^T+vv^T}{2+\exp(\beta^{\star T}w)+\exp(-\beta^{\star T}w)}\Big]$$

Therefore

$$\mathcal{I}_T\beta^\star=\mathbb{E}_{w\sim\mathsf{Uniform}(\mathcal{S}^{d-1}(\sqrt{d}))}\Big[\frac{ww^T+wv^T+vw^T+vv^T}{2+\exp(\beta^{\star T}w)+\exp(-\beta^{\star T}w)}\Big]\beta^\star$$

$$\overset{v^T\beta^\star=0}{=}\mathbb{E}_{w\sim\mathsf{Uniform}(\mathcal{S}^{d-1}(\sqrt{d}))}\Big[\frac{ww^T}{2+\exp(\beta^{\star T}w)+\exp(-\beta^{\star T}w)}\Big]\beta^\star$$

$$=\mathcal{I}_S\beta^\star$$

$$=\lambda_1\beta^\star,$$

where the last line follows from the previous proofs. Similarly, for any $\tilde{\beta}_\perp$ such that $\|\tilde{\beta}_\perp\|_2=1$, $\beta_\perp^{\star T}\tilde{\beta}_\perp=0$,

$$\mathcal{I}_T\tilde{\beta}_\perp=\mathbb{E}_{w\sim\mathsf{Uniform}(\mathcal{S}^{d-1}(\sqrt{d}))}\Big[\frac{ww^T+wv^T+vw^T+vv^T}{2+\exp(\beta^{\star T}w)+\exp(-\beta^{\star T}w)}\Big]\tilde{\beta}_\perp$$

$$\overset{v^T\tilde{\beta}_\perp=0}{=}\mathbb{E}_{w\sim\mathsf{Uniform}(\mathcal{S}^{d-1}(\sqrt{d}))}\Big[\frac{ww^T}{2+\exp(\beta^{\star T}w)+\exp(-\beta^{\star T}w)}\Big]\tilde{\beta}_\perp$$

$$=\mathcal{I}_S\tilde{\beta}_\perp$$

$$=\lambda_2\tilde{\beta}_\perp.$$

For $\beta_\perp^\star$,

$$\mathcal{I}_T\beta_\perp^\star=\mathbb{E}_{w\sim\mathsf{Uniform}(\mathcal{S}^{d-1}(\sqrt{d}))}\Big[\frac{ww^T+wv^T+vw^T+vv^T}{2+\exp(\beta^{\star T}w)+\exp(-\beta^{\star T}w)}\Big]\beta_\perp^\star$$

$$=\mathbb{E}_{w\sim\mathsf{Uniform}(\mathcal{S}^{d-1}(\sqrt{d}))}\Big[\frac{ww^T}{2+\exp(\beta^{\star T}w)+\exp(-\beta^{\star T}w)}\Big]\beta_\perp^\star$$

$$+\mathbb{E}_{w\sim\mathsf{Uniform}(\mathcal{S}^{d-1}(\sqrt{d}))}\Big[\frac{wv^T}{2+\exp(\beta^{\star T}w)+\exp(-\beta^{\star T}w)}\Big]\beta_\perp^\star$$

$$+\mathbb{E}_{w\sim\mathsf{Uniform}(\mathcal{S}^{d-1}(\sqrt{d}))}\Big[\frac{vw^T}{2+\exp(\beta^{\star T}w)+\exp(-\beta^{\star T}w)}\Big]\beta_\perp^\star$$

$$+\mathbb{E}_{w\sim\mathsf{Uniform}(\mathcal{S}^{d-1}(\sqrt{d}))}\Big[\frac{vv^T}{2+\exp(\beta^{\star T}w)+\exp(-\beta^{\star T}w)}\Big]\beta_\perp^\star$$

$$:=I_1+I_2+I_3+I_4.$$

As in the previous proofs,

$$I_1=\mathcal{I}_S\beta_\perp^\star=\lambda_2\beta_\perp^\star.$$

$$I_2 = \mathbb{E}_{w \sim \mathsf{Uniform}(\mathcal{S}^{d-1}(\sqrt{d}))}\Big[\frac{wv^T}{2 + \exp(\beta^{\star T}w) + \exp(-\beta^{\star T}w)}\Big]\beta_\perp^\star$$

$$\overset{v=r\beta_\perp^\star}{=} r\mathbb{E}_{w \sim \mathsf{Uniform}(\mathcal{S}^{d-1}(\sqrt{d}))}\Big[\frac{w\beta_\perp^{\star T}\beta_\perp^\star}{2 + \exp(\beta^{\star T}w) + \exp(-\beta^{\star T}w)}\Big]$$

$$\overset{\|\beta_\perp^\star\|=1}{=} r\mathbb{E}_{w \sim \mathsf{Uniform}(\mathcal{S}^{d-1}(\sqrt{d}))}\Big[\frac{w}{2 + \exp(\beta^{\star T}w) + \exp(-\beta^{\star T}w)}\Big]$$

$$= 0.$$

where the last lines follows from $w$ is symmetric and $\frac{w}{2+\exp(\beta^{\star T}w)+\exp(-\beta^{\star T}w)}$ is a odd function of $w$.

$$I_3 = \mathbb{E}_{w \sim \mathsf{Uniform}(\mathcal{S}^{d-1}(\sqrt{d}))}\Big[\frac{vw^T}{2 + \exp(\beta^{\star T}w) + \exp(-\beta^{\star T}w)}\Big]\beta_\perp^\star$$

$$\overset{v=r\beta_\perp^\star}{=} r\mathbb{E}_{w \sim \mathsf{Uniform}(\mathcal{S}^{d-1}(\sqrt{d}))}\Big[\frac{\beta_\perp^\star w^T \beta_\perp^\star}{2 + \exp(\beta^{\star T}w) + \exp(-\beta^{\star T}w)}\Big]$$

$$= r\mathbb{E}_{w \sim \mathsf{Uniform}(\mathcal{S}^{d-1}(\sqrt{d}))}\Big[\frac{w^T \beta_\perp^\star}{2 + \exp(\beta^{\star T}w) + \exp(-\beta^{\star T}w)}\Big]\beta_\perp^\star$$

$$= 0.$$

where the last lines follows from $w$ is symmetric and $\frac{w^T\beta_\perp^\star}{2+\exp(\beta^{\star T}w)+\exp(-\beta^{\star T}w)}$ is a odd function of $w$.

$$I_4 = \mathbb{E}_{w \sim \mathsf{Uniform}(\mathcal{S}^{d-1}(\sqrt{d}))}\Big[\frac{vv^T}{2 + \exp(\beta^{\star T}w) + \exp(-\beta^{\star T}w)}\Big]\beta_\perp^\star$$

$$\overset{v=r\beta_\perp^\star}{=} r^2\mathbb{E}_{w \sim \mathsf{Uniform}(\mathcal{S}^{d-1}(\sqrt{d}))}\Big[\frac{\beta_\perp^\star \beta_\perp^{\star T}\beta_\perp^\star}{2 + \exp(\beta^{\star T}w) + \exp(-\beta^{\star T}w)}\Big]$$

$$\overset{\|\beta_\perp^\star\|=1}{=} r^2\mathbb{E}_{w \sim \mathsf{Uniform}(\mathcal{S}^{d-1}(\sqrt{d}))}\Big[\frac{1}{2 + \exp(\beta^{\star T}w) + \exp(-\beta^{\star T}w)}\Big]\beta_\perp^\star$$

$$= r^2\lambda_3\beta_\perp^\star.$$

Combine the calculations of $I_1, I_2, I_3, I_4$, we have

$$\mathcal{I}_T\beta_\perp^\star = I_1 + I_2 + I_3 + I_4$$
$$= \lambda_2\beta_\perp^\star + r^2\lambda_3\beta_\perp^\star$$
$$= (\lambda_2 + r^2\lambda_3)\beta_\perp^\star.$$

In conclusion, we have $\mathcal{I}_S = U\mathsf{diag}(\lambda_1, \lambda_2, \ldots, \lambda_2)U^T$ and $\mathcal{I}_T = U\mathsf{diag}(\lambda_1, \lambda_2 + r^2\lambda_3, \lambda_2, \ldots, \lambda_2)U^T$ for an orthonormal matrix $U$, where $U = [\beta^\star, \beta_\perp^\star, \cdots]$. $\qquad\square$

*Proof of Lemma B.2.* Recall the definition of $\lambda_1, \lambda_2, \lambda_3$:

$$\lambda_1 := \mathbb{E}_{x \sim \mathsf{Uniform}(\mathcal{S}^{d-1}(\sqrt{d}))}\Big[\frac{(\beta^{\star T}x)^2}{2 + \exp(\beta^{\star T}x) + \exp(-\beta^{\star T}x)}\Big] = \mathbb{E}_{z \sim \mathcal{N}(0, I_d)}\Big[\frac{\frac{d}{\|z\|_2^2}(\beta^{\star T}z)^2}{2 + \exp(\frac{\sqrt{d}}{\|z\|_2}\beta^{\star T}z) + \exp(-\frac{\sqrt{d}}{\|z\|_2}\beta^{\star T}z)}\Big],$$

$$\lambda_2 := \mathbb{E}_{x \sim \mathsf{Uniform}(\mathcal{S}^{d-1}(\sqrt{d}))}\Big[\frac{(\beta_\perp^{\star T}x)^2}{2 + \exp(\beta^{\star T}x) + \exp(-\beta^{\star T}x)}\Big] = \mathbb{E}_{z \sim \mathcal{N}(0, I_d)}\Big[\frac{\frac{d}{\|z\|_2^2}(\beta_\perp^{\star T}z)^2}{2 + \exp(\frac{\sqrt{d}}{\|z\|_2}\beta^{\star T}z) + \exp(-\frac{\sqrt{d}}{\|z\|_2}\beta^{\star T}z)}\Big],$$

$$\lambda_3 := \mathbb{E}_{x \sim \mathsf{Uniform}(\mathcal{S}^{d-1}(\sqrt{d}))}\Big[\frac{1}{2 + \exp(\beta^{\star T}x) + \exp(-\beta^{\star T}x)}\Big] = \mathbb{E}_{z \sim \mathcal{N}(0, I_d)}\Big[\frac{1}{2 + \exp(\frac{\sqrt{d}}{\|z\|_2}\beta^{\star T}z) + \exp(-\frac{\sqrt{d}}{\|z\|_2}\beta^{\star T}z)}\Big].$$

Next we will show that there exists constants $c, C, c' > 0$ such that when $d \geq c'$, we have $c \leq \lambda_1 \leq C$. The proofs for $\lambda_2$ and $\lambda_3$ are similar. Notice that, when $d$ is large, $\frac{d}{\|z\|_2^2}$ concentrates around 1. If we replace $\frac{d}{\|z\|_2^2}$ by 1 in the above expressions, we have

$$\lambda_1 \approx \mathbb{E}_{z \sim \mathcal{N}(0, I_d)}\Big[\frac{(\beta^{\star T}z)^2}{2 + \exp(\beta^{\star T}z) + \exp(-\beta^{\star T}z)}\Big]$$

Since $\beta^{\star T} z \sim \mathcal{N}(0, 1)$ when $z \sim \mathcal{N}(0, I_d)$ and $\|\beta^\star\| = 1$, we have

$$\mathbb{E}_{z\sim\mathcal{N}(0,I_d)}\left[\frac{(\beta^{\star T}z)^2}{2+\exp(\beta^{\star T}z)+\exp(-\beta^{\star T}z)}\right] = \mathbb{E}_{y\sim\mathcal{N}(0,1)}\left[\frac{y^2}{2+\exp(y)+\exp(-y)}\right]$$

which is a absolute constant greater than zero and not related to $d$. Following this intuition, we can bound $\lambda_1$ as the following. We first state the concentration of the norm of $\mathcal{N}(0, I_d)$. By Vershynin (2018) (3.7),

$$\mathbb{P}(|\|z\| - \sqrt{d}| \geq t) \leq 2e^{-4ct^2} \tag{35}$$

for some absolute constant $c > 0$. Take $t = \frac{\sqrt{d}}{2}$, we have

$$\mathbb{P}(\frac{\|z\|}{\sqrt{d}} \notin [\frac{1}{2}, \frac{3}{2}]) \leq 2e^{-cd}.$$

With this concentration, we do the following truncation:

$$\lambda_1 = \mathbb{E}_{z\sim\mathcal{N}(0,I_d)}\left[\frac{\frac{d}{\|z\|_2^2}(\beta^{\star T}z)^2}{2+\exp(\frac{\sqrt{d}}{\|z\|_2}\beta^{\star T}z)+\exp(-\frac{\sqrt{d}}{\|z\|_2}\beta^{\star T}z)}\right]$$

$$= \mathbb{E}_{z\sim\mathcal{N}(0,I_d)}\left[\frac{\frac{d}{\|z\|_2^2}(\beta^{\star T}z)^2}{2+\exp(\frac{\sqrt{d}}{\|z\|_2}\beta^{\star T}z)+\exp(-\frac{\sqrt{d}}{\|z\|_2}\beta^{\star T}z)}\mathbb{I}_{\frac{\|z\|}{\sqrt{d}}\in[\frac{1}{2},\frac{3}{2}]}\right]$$

$$+ \mathbb{E}_{z\sim\mathcal{N}(0,I_d)}\left[\frac{\frac{d}{\|z\|_2^2}(\beta^{\star T}z)^2}{2+\exp(\frac{\sqrt{d}}{\|z\|_2}\beta^{\star T}z)+\exp(-\frac{\sqrt{d}}{\|z\|_2}\beta^{\star T}z)}\mathbb{I}_{\frac{\|z\|}{\sqrt{d}}\notin[\frac{1}{2},\frac{3}{2}]}\right]$$

$$:= J_1 + J_2.$$

For $J_2$, it is obvious that

$$0 \leq J_2 \leq \frac{d}{4}\mathbb{P}(\frac{\|z\|}{\sqrt{d}} \notin [\frac{1}{2}, \frac{3}{2}]) \leq \frac{d}{2}e^{-cd}. \tag{36}$$

For upper bound of $J_1$,

$$J_1 = \mathbb{E}_{z\sim\mathcal{N}(0,I_d)}\left[\frac{\frac{d}{\|z\|_2^2}(\beta^{\star T}z)^2}{2+\exp(\frac{\sqrt{d}}{\|z\|_2}\beta^{\star T}z)+\exp(-\frac{\sqrt{d}}{\|z\|_2}\beta^{\star T}z)}\mathbb{I}_{\frac{\|z\|}{\sqrt{d}}\in[\frac{1}{2},\frac{3}{2}]}\right]$$

$$\leq \mathbb{E}_{z\sim\mathcal{N}(0,I_d)}\left[\frac{4(\beta^{\star T}z)^2}{4}\right] = 1.$$

Therefore

$$\lambda_1 = J_1 + J_2$$
$$\leq 1 + \frac{d}{2}e^{-cd}.$$

It's obvious that there exists an absolute constant $c'$ such that when $d \geq c'$, $\lambda_1 \leq 2$.

For lower bound of $J_1$, we have

$$J_1 = \mathbb{E}_{z\sim\mathcal{N}(0,I_d)}\left[\frac{\frac{d}{\|z\|_2^2}(\beta^{\star T}z)^2}{2+\exp(\frac{\sqrt{d}}{\|z\|_2}\beta^{\star T}z)+\exp(-\frac{\sqrt{d}}{\|z\|_2}\beta^{\star T}z)}\mathbb{I}_{\frac{\|z\|}{\sqrt{d}}\in[\frac{1}{2},\frac{3}{2}]}\right]$$

$$\geq \mathbb{E}_{z\sim\mathcal{N}(0,I_d)}\left[\frac{\frac{4}{9}(\beta^{\star T}z)^2}{2+\exp(2\beta^{\star T}z)+\exp(-2\beta^{\star T}z)}\mathbb{I}_{\frac{\|z\|}{\sqrt{d}}\in[\frac{1}{2},\frac{3}{2}]}\right]$$

$$= \mathbb{E}_{z\sim\mathcal{N}(0,I_d)}\left[\frac{\frac{4}{9}(\beta^{\star T}z)^2}{2+\exp(2\beta^{\star T}z)+\exp(-2\beta^{\star T}z)}\right] - \mathbb{E}_{z\sim\mathcal{N}(0,I_d)}\left[\frac{\frac{4}{9}(\beta^{\star T}z)^2}{2+\exp(2\beta^{\star T}z)+\exp(-2\beta^{\star T}z)}\mathbb{I}_{\frac{\|z\|}{\sqrt{d}}\notin[\frac{1}{2},\frac{3}{2}]}\right]$$

$$\geq \mathbb{E}_{z\sim\mathcal{N}(0,I_d)}\left[\frac{\frac{4}{9}(\beta^{\star T}z)^2}{2+\exp(2\beta^{\star T}z)+\exp(-2\beta^{\star T}z)}\right] - \mathbb{E}_{z\sim\mathcal{N}(0,I_d)}\left[\frac{\frac{4}{9}(\beta^{\star T}z)^2}{4}\mathbb{I}_{\frac{\|z\|}{\sqrt{d}}\notin[\frac{1}{2},\frac{3}{2}]}\right]$$

$$\geq \mathbb{E}_{z\sim\mathcal{N}(0,I_d)}\left[\frac{\frac{4}{9}(\beta^{\star T}z)^2}{2+\exp(2\beta^{\star T}z)+\exp(-2\beta^{\star T}z)}\right] - \mathbb{E}_{z\sim\mathcal{N}(0,I_d)}\left[\frac{\|z\|_2^2}{9}\mathbb{I}_{\frac{\|z\|}{\sqrt{d}}\notin[\frac{1}{2},\frac{3}{2}]}\right]$$

$$= \mathbb{E}_{y\sim\mathcal{N}(0,1)}\left[\frac{\frac{4}{9}y^2}{2+\exp(2y)+\exp(-2y)}\right] - \mathbb{E}_{z\sim\mathcal{N}(0,I_d)}\left[\frac{\|z\|_2^2}{9}\mathbb{I}_{\frac{\|z\|}{\sqrt{d}}\notin[\frac{1}{2},\frac{3}{2}]}\right]$$

$$:= c_1 - \mathbb{E}_{z\sim\mathcal{N}(0,I_d)}\left[\frac{\|z\|_2^2}{9}\mathbb{I}_{\frac{\|z\|}{\sqrt{d}}\notin[\frac{1}{2},\frac{3}{2}]}\right]$$

Notice that here $c_1$ is a positive constant not related to $d$. For the second term,

$$\mathbb{E}_{z\sim\mathcal{N}(0,I_d)}\left[\frac{\|z\|_2^2}{9}\mathbb{I}_{\frac{\|z\|}{\sqrt{d}}\notin[\frac{1}{2},\frac{3}{2}]}\right]$$

$$= \mathbb{E}_{z\sim\mathcal{N}(0,I_d)}\left[\frac{\|z\|_2^2}{9}\mathbb{I}_{\frac{\|z\|}{\sqrt{d}}\leq\frac{1}{2}}\right] + \mathbb{E}_{z\sim\mathcal{N}(0,I_d)}\left[\frac{\|z\|_2^2}{9}\mathbb{I}_{\frac{\|z\|}{\sqrt{d}}\geq\frac{3}{2}}\right]$$

$$\leq \frac{d}{36}\mathbb{P}\left(\frac{\|z\|}{\sqrt{d}}\leq\frac{1}{2}\right) + \frac{1}{9}\int_{\frac{9}{4}d}^{\infty}\mathbb{P}(\|z\|_2^2\geq t)\mathrm{d}t + \frac{1}{9}\cdot\frac{9}{4}d\mathbb{P}(\|z\|_2^2\geq\frac{9}{4}d)$$

$$\overset{\text{by (35)}}{\leq} \frac{d}{36}2e^{-cd} + \frac{1}{9}\int_{\frac{9}{4}d}^{\infty}\mathbb{P}(\|z\|_2^2\geq t)\mathrm{d}t + \frac{d}{4}2e^{-cd}$$

$$\overset{t=d(y+1)^2}{\leq} de^{-cd} + \frac{1}{9}\int_{\frac{1}{2}}^{\infty}2d(y+1)\mathbb{P}(\|z\|_2\geq\sqrt{d}+\sqrt{d}y)\mathrm{d}y$$

$$\overset{\text{by (35)}}{\leq} de^{-cd} + \frac{1}{9}\int_{\frac{1}{2}}^{\infty}2d(y+1)2e^{-4cdy^2}\mathrm{d}y$$

$$\leq de^{-cd} + 2d\int_{\frac{1}{2}}^{\infty}ye^{-4cdy^2}\mathrm{d}y$$

$$\leq de^{-cd} + \frac{1}{4c}e^{-cd}$$

Combine this inequality and previous inequalities of $J_1$ and $J_2$, we have

$$\lambda_1 = J_1 + J_2$$

$$\geq c_1 - de^{-cd} - \frac{1}{4c}e^{-cd}$$

Therefore it's obvious that there exists an absolute constant $c'$ such that when $d \geq c'$, $\lambda_1 \geq \frac{c_1}{2}$.

The proof for $\lambda_2$ is almost the same, the only difference is that in the numerator, we replace $\beta^{\star T}z$ by $\beta_\perp^{\star T}z$. The proof for $\lambda_3$ is even simpler. For upper bound,

$$\lambda_3 = \mathbb{E}_{z\sim\mathcal{N}(0,I_d)}\left[\frac{1}{2+\exp(\frac{\sqrt{d}}{\|z\|_2}\beta^{\star T}z)+\exp(-\frac{\sqrt{d}}{\|z\|_2}\beta^{\star T}z)}\right] \leq \frac{1}{4}.$$

For lower bound,

$$\lambda_3 = \mathbb{E}_{z \sim \mathcal{N}(0, I_d)}\Big[\frac{1}{2 + \exp(\frac{\sqrt{d}}{\|z\|_2}\beta^{\star T}z) + \exp(-\frac{\sqrt{d}}{\|z\|_2}\beta^{\star T}z)}\Big]$$

$$\geq \mathbb{E}_{z \sim \mathcal{N}(0, I_d)}\Big[\frac{1}{2 + \exp(\frac{\sqrt{d}}{\|z\|_2}\beta^{\star T}z) + \exp(-\frac{\sqrt{d}}{\|z\|_2}\beta^{\star T}z)}\mathbb{I}_{\frac{\|z\|}{\sqrt{d}} \in [\frac{1}{2}, \frac{3}{2}]}\Big]$$

$$\geq \mathbb{E}_{z \sim \mathcal{N}(0, I_d)}\Big[\frac{1}{2 + \exp(2\beta^{\star T}z) + \exp(-2\beta^{\star T}z)}\mathbb{I}_{\frac{\|z\|}{\sqrt{d}} \in [\frac{1}{2}, \frac{3}{2}]}\Big]$$

$$= \mathbb{E}_{z \sim \mathcal{N}(0, I_d)}\Big[\frac{1}{2 + \exp(2\beta^{\star T}z) + \exp(-2\beta^{\star T}z)}\Big] - \mathbb{E}_{z \sim \mathcal{N}(0, I_d)}\Big[\frac{1}{2 + \exp(2\beta^{\star T}z) + \exp(-2\beta^{\star T}z)}\mathbb{I}_{\frac{\|z\|}{\sqrt{d}} \notin [\frac{1}{2}, \frac{3}{2}]}\Big]$$

$$= c_2 - \frac{1}{4}\mathbb{P}(\frac{\|z\|}{\sqrt{d}} \notin [\frac{1}{2}, \frac{3}{2}])$$

$$\geq c_2 - \frac{1}{2}e^{-cd}.$$

Therefore there exists constant $c'$ such that when $d \geq c'$, $\lambda_3 \leq \frac{c_2}{2}$. $\qquad\square$

### B.3 PROOFS FOR THEOREM 4.5

In this section, our objective is to establish the upper bound of MLE for the phase retrieval model. A direct application of Theorem 3.1 is impractical, as Assumption A.3 is not met; notably, both $\beta^\star, -\beta^\star$ serve as global minimums of population loss. To circumvent the issue of non-unique global minimums, we employ a methodology similar to that used in proving Theorem 3.1, though with a slightly refined analysis.

*Proof of Theorem 4.5.* In the sequel, we will use the same notations as in the proof of Theorem 3.1. Even though the global minimum of population loss for the phase retrieval model isn't unique, meaning it could be either $\beta^\star$ or $-\beta^\star$, we can still show that the MLE falls into a small ball around either $\beta^\star$ or $-\beta^\star$.

**Lemma B.3.** *Under the settings of Theorem 4.5, if $n \geq \mathcal{O}(d^4 \log \frac{d}{\delta})$, then with probability at least $1 - \delta$, we have*

$$\min\{\|\beta_{\mathsf{MLE}} - \beta^\star\|_2, \|\beta_{\mathsf{MLE}} + \beta^\star\|_2\} \lesssim \sqrt{\frac{d^2 \log \frac{d}{\delta}}{n}}.$$

Without loss of generality, in the sequel, we consider $n \geq \mathcal{O}(d^4 \log \frac{d}{\delta})$ and assume

$$\|\beta_{\mathsf{MLE}} - \beta^\star\|_2 \lesssim \sqrt{\frac{d^2 \log \frac{d}{\delta}}{n}}, \tag{37}$$

which implies $\beta_{\mathsf{MLE}} \in \mathbb{B}_{\beta^\star}(1)$.

Recall that for the phase retrieval model,

$$\ell(x, y, \beta) = \frac{1}{2}\log(2\pi) + \frac{1}{2}\left(y - (x^T\beta)^2\right)^2.$$

It then holds that

$$\nabla\ell(x, y, \beta) = 2(x^T\beta)^3 x - 2(x^T\beta)yx,$$
$$\nabla^2\ell(x, y, \beta) = 6(x^T\beta)^2 xx^T - 2yxx^T,$$
$$\nabla^3\ell(x, y, \beta) = 12(x^T\beta)x \otimes x \otimes x.$$

Note that for $Y = (X^T\beta^\star)^2 + \varepsilon$, we have $\nabla\ell(X, Y, \beta^\star) = -2(X^T\beta^\star)X\varepsilon$. Therefore (recall that $\|\beta^\star\| = 1$) $\|\nabla\ell(x_i, y_i, \beta^\star)\|$ is $2d$-subgaussian, by Lemma D.1, we have for any $\delta$, with probability at least $1 - \delta$,

$$\|\mathcal{I}_S^{-1}g\|_2 \lesssim \sqrt{\frac{\mathsf{Tr}(\mathcal{I}_S^{-1})\log\frac{d}{\delta}}{n}} + d\|\mathcal{I}_S^{-1}\|\sqrt{\log\frac{d^2\|\mathcal{I}_S^{-1}\|^2}{\mathsf{Tr}(\mathcal{I}_S^{-1})}}\frac{\log\frac{d}{\delta}}{n}. \tag{38}$$

Which can be viewed as setting $B_1 = d$ and $\gamma = \frac{1}{2}$ in Assumption A.1. Hence $\beta^\star + z = \beta^\star - \mathcal{I}_S^{-1} g \in \mathbb{B}_{\beta^\star}(1)$ when $n \geq \mathcal{O}(\max\{\mathsf{Tr}(\mathcal{I}_S^{-1}) \log \frac{d}{\delta}, d\|\mathcal{I}_S^{-1}\|_2 \sqrt{\log \frac{d^2 \|\mathcal{I}_S^{-1}\|^2}{\mathsf{Tr}(\mathcal{I}_S^{-1})} \log \frac{d}{\delta}}\})$.

We then show the concentration inequality for the Hessian matrix. Note that

$$\nabla^2 \ell_n(\beta^\star) = \frac{1}{n} \sum_{i=1}^n \nabla^2 \ell(x_i, y_i, \beta^\star) = \frac{4}{n} \sum_{i=1}^n (x_i^T \beta^\star)^2 x_i x_i^T - \frac{2}{n} \sum_{i=1}^n \varepsilon_i x_i x_i^T.$$

Since $\|(x^T \beta^\star)^2 x x^T\| \leq d^2$, by matrix Hoeffding, with probability at least $1 - \delta$, we have

$$\mathbb{E}_{\mathbb{P}_S}[(x^T \beta^\star)^2 x x^T] - d^2 \sqrt{\frac{8 \log \frac{d}{\delta}}{n}} I_d \preceq \frac{1}{n} \sum_{i=1}^n (x_i^T \beta^\star)^2 x_i x_i^T \preceq \mathbb{E}_{\mathbb{P}_S}[(x^T \beta^\star)^2 x x^T] + d^2 \sqrt{\frac{8 \log \frac{d}{\delta}}{n}} I_d \tag{39}$$

Moreover, by matrix Chernoff bound, with probability at least $1 - \delta$, we have

$$-d\sqrt{\frac{8 \log \frac{d}{\delta}}{n}} I_d \preceq -\frac{1}{n} \sum_{i=1}^n \varepsilon_i x_i x_i^T \preceq d\sqrt{\frac{8 \log \frac{d}{\delta}}{n}} I_d. \tag{40}$$

Combine (39) and (40), we obtain

$$\nabla^2 \ell(\beta^\star) - 6d^2 \sqrt{\frac{8 \log \frac{d}{\delta}}{n}} I_d \preceq \nabla^2 \ell_n(\beta^\star) \preceq \nabla^2 \ell(\beta^\star) + 6d^2 \sqrt{\frac{8 \log \frac{d}{\delta}}{n}} I_d, \tag{41}$$

which can be viewed as setting $B_2 = d^2$ in (9).

For any $\beta \in \mathbb{B}_{\beta^\star}(1)$, we have

$$\|\nabla^3 \ell(x, y, \beta)\|_2 = 12\|(x^T \beta) x \otimes x \otimes x\| \leq 24(\sqrt{d} + r)^4.$$

Thus, we can view as if this model satisfies $B_3 = (\sqrt{d} + r)^4$ in Assumption A.2.

Then same as (12) we have with probability $1 - \delta$,

$$\ell_n(\beta^\star + z) - \ell_n(\beta^\star) \leq -\frac{1}{2} z^T \mathcal{I}_S z + 2c^2 B_2 \mathsf{Tr}(\mathcal{I}_S^{-1})(\frac{\log \frac{d}{\delta}}{n})^{1.5} + 2B_1^2 B_2 \|\mathcal{I}_S^{-1}\|_2^2 \log(\tilde{\kappa}^{-1/2} \alpha_1)(\frac{\log \frac{d}{\delta}}{n})^{2.5}$$
$$+ \frac{2}{3} c^3 B_3 \mathsf{Tr}(\mathcal{I}_S^{-1})^{1.5}(\frac{\log \frac{d}{\delta}}{n})^{1.5} + \frac{2}{3} B_1^3 B_3 \|\mathcal{I}_S^{-1}\|_2^3 \log^{1.5}(\tilde{\kappa}^{-1/2} \alpha_1)(\frac{\log \frac{d}{\delta}}{n})^3, \tag{42}$$

By Lemma B.3, we have (37). Then same as (13) we have with probability at least $1 - \delta$,

$$\ell_n(\beta_{\mathsf{MLE}}) - \ell_n(\beta^\star) \geq \frac{1}{2}(\Delta_{\beta_{\mathsf{MLE}}} - z)^T \mathcal{I}_S (\Delta_{\beta_{\mathsf{MLE}}} - z) - \frac{1}{2} z^T \mathcal{I}_S z$$
$$- \mathcal{O}(B_2 d^2 (\frac{\log \frac{d}{\delta}}{n})^{1.5} + B_3 d^3 (\frac{\log \frac{d}{\delta}}{n})^{1.5}). \tag{43}$$

Consequently, by (42), (43) and the fact that $\ell_n(\beta_{\mathsf{MLE}}) - \ell_n(\beta^\star + z) \leq 0$, we have

$$(\Delta_{\beta_{\mathsf{MLE}}} - z)^T \mathcal{I}_S (\Delta_{\beta_{\mathsf{MLE}}} - z) \leq \mathcal{O}\Bigg( B_2 \mathsf{Tr}(\mathcal{I}_S^{-1})(\frac{\log \frac{d}{\delta}}{n})^{1.5} + B_1^2 B_2 \|\mathcal{I}_S^{-1}\|_2^2 \log(\tilde{\kappa}^{-1/2} \alpha_1)(\frac{\log \frac{d}{\delta}}{n})^{2.5}$$
$$+ B_3 \mathsf{Tr}(\mathcal{I}_S^{-1})^{1.5}(\frac{\log \frac{d}{\delta}}{n})^{1.5} + B_1^3 B_3 \|\mathcal{I}_S^{-1}\|_2^3 (\log(\tilde{\kappa}^{-1/2} \alpha_1))^{1.5}(\frac{\log \frac{d}{\delta}}{n})^3$$
$$+ B_2 d^2 (\frac{\log \frac{d}{\delta}}{n})^{1.5} + B_3 d^3 (\frac{\log \frac{d}{\delta}}{n})^{1.5} \Bigg)$$

Then, same as the proof of Lemma A.3, we further have for any $\delta$, with probability at least $1 - 2\delta$,

$$(\beta_{\mathsf{MLE}} - \beta^\star)^T \mathcal{I}_T (\beta_{\mathsf{MLE}} - \beta^\star)$$

$$\lesssim \frac{\mathsf{Tr}(\mathcal{I}_T \mathcal{I}_S^{-1}) \log \frac{d}{\delta}}{n}$$

$$+ \mathcal{O}\Bigg( B_2 \|\mathcal{I}_T^{\frac{1}{2}} \mathcal{I}_S^{-\frac{1}{2}}\|_2^2 \mathsf{Tr}(\mathcal{I}_S^{-1}) (\frac{\log \frac{d}{\delta}}{n})^{1.5} + B_1^2 B_2 \|\mathcal{I}_T^{\frac{1}{2}} \mathcal{I}_S^{-\frac{1}{2}}\|_2^2 \|\mathcal{I}_S^{-1}\|_2^2 \log(\tilde{\kappa}^{-1/2} \alpha_1)(\frac{\log \frac{d}{\delta}}{n})^{2.5}$$

$$+ B_3 \|\mathcal{I}_T^{\frac{1}{2}} \mathcal{I}_S^{-\frac{1}{2}}\|_2^2 \mathsf{Tr}(\mathcal{I}_S^{-1})^{1.5} (\frac{\log \frac{d}{\delta}}{n})^{1.5} + B_1^3 B_3 \|\mathcal{I}_T^{\frac{1}{2}} \mathcal{I}_S^{-\frac{1}{2}}\|_2^2 \|\mathcal{I}_S^{-1}\|_2^3 (\log(\tilde{\kappa}^{-1/2} \alpha_1))^{1.5}(\frac{\log \frac{d}{\delta}}{n})^3$$

$$+ B_2 \|\mathcal{I}_T^{\frac{1}{2}} \mathcal{I}_S^{-\frac{1}{2}}\|_2^2 d^2 (\frac{\log \frac{d}{\delta}}{n})^{1.5} + B_3 \|\mathcal{I}_T^{\frac{1}{2}} \mathcal{I}_S^{-\frac{1}{2}}\|_2^2 d^3 (\frac{\log \frac{d}{\delta}}{n})^{1.5}$$

$$+ B_1^2 \|\mathcal{I}_T^{\frac{1}{2}} \mathcal{I}_S^{-\frac{1}{2}}\|_2^2 \|\mathcal{I}_S^{-1}\|_2 \log(\kappa^{-1/2} \alpha_1)(\frac{\log \frac{d}{\delta}}{n})^2 \Bigg)$$

$$= \frac{\mathsf{Tr}(\mathcal{I}_T \mathcal{I}_S^{-1}) \log \frac{d}{\delta}}{n}$$

$$+ \mathcal{O}\Bigg( d^2 \|\mathcal{I}_T^{\frac{1}{2}} \mathcal{I}_S^{-\frac{1}{2}}\|_2^2 \mathsf{Tr}(\mathcal{I}_S^{-1}) (\frac{\log \frac{d}{\delta}}{n})^{1.5} + d^4 \|\mathcal{I}_T^{\frac{1}{2}} \mathcal{I}_S^{-\frac{1}{2}}\|_2^2 \|\mathcal{I}_S^{-1}\|_2^2 \log(\tilde{\kappa}^{-1/2} \alpha_1)(\frac{\log \frac{d}{\delta}}{n})^{2.5}$$

$$+ (\sqrt{d} + r)^4 \|\mathcal{I}_T^{\frac{1}{2}} \mathcal{I}_S^{-\frac{1}{2}}\|_2^2 \mathsf{Tr}(\mathcal{I}_S^{-1})^{1.5} (\frac{\log \frac{d}{\delta}}{n})^{1.5} + d^3(\sqrt{d} + r)^4 \|\mathcal{I}_T^{\frac{1}{2}} \mathcal{I}_S^{-\frac{1}{2}}\|_2^2 \|\mathcal{I}_S^{-1}\|_2^3 (\log(\tilde{\kappa}^{-1/2} \alpha_1))^{1.5}(\frac{\log \frac{d}{\delta}}{n})^3$$

$$+ d^4 \|\mathcal{I}_T^{\frac{1}{2}} \mathcal{I}_S^{-\frac{1}{2}}\|_2^2 (\frac{\log \frac{d}{\delta}}{n})^{1.5} + d^3(\sqrt{d} + r)^4 \|\mathcal{I}_T^{\frac{1}{2}} \mathcal{I}_S^{-\frac{1}{2}}\|_2^2 (\frac{\log \frac{d}{\delta}}{n})^{1.5}$$

$$+ d^2 \|\mathcal{I}_T^{\frac{1}{2}} \mathcal{I}_S^{-\frac{1}{2}}\|_2^2 \|\mathcal{I}_S^{-1}\|_2 \log(\kappa^{-1/2} \alpha_1)(\frac{\log \frac{d}{\delta}}{n})^2 \Bigg)$$

$$(44)$$

To guarantee $\frac{\mathsf{Tr}(\mathcal{I}_T \mathcal{I}_S^{-1}) \log \frac{d}{\delta}}{n}$ is the leading term, we only need $n \geq \mathcal{O}(N_1 \log \frac{d}{\delta})$, where

$$N_1 := \max \Bigg\{ \left( \frac{d^2 \|\mathcal{I}_T^{\frac{1}{2}} \mathcal{I}_S^{-\frac{1}{2}}\|_2^2 \mathsf{Tr}(\mathcal{I}_S^{-1})}{\mathsf{Tr}(\mathcal{I}_T \mathcal{I}_S^{-1})} \right)^2, \left( \frac{d^4 \|\mathcal{I}_T^{\frac{1}{2}} \mathcal{I}_S^{-\frac{1}{2}}\|_2^2 \|\mathcal{I}_S^{-1}\|_2^2 \log(\tilde{\kappa}^{-1/2} \alpha_1)}{\mathsf{Tr}(\mathcal{I}_T \mathcal{I}_S^{-1})} \right)^{\frac{2}{3}}, \left( \frac{(\sqrt{d} + r)^4 \|\mathcal{I}_T^{\frac{1}{2}} \mathcal{I}_S^{-\frac{1}{2}}\|_2^2 \mathsf{Tr}(\mathcal{I}_S^{-1})^{1.5}}{\mathsf{Tr}(\mathcal{I}_T \mathcal{I}_S^{-1})} \right)^2$$

$$\left( \frac{d^3(\sqrt{d} + r)^4 \|\mathcal{I}_T^{\frac{1}{2}} \mathcal{I}_S^{-\frac{1}{2}}\|_2^2 \|\mathcal{I}_S^{-1}\|_2^3 (\log(\tilde{\kappa}^{-1/2} \alpha_1))^{1.5}}{\mathsf{Tr}(\mathcal{I}_T \mathcal{I}_S^{-1})} \right)^{\frac{1}{2}}, \left( \frac{d^4 \|\mathcal{I}_T^{\frac{1}{2}} \mathcal{I}_S^{-\frac{1}{2}}\|_2^2}{\mathsf{Tr}(\mathcal{I}_T \mathcal{I}_S^{-1})} \right)^2, \left( \frac{d^3(\sqrt{d} + r)^4 \|\mathcal{I}_T^{\frac{1}{2}} \mathcal{I}_S^{-\frac{1}{2}}\|_2^2}{\mathsf{Tr}(\mathcal{I}_T \mathcal{I}_S^{-1})} \right)^2,$$

$$\frac{d^2 \|\mathcal{I}_T^{\frac{1}{2}} \mathcal{I}_S^{-\frac{1}{2}}\|_2^2 \|\mathcal{I}_S^{-1}\|_2 \log(\kappa^{-1/2} \alpha_1)}{\mathsf{Tr}(\mathcal{I}_T \mathcal{I}_S^{-1})} \Bigg\}.$$

That is, for any $\delta$, when $n \geq \mathcal{O}(\max\{d^4, \mathsf{Tr}(\mathcal{I}_S^{-1}), d\|\mathcal{I}_S^{-1}\|_2 \log^{0.5}(\tilde{\kappa}^{-\frac{1}{2}} \alpha_1), N_1\} \log \frac{d}{\delta})$, with probability $1 - 2\delta$,

$$(\beta_{\mathsf{MLE}} - \beta^\star)^T \mathcal{I}_T (\beta_{\mathsf{MLE}} - \beta^\star) \lesssim \frac{\mathsf{Tr}(\mathcal{I}_T \mathcal{I}_S^{-1}) \log \frac{d}{\delta}}{n}.$$

Then following the proof of Theorem 3.1, do Taylor expansion w.r.t. $\beta$ as the following:

$$R_{\beta^\star}(\beta_{\mathsf{MLE}}) = \mathbb{E}_{\substack{x \sim \mathbb{P}_T(X) \\ y|x \sim f(y|x;\beta^\star)}} [\ell(x, y, \beta_{\mathsf{MLE}}) - \ell(x, y, \beta^\star)]$$

$$\leq \mathbb{E}_{\substack{x \sim \mathbb{P}_T(X) \\ y|x \sim f(y|x;\beta^\star)}} [\nabla \ell(x, y, \beta^\star)]^T (\beta_{\mathsf{MLE}} - \beta^\star)$$

$$+ \frac{1}{2}(\beta_{\mathsf{MLE}} - \beta^\star)^T \mathcal{I}_T (\beta_{\mathsf{MLE}} - \beta^\star) + \frac{B_3}{6} \|\beta_{\mathsf{MLE}} - \beta^\star\|_2^3.$$

$$\leq \frac{c}{2} \frac{\mathsf{Tr}(\mathcal{I}_T \mathcal{I}_S^{-1}) \log \frac{d}{\delta}}{n} + \frac{c^3}{6} d^3(\sqrt{d} + r)^4 (\frac{\log \frac{d}{\delta}}{n})^{1.5}.$$

with probability at least $1 - 2\delta$. If we further assume $n \geq \mathcal{O}((\frac{d^3(\sqrt{d}+r)^4}{\mathsf{Tr}(\mathcal{I}_T\mathcal{I}_S^{-1})})^2 \log \frac{d}{\delta})$, it then holds that

$$R_{\beta^*}(\beta_{\mathsf{MLE}}) \leq c \frac{\mathsf{Tr}(\mathcal{I}_T\mathcal{I}_S^{-1}) \log \frac{d}{\delta}}{n}.$$

Therefore we conclude that for any $\delta$, when $n \geq \mathcal{O}(N \log \frac{d}{\delta})$, with probability at least $1 - 2\delta$,

$$R_{\beta^*}(\beta_{\mathsf{MLE}}) \leq c \frac{\mathsf{Tr}(\mathcal{I}_T\mathcal{I}_S^{-1}) \log \frac{d}{\delta}}{n},$$

where

$$N := \max\{d^4, \mathsf{Tr}(\mathcal{I}_S^{-1}), d\|\mathcal{I}_S^{-1}\|_2 \log^{0.5}(\tilde{\kappa}^{-\frac{1}{2}}\alpha_1), N_1, (\frac{d^3(\sqrt{d}+r)^4}{\mathsf{Tr}(\mathcal{I}_T\mathcal{I}_S^{-1})})^2\}$$

$$= \max\left\{ \left(\frac{d^2\|\mathcal{I}_T^{\frac{1}{2}}\mathcal{I}_S^{-\frac{1}{2}}\|_2^2 \mathsf{Tr}(\mathcal{I}_S^{-1})}{\mathsf{Tr}(\mathcal{I}_T\mathcal{I}_S^{-1})}\right)^2, \left(\frac{d^4\|\mathcal{I}_T^{\frac{1}{2}}\mathcal{I}_S^{-\frac{1}{2}}\|_2^2\|\mathcal{I}_S^{-1}\|_2^2 \log(\tilde{\kappa}^{-1/2}\alpha_1)}{\mathsf{Tr}(\mathcal{I}_T\mathcal{I}_S^{-1})}\right)^{\frac{2}{3}}, \left(\frac{(\sqrt{d}+r)^4\|\mathcal{I}_T^{\frac{1}{2}}\mathcal{I}_S^{-\frac{1}{2}}\|_2^2 \mathsf{Tr}(\mathcal{I}_S^{-1})^{1.5}}{\mathsf{Tr}(\mathcal{I}_T\mathcal{I}_S^{-1})}\right)^2, \right.$$

$$\left(\frac{d^3(\sqrt{d}+r)^4\|\mathcal{I}_T^{\frac{1}{2}}\mathcal{I}_S^{-\frac{1}{2}}\|_2^2\|\mathcal{I}_S^{-1}\|_2^3 (\log(\tilde{\kappa}^{-1/2}\alpha_1))^{1.5}}{\mathsf{Tr}(\mathcal{I}_T\mathcal{I}_S^{-1})}\right)^{\frac{1}{2}}, \left(\frac{d^4\|\mathcal{I}_T^{\frac{1}{2}}\mathcal{I}_S^{-\frac{1}{2}}\|_2^2}{\mathsf{Tr}(\mathcal{I}_T\mathcal{I}_S^{-1})}\right)^2, \left(\frac{d^3(\sqrt{d}+r)^4\|\mathcal{I}_T^{\frac{1}{2}}\mathcal{I}_S^{-\frac{1}{2}}\|_2^2}{\mathsf{Tr}(\mathcal{I}_T\mathcal{I}_S^{-1})}\right)^2, $$

$$\left. \frac{d^2\|\mathcal{I}_T^{\frac{1}{2}}\mathcal{I}_S^{-\frac{1}{2}}\|_2^2\|\mathcal{I}_S^{-1}\|_2 \log(\kappa^{-1/2}\alpha_1)}{\mathsf{Tr}(\mathcal{I}_T\mathcal{I}_S^{-1})}, d^4, \mathsf{Tr}(\mathcal{I}_S^{-1}), d\|\mathcal{I}_S^{-1}\|_2 \log^{0.5}(\tilde{\kappa}^{-\frac{1}{2}}\alpha_1), (\frac{d^3(\sqrt{d}+r)^4}{\mathsf{Tr}(\mathcal{I}_T\mathcal{I}_S^{-1})})^2 \right\}.$$

Now it remains to calculate $N$ and $\mathsf{Tr}(\mathcal{I}_T\mathcal{I}_S^{-1})$. Similar to logistic regression (see Lemma B.1 and B.2), we have the following two lemmas that characterize $\mathcal{I}_S$ and $\mathcal{I}_T$.

**Lemma B.4.** *Under the conditions of Theorem 4.5, we have $\mathcal{I}_S = U\mathsf{diag}(\lambda_1, \lambda_2, \dots, \lambda_2)U^T$ and $\mathcal{I}_T = U\mathsf{diag}(\lambda_1, \lambda_2 + r^2\lambda_3, \lambda_2, \dots, \lambda_2)U^T$ for an orthonormal matrix $U$. Where*

$$\lambda_1 := 4\mathbb{E}_{x\sim\mathsf{Uniform}(\mathcal{S}^{d-1}(\sqrt{d}))}[(\beta^{*T}x)^4],$$

$$\lambda_2 := 4\mathbb{E}_{x\sim\mathsf{Uniform}(\mathcal{S}^{d-1}(\sqrt{d}))}[(\beta^{*T}x)^2(\beta_\perp^{*T}x)^2],$$

$$\lambda_3 := 4\mathbb{E}_{x\sim\mathsf{Uniform}(\mathcal{S}^{d-1}(\sqrt{d}))}[(\beta^{*T}x)^2].$$

**Lemma B.5.** *Under the conditions of Theorem 4.5, there exist absolute constants $c, C, c' > 0$ such that $c < \lambda_1, \lambda_2, \lambda_3 < C$, for $d \geq c'$.*

The proofs for these two lemmas are in the next section. With Lemma B.4, we have $\mathcal{I}_T\mathcal{I}_S^{-1} = U\mathsf{diag}(1, 1 + r^2\frac{\lambda_3}{\lambda_2}, \dots, 1)U^T$, $\mathcal{I}_S^{-1} = U\mathsf{diag}(\frac{1}{\lambda_1}, \frac{1}{\lambda_2}, \dots, \frac{1}{\lambda_2})U^T$. By Lemma B.5, since $\lambda_1, \lambda_2, \lambda_3 = O(1)$, we have $\mathsf{Tr}(\mathcal{I}_T\mathcal{I}_S^{-1}) = d + r^2\frac{\lambda_3}{\lambda_2} \asymp d + r^2$, $\|\mathcal{I}_T\mathcal{I}_S^{-1}\|_2 = 1 + r^2\frac{\lambda_3}{\lambda_2} \asymp 1 + r^2$. Similarly $\mathsf{Tr}(\mathcal{I}_S^{-1}) = \lambda_1^{-1} + (d-1)\lambda_2^{-1} \asymp d$, $\|\mathcal{I}_S^{-1}\|_2 = \max\{\lambda_1^{-1}, \lambda_2^{-1}\} \asymp 1$, $\alpha_1 = B_1\|\mathcal{I}_S^{-1}\|_2^{\frac{1}{2}} \asymp d$. Plug in these quantities, recall

$$\kappa := \frac{\mathsf{Tr}(\mathcal{I}_T\mathcal{I}_S^{-1})}{\|\mathcal{I}_T^{\frac{1}{2}}\mathcal{I}_S^{-1}\mathcal{I}_T^{\frac{1}{2}}\|_2} \asymp \frac{d + r^2}{1 + r^2}$$

we have

$$N = \max\left\{ d^6\kappa^{-2}, d^{\frac{8}{3}}\kappa^{-\frac{2}{3}}\log^{\frac{2}{3}}(\tilde{\kappa}^{-1/2}\alpha_1), d^3(\sqrt{d}+r)^8\kappa^{-2}, d^{\frac{3}{2}}(\sqrt{d}+r)^2\kappa^{-\frac{1}{2}}\log^{\frac{3}{4}}(\tilde{\kappa}^{-1/2}\alpha_1), d^8\kappa^{-2}, d^6(\sqrt{d}+r)^8\kappa^{-2}, \right.$$

$$\left. d^2\kappa^{-1}\log(\kappa^{-1/2}\alpha_1), d^4, d, d\log^{\frac{1}{2}}(\tilde{\kappa}^{-1/2}\alpha_1), d^6(\sqrt{d}+r)^8\kappa^{-2}\|\mathcal{I}_T\mathcal{I}_S^{-1}\|^{-2} \right\}$$

$$\overset{1\leq\kappa\leq d}{=} \max\left\{ d^6(\sqrt{d}+r)^8\kappa^{-2}, d^6(\sqrt{d}+r)^8\kappa^{-2}\|\mathcal{I}_T\mathcal{I}_S^{-1}\|^{-2} \right\}$$

$$\overset{\|\mathcal{I}_T\mathcal{I}_S^{-1}\|\asymp 1+r^2\geq 1}{=} d^6(\sqrt{d}+r)^8\kappa^{-2}$$

$$\asymp \frac{d^6(\sqrt{d}+r)^8(1+r^2)^2}{(d+r^2)^2} \asymp d^6(d+r^2)^2(1+r^2)^2$$

We can see that when $r \leq 1$, $N \asymp d^8$. When $1 \leq r \leq \sqrt{d}$, $N \asymp d^8r^4$. When $r \geq \sqrt{d}$, $N \asymp d^6r^8$.

$\square$

### B.3.1   PROOF OF LEMMA B.3

In the following, we prove Lemma B.3. The intuition is that, although $\ell$ is not convex in $\beta$, $\ell$ is quadratic in $M := \beta\beta^T$.

*Proof of Lemma B.3.* With a little bit abuse of notation, for matrix $M \in \mathbb{R}^{d \times d}$, we denote

$$\ell(x, y, M) := \frac{1}{2}(y - \langle xx^T, M \rangle)^2.$$

Under the case where $M = \beta\beta^T$, we have

$$\ell(x, y, M) := \frac{1}{2}(y - \langle xx^T, \beta\beta^T \rangle)^2 = \frac{1}{2}(y - (x^T\beta)^2)^2 = \ell(x, y, \beta).$$

We further denote

$$\ell_n(M) := \frac{1}{n}\sum_{i=1}^n \ell(x_i, y_i, M) = \frac{1}{2n}\sum_{i=1}^n (y_i - \langle x_i x_i^T, M \rangle)^2.$$

and $M^\star := \beta^\star\beta^{\star T}$.

It then holds that

$$\nabla \ell_n(M^\star) = -\frac{1}{n}\sum_{i=1}^n \mathsf{vec}(x_i x_i^T)\varepsilon_i, \quad \nabla^2 \ell_n(M^\star) = \frac{1}{n}\sum_{i=1}^n \mathsf{vec}(x_i x_i^T)\mathsf{vec}(x_i x_i^T)^T, \quad \nabla^3 \ell_n(M) = 0.$$

Denote $\Sigma_S := \mathbb{E}_{x \sim \mathbb{P}_S(X)}[\mathsf{vec}(xx^T)\mathsf{vec}(xx^T)^T]$, then by Lemma D.1 with $V = \mathsf{Tr}(\Sigma_S)$, $\alpha = 2$, $B_u^\alpha = cd$ for some absolute constants $c, c'$, we have with probability at least $1 - \delta$,

$$\|\nabla \ell_n(M^\star)\|_2 \leq c'\left(\sqrt{\frac{\mathsf{Tr}(\Sigma_S)\log\frac{d}{\delta}}{n}} + d(\log\frac{c^2 d^2}{\mathsf{Tr}(\Sigma_S)})^{\frac{1}{2}}\frac{\log\frac{d}{\delta}}{n}\right). \tag{45}$$

By matrix Hoeffding, we have with probability at least $1 - \delta$,

$$\Sigma_S - d^2\sqrt{\frac{8\log\frac{d}{\delta}}{n}}I_d \preceq \nabla^2 \ell_n(M^\star) \preceq \Sigma_S + d^2\sqrt{\frac{8\log\frac{d}{\delta}}{n}}I_d. \tag{46}$$

Before conducting further analysis, we need some characterizations of $\Sigma_S$. By the definition of $\Sigma_S$, we can see that the $((i, j), (k, l))$ entry of $\Sigma_S$ is $\mathbb{E}_{X \sim \mathbb{P}_S(X)}[X_i X_j X_k X_l]$. Since $X$ is symmetric and isotropic, we have

$$\mathbb{E}_{X \sim \mathbb{P}_S(X)}[X_i X_j X_k X_l] = \begin{cases} \mathbb{E}_{X \sim \mathbb{P}_S(X)}[X_i^2 X_k^2] & \text{if } i = j, k = l \text{ and } i \neq k \\ \mathbb{E}_{X \sim \mathbb{P}_S(X)}[X_i^2 X_j^2] & \text{if } \{i, j\} = \{k, l\} \text{ and } i \neq j \\ \mathbb{E}_{X \sim \mathbb{P}_S(X)}[X_i^4] & \text{if } i = j = k = l \\ 0 & \text{Otherwise} \end{cases}$$

For the calculation of moments, using (3a) in Cao (2020) with $a = (1, 0, \cdots, 0)^T$ and $\epsilon = \frac{1}{\sqrt{d}}X$, we have $\mathbb{E}_{X \sim \mathbb{P}_S(X)}[X_1^4] = \frac{3d}{d+2}$, $\mathbb{E}_{X \sim \mathbb{P}_S(X)}[X_1^2 X_2^2] = \frac{d}{d+2}$. Since $X$ is isotropic, we have

$$(\Sigma_S)_{((i,j),(k,l))} = \begin{cases} \frac{d}{d+2} & \text{if } i = j, k = l \text{ and } i \neq k \\ \frac{d}{d+2} & \text{if } \{i, j\} = \{k, l\} \text{ and } i \neq j \\ \frac{3d}{d+2} & \text{if } i = j = k = l \\ 0 & \text{Otherwise} \end{cases} \tag{47}$$

Therefore

$$\mathsf{Tr}(\Sigma_S) = \sum_{i,j}\mathbb{E}[X_i^2 X_j^2] = d(d-1)\frac{d}{d+2} + d\frac{3d}{d+2} = d^2. \tag{48}$$

The following lemma characterizes the "minimum eigenvalue" of $\Sigma_S$ on a special subspace, which will be useful in our analysis.

**Lemma B.6.** *For any vector $a = (a_{ij})_{(i,j)\in[d]\times[d]} \in \mathbb{R}^{d^2}$ satisfies $a_{ij} = a_{ji}$,*

$$a^T \Sigma_S a \geq \frac{2d}{d+2}\|a\|_2^2.$$

*Proof.*

$$
\begin{aligned}
a^T \Sigma_S a &= \sum_{i,j,k,l} a_{ij}a_{kl}(\Sigma_S)_{((i,j),(k,l))} \\
&\overset{\text{by (47)}}{=} \frac{d}{d+2}\Big(\sum_{i\neq j} a_{ij}^2 + \sum_{i\neq j} a_{ij}a_{ji} + \sum_{i\neq j} a_{ii}a_{jj} + 3\sum_i a_{ii}^2\Big) \\
&\overset{a_{ij}=a_{ji}}{=} \frac{d}{d+2}\Big(2\sum_{i\neq j} a_{ij}^2 + \sum_{i\neq j} a_{ii}a_{jj} + 3\sum_i a_{ii}^2\Big) \\
&= \frac{d}{d+2}\Big(2(\sum_{i\neq j} a_{ij}^2 + \sum_i a_{ii}^2) + (\sum_{i\neq j} a_{ii}a_{jj} + \sum_i a_{ii}^2)\Big) \\
&= \frac{d}{d+2}\Big(2\|a\|_2^2 + (\sum_i a_{ii})^2\Big) \\
&\geq \frac{2d}{d+2}\|a\|_2^2.
\end{aligned}
$$

$\square$

With Lemma B.6 and (48), we are now able to prove Lemma B.3. By Taylor expansion, we have for $M = \beta\beta^T$, $M^\star = \beta^\star\beta^{\star T}$, with probability at least $1 - \delta$,

$$
\begin{aligned}
\ell_n(M) - \ell_n(M^\star) &\overset{\nabla^3 \ell_n \equiv 0}{=} \mathsf{vec}(M - M^\star)^T \nabla\ell_n(M^\star) + \frac{1}{2}\mathsf{vec}(M - M^\star)^T \nabla^2\ell_n(M^\star)\mathsf{vec}(M - M^\star) \\
&\overset{\text{by (45),(46)}}{\geq} -c'\|M - M^\star\|_F \left(\sqrt{\frac{\mathsf{Tr}(\Sigma_S)\log\frac{d}{\delta}}{n}} + d(\log\frac{c^2 d^2}{\mathsf{Tr}(\Sigma_S)})^{\frac{1}{2}}\frac{\log\frac{d}{\delta}}{n}\right) \\
&\quad + \frac{1}{2}\mathsf{vec}(M - M^\star)^T \Sigma_S \mathsf{vec}(M - M^\star) - \|M - M^\star\|_F^2 d^2 \sqrt{\frac{8\log\frac{d}{\delta}}{n}} \\
&\overset{\text{by Lemma B.6 and (48)}}{\geq} \left(\frac{d}{d+2} - d^2\sqrt{\frac{8\log\frac{d}{\delta}}{n}}\right)\|M - M^\star\|_F^2 - c''\left(\sqrt{\frac{d^2\log\frac{d}{\delta}}{n}} + d\frac{\log\frac{d}{\delta}}{n}\right)\|M - M^\star\|_F \\
&\geq \frac{1}{2}\|M - M^\star\|_F^2 - c''\left(\sqrt{\frac{d^2\log\frac{d}{\delta}}{n}} + d\frac{\log\frac{d}{\delta}}{n}\right)\|M - M^\star\|_F
\end{aligned}
$$

when $n \geq \mathcal{O}(d^4\log\frac{d}{\delta})$.

We denote $M_{\mathsf{MLE}} := \beta_{\mathsf{MLE}}\beta_{\mathsf{MLE}}^T$. Note that $\ell_n(M_{\mathsf{MLE}}) - \ell_n(M^\star) = \ell_n(\beta_{\mathsf{MLE}}) - \ell_n(\beta^\star) \leq 0$. Thus we have

$$\frac{1}{2}\|M_{\mathsf{MLE}} - M^\star\|_F^2 - c''\left(\sqrt{\frac{d^2\log\frac{d}{\delta}}{n}} + d\frac{\log\frac{d}{\delta}}{n}\right)\|M_{\mathsf{MLE}} - M^\star\|_F \leq 0,$$

which implies

$$\|M_{\mathsf{MLE}} - M^\star\|_F \lesssim \left(\sqrt{\frac{d^2\log\frac{d}{\delta}}{n}} + d\frac{\log\frac{d}{\delta}}{n}\right) \lesssim \sqrt{\frac{d^2\log\frac{d}{\delta}}{n}}.$$

Thus so far we have shown, if $n \geq \mathcal{O}(d^4 \log \frac{d}{\delta})$, then with probability at least $1 - \delta$, we have

$$\|M_{\mathsf{MLE}} - M^\star\|_F \lesssim \sqrt{\frac{d^2 \log \frac{d}{\delta}}{n}}.$$

By Lemma 6 in Ge et al. (2017), we further have

$$\min\{\|\beta_{\mathsf{MLE}} - \beta^\star\|_2, \|\beta_{\mathsf{MLE}} + \beta^\star\|_2\} \lesssim \frac{1}{\|\beta^\star\|_2} \|M_{\mathsf{MLE}} - M^\star\|_F \lesssim \sqrt{\frac{d^2 \log \frac{d}{\delta}}{n}}.$$

$\square$

### B.3.2 Proofs for Lemma B.4 and B.5

The proofs for Lemma B.4 and B.5 are similar to proofs for Lemma B.1 and B.2.

*Proof of Lemma B.4.* By definition,

$$\mathcal{I}_S := 4\mathbb{E}_{x \sim \mathsf{Uniform}(\mathcal{S}^{d-1}(\sqrt{d}))}[xx^T(x^T\beta^\star)^2]$$

Let $z \sim \mathcal{N}(0, I_d)$, then $x$ and $z\frac{\sqrt{d}}{\|z\|_2}$ have the same distribution. Therefore

$$\mathcal{I}_S = 4\mathbb{E}_{x \sim \mathsf{Uniform}(\mathcal{S}^{d-1}(\sqrt{d}))}[xx^T(x^T\beta^\star)^2]$$

$$= 4\mathbb{E}_{z \sim \mathcal{N}(0, I_d)}[zz^T\frac{d}{\|z\|_2^2}(\beta^{\star T}z \cdot \frac{\sqrt{d}}{\|z\|_2})^2]$$

$$= 4\mathbb{E}_{z \sim \mathcal{N}(0, I_d)}[(\beta^\star\beta^{\star T} + U_\perp U_\perp^T)zz^T(\beta^{\star T}z)^2\frac{d^2}{\|z\|_2^4}]$$

where $[\beta^\star, U_\perp] \in \mathbb{R}^{d \times d}$ is a orthogonal basis.

With this expression, we first prove $\beta^\star$ is an eigenvector of $\mathcal{I}_S$ with corresponding eigenvalue $\lambda_1$.

$$\mathcal{I}_S\beta^\star = 4\mathbb{E}_{z \sim \mathcal{N}(0, I_d)}[(\beta^\star\beta^{\star T} + U_\perp U_\perp^T)zz^T(\beta^{\star T}z)^2\frac{d^2}{\|z\|_2^4}]\beta^\star$$

$$= 4\mathbb{E}_{z \sim \mathcal{N}(0, I_d)}[\beta^\star\beta^{\star T}zz^T(\beta^{\star T}z)^2\frac{d^2}{\|z\|_2^4}\beta^\star]$$

$$+ 4\mathbb{E}_{z \sim \mathcal{N}(0, I_d)}[U_\perp U_\perp^Tzz^T(\beta^{\star T}z)^2\frac{d^2}{\|z\|_2^4}\beta^\star]$$

$$= 4\mathbb{E}_{z \sim \mathcal{N}(0, I_d)}[(\beta^{\star T}z)^4\frac{d^2}{\|z\|_2^4}]\beta^\star$$

$$+ 4\mathbb{E}_{z \sim \mathcal{N}(0, I_d)}[U_\perp U_\perp^Tzz^T(\beta^{\star T}z)^2\frac{d^2}{\|z\|_2^4}\beta^\star]$$

$$= \lambda_1\beta^\star + 4\mathbb{E}_{z \sim \mathcal{N}(0, I_d)}[U_\perp U_\perp^Tzz^T(\beta^{\star T}z)^2\frac{d^2}{\|z\|_2^4}\beta^\star].$$

Therefore we only need to prove

$$\mathbb{E}_{z \sim \mathcal{N}(0, I_d)}[U_\perp U_\perp^Tzz^T(\beta^{\star T}z)^2\frac{d^2}{\|z\|_2^4}\beta^\star] = 0.$$

In fact,

$$\mathbb{E}_{z \sim \mathcal{N}(0, I_d)}[U_\perp^Tzz^T(\beta^{\star T}z)^2\frac{d^2}{\|z\|_2^4}\beta^\star]$$

$$= \mathbb{E}_{z \sim \mathcal{N}(0, I_d)}[(U_\perp^Tz)(\beta^{\star T}z)^3\frac{d^2}{\|z\|_2^4}]$$

$$= \mathbb{E}_{z \sim \mathcal{N}(0, I_d)}[(\frac{d}{|A|^2 + \|B\|^2})^2 A^3 B]$$

where we let $A := z^T \beta^\star$, $B := U_\perp^T z$. Notice that by the property of $z \sim \mathcal{N}(0, I_d)$, $A$ and $B$ are independent. Also, $B$ is symmetric, i.e., $B$ and $-B$ have the same distribution. Therefore

$$\mathbb{E}_{z\sim\mathcal{N}(0,I_d)}[U_\perp U_\perp^T zz^T (\beta^{\star T} z)^2 \frac{d^2}{\|z\|_2^4} \beta^\star] = \mathbb{E}_{z\sim\mathcal{N}(0,I_d)}[(\frac{d}{|A|^2 + \|B\|^2})^2 A^3 B] = 0.$$

Next we will prove that for any $\beta_\perp$ such that $\|\beta_\perp\|_2 = 1$, $\beta^{\star T}\beta_\perp = 0$, $\beta_\perp$ is an eigenvector of $\mathcal{I}_S$ with corresponding eigenvalue $\lambda_2$. Let $[\beta_\perp, U]$ be an orthogonal basis ($\beta^\star$ is the first column of $U$).

$$\begin{aligned}
\mathcal{I}_S \beta_\perp &= 4\mathbb{E}_{z\sim\mathcal{N}(0,I_d)}[(\beta_\perp \beta_\perp^T + UU^T)zz^T (\beta^{\star T} z)^2 \frac{d^2}{\|z\|_2^4}]\beta_\perp \\
&= 4\mathbb{E}_{z\sim\mathcal{N}(0,I_d)}[\beta_\perp \beta_\perp^T zz^T (\beta^{\star T} z)^2 \frac{d^2}{\|z\|_2^4}\beta_\perp] \\
&\quad + 4\mathbb{E}_{z\sim\mathcal{N}(0,I_d)}[UU^T zz^T (\beta^{\star T} z)^2 \frac{d^2}{\|z\|_2^4}\beta_\perp] \\
&= 4\mathbb{E}_{z\sim\mathcal{N}(0,I_d)}[(\beta_\perp^T z)^2 (\beta^{\star T} z)^2 \frac{d^2}{\|z\|_2^4}]\beta_\perp \\
&\quad + 4\mathbb{E}_{z\sim\mathcal{N}(0,I_d)}[UU^T zz^T (\beta^{\star T} z)^2 \frac{d^2}{\|z\|_2^4}\beta_\perp] \\
&= \lambda_2 \beta_\perp + 0 \\
&= \lambda_2 \beta_\perp
\end{aligned}$$

Here

$$4\mathbb{E}_{z\sim\mathcal{N}(0,I_d)}[UU^T zz^T (\beta^{\star T} z)^2 \frac{d^2}{\|z\|_2^4}\beta_\perp] = 0$$

because of a similar reason as in the previous part.

For $\mathcal{I}_T$, the proving strategy is similar. For $x \sim \mathsf{Uniform}(\mathcal{S}^{d-1}(\sqrt{d})) + v$ on the target domain, where $v = r\beta_\perp^\star$, let $w = x - v = x - r\beta_\perp^\star$, then $w \sim \mathsf{Uniform}(\mathcal{S}^{d-1}(\sqrt{d}))$. Let $z \sim \mathcal{N}(0, I_d)$, then $w$ and $z\frac{\sqrt{d}}{\|z\|_2}$ have the same distribution. We have

$$\begin{aligned}
\mathcal{I}_T &= 4\mathbb{E}_{x\sim\mathsf{Uniform}(\mathcal{S}^{d-1}(\sqrt{d}))+v}[xx^T (x^T \beta^\star)^2] \\
&= 4\mathbb{E}_{w\sim\mathsf{Uniform}(\mathcal{S}^{d-1}(\sqrt{d}))}[(w + v)(w + v)^T ((w + v)^T \beta^\star)^2] \\
&\overset{v^T\beta^\star=0}{=} 4\mathbb{E}_{w\sim\mathsf{Uniform}(\mathcal{S}^{d-1}(\sqrt{d}))}[(ww^T + wv^T + vw^T + vv^T)(w^T \beta^\star)^2]
\end{aligned}$$

Therefore

$$\begin{aligned}
\mathcal{I}_T \beta^\star &= 4\mathbb{E}_{w\sim\mathsf{Uniform}(\mathcal{S}^{d-1}(\sqrt{d}))}[(ww^T + wv^T + vw^T + vv^T)(w^T \beta^\star)^2]\beta^\star \\
&\overset{v^T\beta^\star=0}{=} 4\mathbb{E}_{w\sim\mathsf{Uniform}(\mathcal{S}^{d-1}(\sqrt{d}))}[ww^T (w^T \beta^\star)^2]\beta^\star \\
&= \mathcal{I}_S \beta^\star \\
&= \lambda_1 \beta^\star,
\end{aligned}$$

where the last line follows from the previous proofs. Similarly, for any $\tilde{\beta}_\perp$ such that $\|\tilde{\beta}_\perp\|_2 = 1$, $\beta_\perp^{\star T}\tilde{\beta}_\perp = 0$,

$$\begin{aligned}
\mathcal{I}_T \tilde{\beta}_\perp &= 4\mathbb{E}_{w\sim\mathsf{Uniform}(\mathcal{S}^{d-1}(\sqrt{d}))}[(ww^T + wv^T + vw^T + vv^T)(w^T \beta^\star)^2]\tilde{\beta}_\perp \\
&\overset{v^T\tilde{\beta}_\perp=0}{=} 4\mathbb{E}_{w\sim\mathsf{Uniform}(\mathcal{S}^{d-1}(\sqrt{d}))}[ww^T (w^T \beta^\star)^2]\tilde{\beta}_\perp \\
&= \mathcal{I}_S \tilde{\beta}_\perp \\
&= \lambda_2 \tilde{\beta}_\perp.
\end{aligned}$$

For $\beta_\perp^\star$,

$$\mathcal{I}_T \beta_\perp^\star = 4\mathbb{E}_{w\sim\mathsf{Uniform}(\mathcal{S}^{d-1}(\sqrt{d}))}[(ww^T + wv^T + vw^T + vv^T)(w^T\beta^\star)^2]\beta_\perp^\star$$

$$= 4\mathbb{E}_{w\sim\mathsf{Uniform}(\mathcal{S}^{d-1}(\sqrt{d}))}[ww^T(w^T\beta^\star)^2]\beta_\perp^\star + 4\mathbb{E}_{w\sim\mathsf{Uniform}(\mathcal{S}^{d-1}(\sqrt{d}))}[wv^T(w^T\beta^\star)^2]\beta_\perp^\star$$

$$+ 4\mathbb{E}_{w\sim\mathsf{Uniform}(\mathcal{S}^{d-1}(\sqrt{d}))}[vw^T(w^T\beta^\star)^2]\beta_\perp^\star + 4\mathbb{E}_{w\sim\mathsf{Uniform}(\mathcal{S}^{d-1}(\sqrt{d}))}[vv^T(w^T\beta^\star)^2]\beta_\perp^\star$$

$$:= I_1 + I_2 + I_3 + I_4.$$

As in the previous proofs,

$$I_1 = \mathcal{I}_S \beta_\perp^\star = \lambda_2 \beta_\perp^\star.$$

$$I_2 = 4\mathbb{E}_{w\sim\mathsf{Uniform}(\mathcal{S}^{d-1}(\sqrt{d}))}[wv^T(w^T\beta^\star)^2]\beta_\perp^\star$$

$$\overset{v=r\beta_\perp^\star}{=} 4r\mathbb{E}_{w\sim\mathsf{Uniform}(\mathcal{S}^{d-1}(\sqrt{d}))}[w(\beta_\perp^{\star T}\beta_\perp^\star)(w^T\beta^\star)^2]$$

$$\overset{\|\beta_\perp^\star\|=1}{=} 4r\mathbb{E}_{w\sim\mathsf{Uniform}(\mathcal{S}^{d-1}(\sqrt{d}))}[w(w^T\beta^\star)^2]$$

$$= 0.$$

where the last lines follows from $w$ is symmetric and $w(w^T\beta^\star)^2$ is a odd function of $w$.

$$I_3 = 4\mathbb{E}_{w\sim\mathsf{Uniform}(\mathcal{S}^{d-1}(\sqrt{d}))}[vw^T(w^T\beta^\star)^2]\beta_\perp^\star$$

$$\overset{v=r\beta_\perp^\star}{=} 4r\mathbb{E}_{w\sim\mathsf{Uniform}(\mathcal{S}^{d-1}(\sqrt{d}))}[\beta_\perp^\star w^T\beta_\perp^\star(w^T\beta^\star)^2]$$

$$= 4r\mathbb{E}_{w\sim\mathsf{Uniform}(\mathcal{S}^{d-1}(\sqrt{d}))}[(w^T\beta_\perp^\star)(w^T\beta^\star)^2]\beta_\perp^\star$$

$$= 0.$$

where the last lines follows from $w$ is symmetric and $(w^T\beta_\perp^\star)(w^T\beta^\star)^2$ is a odd function of $w$.

$$I_4 = 4\mathbb{E}_{w\sim\mathsf{Uniform}(\mathcal{S}^{d-1}(\sqrt{d}))}[vv^T(w^T\beta^\star)^2]\beta_\perp^\star$$

$$\overset{v=r\beta_\perp^\star}{=} 4r^2\mathbb{E}_{w\sim\mathsf{Uniform}(\mathcal{S}^{d-1}(\sqrt{d}))}[\beta_\perp^\star\beta_\perp^{\star T}\beta_\perp^\star(w^T\beta^\star)^2]$$

$$\overset{\|\beta_\perp^\star\|=1}{=} 4r^2\mathbb{E}_{w\sim\mathsf{Uniform}(\mathcal{S}^{d-1}(\sqrt{d}))}[\beta_\perp^\star(w^T\beta^\star)^2]$$

$$= r^2\lambda_3\beta_\perp^\star.$$

Combine the calculations of $I_1, I_2, I_3, I_4$, we have

$$\mathcal{I}_T\beta_\perp^\star = I_1 + I_2 + I_3 + I_4$$

$$= \lambda_2\beta_\perp^\star + r^2\lambda_3\beta_\perp^\star$$

$$= (\lambda_2 + r^2\lambda_3)\beta_\perp^\star.$$

In conclusion, we have $\mathcal{I}_S = U\mathsf{diag}(\lambda_1, \lambda_2, \ldots, \lambda_2)U^T$ and $\mathcal{I}_T = U\mathsf{diag}(\lambda_1, \lambda_2 + r^2\lambda_3, \lambda_2, \ldots, \lambda_2)U^T$ for an orthonormal matrix $U$, where $U = [\beta^\star, \beta_\perp^\star, \cdots]$. $\square$

*Proof of Lemma B.5.* Recall the definition of $\lambda_1, \lambda_2, \lambda_3$:

$$\lambda_1 := 4\mathbb{E}_{x\sim\mathsf{Uniform}(\mathcal{S}^{d-1}(\sqrt{d}))}[(\beta^{\star T}x)^4] = 4\mathbb{E}_{z\sim\mathcal{N}(0,I_d)}[(\beta^{\star T}z)^4 \frac{d^2}{\|z\|_2^4}],$$

$$\lambda_2 := 4\mathbb{E}_{x\sim\mathsf{Uniform}(\mathcal{S}^{d-1}(\sqrt{d}))}[(\beta^{\star T}x)^2(\beta_\perp^{\star T}x)^2] = 4\mathbb{E}_{z\sim\mathcal{N}(0,I_d)}[(\beta^{\star T}z)^2(\beta_\perp^{\star T}z)^2 \frac{d^2}{\|z\|_2^4}],$$

$$\lambda_3 := 4\mathbb{E}_{x\sim\mathsf{Uniform}(\mathcal{S}^{d-1}(\sqrt{d}))}[(\beta^{\star T}x)^2] = 4\mathbb{E}_{z\sim\mathcal{N}(0,I_d)}[(\beta^{\star T}z)^2 \frac{d}{\|z\|_2^2}].$$

Next we will show that there exists constants $c, C, c' > 0$ such that when $d \geq c'$, we have $c \leq \lambda_1 \leq C$. The proofs for $\lambda_2$ and $\lambda_3$ are similar. [35] With this concentration, we do the following truncation:

$$\frac{1}{4}\lambda_1 = \mathbb{E}_{z\sim\mathcal{N}(0,I_d)}[(\beta^{\star T}z)^4 \frac{d^2}{\|z\|_2^4}]$$

$$= \mathbb{E}_{z\sim\mathcal{N}(0,I_d)}[(\beta^{\star T}z)^4 \frac{d^2}{\|z\|_2^4}\mathbb{I}_{\frac{\|z\|}{\sqrt{d}}\in[\frac{1}{2},\frac{3}{2}]}] + \mathbb{E}_{z\sim\mathcal{N}(0,I_d)}[(\beta^{\star T}z)^4 \frac{d^2}{\|z\|_2^4}\mathbb{I}_{\frac{\|z\|}{\sqrt{d}}\notin[\frac{1}{2},\frac{3}{2}]}]$$

$$:= J_1 + J_2.$$

For $J_2$, it is obvious that

$$0 \leq J_2 \leq d^2 \mathbb{P}(\frac{\|z\|}{\sqrt{d}} \notin [\frac{1}{2}, \frac{3}{2}]) \leq 2d^2 e^{-cd}. \tag{49}$$

For upper bound of $J_1$,

$$J_1 = \mathbb{E}_{z \sim \mathcal{N}(0, I_d)}[(\beta^{\star T} z)^4 \frac{d^2}{\|z\|_2^4} \mathbb{I}_{\frac{\|z\|}{\sqrt{d}} \in [\frac{1}{2}, \frac{3}{2}]}]$$

$$\leq \mathbb{E}_{z \sim \mathcal{N}(0, I_d)}[16(\beta^{\star T} z)^4] = 48.$$

Therefore

$$\frac{1}{4} \lambda_1 = J_1 + J_2 \leq 48 + 2d^2 e^{-cd}.$$

It's obvious that there exists an absolute constant $c'$ such that when $d \geq c'$, $\frac{1}{4}\lambda_1 \leq 50$.

For lower bound of $J_1$, we have

$$J_1 = \mathbb{E}_{z \sim \mathcal{N}(0, I_d)}[(\beta^{\star T} z)^4 \frac{d^2}{\|z\|_2^4} \mathbb{I}_{\frac{\|z\|}{\sqrt{d}} \in [\frac{1}{2}, \frac{3}{2}]}]$$

$$\geq \mathbb{E}_{z \sim \mathcal{N}(0, I_d)}[(\frac{2}{3})^4 (\beta^{\star T} z)^4] = (\frac{2}{3})^4 \cdot 3.$$

Therefore

$$\frac{1}{4} \lambda_1 = J_1 + J_2 \geq (\frac{2}{3})^4 \cdot 3$$

Therefore it's obvious that there exists an absolute constant $c'$ such that when $d \geq c'$, $\frac{1}{4}\lambda_1 \geq \frac{1}{2}$. The proofs for $\lambda_2$ and $\lambda_3$ are almost the same.

$\square$

## C  PROOFS FOR SECTION 5

### C.1  POOFS FOR PROPOSITION 5.1

*Proof.* We consider the case where $Y = X^2 + \varepsilon$, $\varepsilon \sim \mathcal{N}(0, 1)$, $\varepsilon \perp\!\!\!\perp X$, and we have $X \sim \mathcal{N}(-10, 1)$ on the source domain and $X \sim \mathcal{N}(10, 1)$ on the target domain. Then the optimal linear fit on the target is given by

$$\beta^{\star} = \arg\min_{\beta \in \mathbb{R}} \mathbb{E}_{(x,y) \sim \mathbb{P}_T(X,Y)} \left[ (y - x\beta)^2 \right] = \left( \mathbb{E}_{x \sim \mathcal{N}(10,1)}[x^2] \right)^{-1} \mathbb{E}_{x \sim \mathcal{N}(10,1)}[x^3] > 0.$$

However, the linear fit learned via classical MLE asymptotically behaves as

$$\beta_{\text{MLE}} = \arg\min_{\beta \in \mathbb{R}} \frac{1}{2n} \sum_{i=1}^{n} (y_i - x_i\beta)^2 = \left( \frac{1}{n} \sum_{i=1}^{n} x_i^2 \right)^{-1} \left( \frac{1}{n} \sum_{i=1}^{n} x_i y_i \right)$$

$$\xrightarrow{n \to \infty} \left( \mathbb{E}_{x \sim \mathcal{N}(-10,1)}[x^2] \right)^{-1} \mathbb{E}_{x \sim \mathcal{N}(-10,1)}[x^3] < 0.$$

Hence, the classical MLE losses consistency. For MWLE, we have

$$\beta_{\text{MWLE}} = \arg\min_{\beta \in \mathbb{R}} \frac{1}{2n} \sum_{i=1}^{n} w(x_i)(y_i - x_i\beta)^2$$

$$= \left( \frac{1}{n} \sum_{i=1}^{n} w(x_i) x_i^2 \right)^{-1} \left( \frac{1}{n} \sum_{i=1}^{n} w(x_i) x_i y_i \right) \xrightarrow{n \to \infty} \beta^{\star},$$

which asymptotically provides a good estimator.

$\square$

## C.2 PROOFS FOR THEOREM 5.2

The detailed version of Theorem 5.2 is stated as the following.

**Theorem C.1.** *Suppose the function class $\mathcal{F}$ satisfies Assumption C. Let $G_w := G_w(M)$ and $H_w := H_w(M)$. For any $\delta \in (0,1)$, if $n \geq c \max\{N^\star \log(d/\delta), N(\delta), N'(\delta)\}$, then with probability at least $1 - 3\delta$, we have*

$$R_M(\beta_{\mathsf{MWLE}}) \leq c \frac{\mathsf{Tr}\left(G_w H_w^{-1}\right) \log \frac{d}{\delta}}{n}$$

*for an absolute constant $c$. Here*

$$N^\star := W^2 \cdot \max\{\lambda^{-1}\tilde{\alpha}_1^2 \log^{2\gamma}(W^2 \lambda^{-1}\tilde{\alpha}_1^2), \tilde{\alpha}_2^2, \lambda\tilde{\alpha}_3^2\},$$

*where $\tilde{\alpha}_1 := B_1\|H_w^{-1}\|_2^{0.5}$, $\tilde{\alpha}_2 := B_2\|H_w^{-1}\|_2$, $\tilde{\alpha}_3 := B_3\|H_w^{-1}\|_2^{1.5}$, and $\lambda := \mathsf{Tr}(G_w H_w^{-2})/\|H_w^{-1}\|_2$.*

The proofs for Theorem C.1 is similar to proofs for Theorem A.1. For notation simplicity, through out the proofs for Theorem C.1, let $\beta^\star := \beta^\star(M)$, $H_w := H_w(M)$, $G_w := G_w(M)$. We first state two main lemmas, which capture the distance between $\beta_{\mathsf{MWLE}}$ and $\beta^\star$ under different measurements.

**Lemma C.2.** *Suppose Assumption C holds. For any $\delta \in (0,1)$ and any $n \geq c \max\{N_1 \log(d/\delta), N(\delta), N'(\delta)\}$, with probability at least $1 - 2\delta$, we have $\beta_{\mathsf{MWLE}} \in \mathbb{B}_{\beta^\star}(c\sqrt{\frac{\mathsf{Tr}(G_w H_w^{-2}) \log \frac{d}{\delta}}{n}})$ for some absolute constant $c$. Here*

$$N_1 := \max\left\{W^2 B_2^2 \|H_w^{-1}\|_2^2, W^2 B_3^2 \mathsf{Tr}(G_w H_w^{-2})\|H_w^{-1}\|_2^2, \left(\frac{W^3 B_1^2 B_2 \|H_w^{-1}\|_2^3 \log^{2\gamma}(W\lambda^{-1/2}\tilde{\alpha}_1)}{\mathsf{Tr}(G_w H_w^{-2})}\right)^{\frac{2}{3}}, \right.$$
$$\left.\left(\frac{W^4 B_1^3 B_3 \|H_w^{-1}\|_2^4 \log^{3\gamma}(W\lambda^{-1/2}\tilde{\alpha}_1)}{\mathsf{Tr}(G_w H_w^{-2})}\right)^{\frac{1}{2}}, \frac{W^2 B_1^2 \|H_w^{-1}\|_2^2 \log^{2\gamma}(W\lambda^{-1/2}\tilde{\alpha}_1)}{\mathsf{Tr}(G_w H_w^{-2})}\right\}.$$

**Lemma C.3.** *Suppose Assumption C holds. For any $\delta \in (0,1)$ and any $n \geq c \max\{N_1 \log(d/\delta), N_2 \log(d/\delta), N(\delta), N'(\delta)\}$, with probability at least $1 - 3\delta$, we have*

$$\|H_w^{\frac{1}{2}}(\beta_{\mathsf{MWLE}} - \beta^\star)\|_2^2 \leq c \frac{\mathsf{Tr}(G_w H_w^{-1}) \log \frac{d}{\delta}}{n}.$$

*for some absolute constant $c$. Here $N_1$ is defined in Lemma C.2 and*

$$N_2 := \max\left\{\left(\frac{W B_2 \mathsf{Tr}(G_w H_w^{-2})}{\mathsf{Tr}(G_w H_w^{-1})}\right)^2, \left(\frac{W B_3 \mathsf{Tr}(G_w H_w^{-2})^{1.5}}{\mathsf{Tr}(G_w H_w^{-1})}\right)^2, \left(\frac{W^3 B_1^2 B_2 \|H_w^{-1}\|_2^2 \log^{2\gamma}(W\lambda^{-1/2}\tilde{\alpha}_1)}{\mathsf{Tr}(G_w H_w^{-1})}\right)^{\frac{2}{3}}, \right.$$
$$\left.\left(\frac{W^4 B_1^3 B_3 \|H_w^{-1}\|_2^3 \log^{3\gamma}(W\lambda^{-1/2}\tilde{\alpha}_1)}{\mathsf{Tr}(G_w H_w^{-1})}\right)^{\frac{1}{2}}, \frac{W^2 B_1^2 \|H_w^{-1}\|_2 \log^{2\gamma}(W\lambda^{-1/2}\tilde{\alpha}_1)}{\mathsf{Tr}(G_w H_w^{-1})}\right\}.$$

The proofs for Lemma C.2 and C.3 are delayed to the end of this subsection. With these two lemmas, we can now state the proof for Theorem C.1.

*Proof of Theorem C.1.* By Assumption C.1 and C.3, we can do Taylor expansion w.r.t. $\beta$ as the following:

$$R_M(\beta_{\mathsf{MWLE}}) = \mathbb{E}_{(x,y)\sim\mathbb{P}_T(x,y)}\left[\ell(x,y,\beta_{\mathsf{MWLE}}) - \ell(x,y,\beta^\star)\right]$$
$$\leq \mathbb{E}_{(x,y)\sim\mathbb{P}_T(x,y)}[\nabla\ell(x,y,\beta^\star)]^T(\beta_{\mathsf{MWLE}} - \beta^\star)$$
$$+ \frac{1}{2}(\beta_{\mathsf{MWLE}} - \beta^\star)^T H_w(\beta_{\mathsf{MWLE}} - \beta^\star) + \frac{W B_3}{6}\|\beta_{\mathsf{MWLE}} - \beta^\star\|_2^3.$$

Applying Lemma C.2 and C.3, we know for any $\delta$ and any $n \geq c \max\{N_1 \log(d/\delta), N_2 \log(d/\delta), N(\delta), N'(\delta)\}$, with probability at least $1 - 3\delta$, we have

$$(\beta_{\mathsf{MWLE}} - \beta^\star)^T H_w(\beta_{\mathsf{MWLE}} - \beta^\star) \leq c \frac{\mathsf{Tr}(G_w H_w^{-1}) \log \frac{d}{\delta}}{n}$$

and
$$\|\beta_{\text{MWLE}} - \beta^\star\|_2 \le c\sqrt{\frac{\text{Tr}(G_w H_w^{-2})\log\frac{d}{\delta}}{n}}.$$

Also notice that, $\mathbb{E}_{(x,y)\sim\mathbb{P}_T(x,y)}[\nabla\ell(x,y,\beta^\star)] = 0$. Therefore, with probability at least $1 - 3\delta$, we have

$$R_M(\beta_{\text{MWLE}}) \le \frac{c}{2}\frac{\text{Tr}(G_w H_w^{-1})\log\frac{d}{\delta}}{n} + \frac{c^3}{6}WB_3\text{Tr}(G_w H_w^{-2})^{1.5}(\frac{\log\frac{d}{\delta}}{n})^{1.5}.$$

If we further have $n \ge c(\frac{WB_3\text{Tr}(G_w H_w^{-2})^{1.5}}{\text{Tr}(G_w H_w^{-1})})^2\log(d/\delta)$, it then holds that

$$R_M(\beta_{\text{MWLE}}) \le c\frac{\text{Tr}(G_w H_w^{-1})\log\frac{d}{\delta}}{n}.$$

Note that

$$\max\left\{N_1, N_2, \left(\frac{WB_3\text{Tr}(G_w H_w^{-2})^{1.5}}{\text{Tr}(G_w H_w^{-1})}\right)^2\right\}$$

$$= \max\left\{W^2 B_2^2\|H_w^{-1}\|_2^2, W^2 B_3^2\text{Tr}(G_w H_w^{-2})\|H_w^{-1}\|_2^2, \left(\frac{W^3 B_1^2 B_2\|H_w^{-1}\|_2^3\log^{2\gamma}(W\lambda^{-1/2}\tilde\alpha_1)}{\text{Tr}(G_w H_w^{-2})}\right)^{\frac{2}{3}},\right.$$

$$\left.\left(\frac{W^4 B_1^3 B_3\|H_w^{-1}\|_2^4\log^{3\gamma}(W\lambda^{-1/2}\tilde\alpha_1)}{\text{Tr}(G_w H_w^{-2})}\right)^{\frac{1}{2}}, \frac{W^2 B_1^2\|H_w^{-1}\|_2^2\log^{2\gamma}(W\lambda^{-1/2}\tilde\alpha_1)}{\text{Tr}(G_w H_w^{-2})}\right\}$$

$$= W^2\cdot\max\{\tilde\alpha_2^2, \lambda\tilde\alpha_3^2, \tilde\alpha_1^{4/3}\tilde\alpha_2^{2/3}\lambda^{-2/3}\log^{4\gamma/3}(W\lambda^{-1/2}\tilde\alpha_1), \tilde\alpha_1^{3/2}\tilde\alpha_3^{1/2}\lambda^{-1/2}\log^{3\gamma/2}(W\lambda^{-1/2}\tilde\alpha_1), \lambda^{-1}\tilde\alpha_1^2\log^{2\gamma}(W\lambda^{-1/2}\tilde\alpha_1)\}$$

$$\le W^2\cdot\max\{\lambda^{-1}\tilde\alpha_1^2\log^{2\gamma}(W^2\lambda^{-1}\tilde\alpha_1^2), \tilde\alpha_2^2, \lambda\tilde\alpha_3^2\}$$

$$=: N^\star.$$

Here the first equation follows from the fact that
$\text{Tr}(G_w H_w^{-2}) = \text{Tr}(H_w^{-1/2}G_w H_w^{-1/2}H_w^{-1}) \le \|H_w^{-1}\|_2\text{Tr}(H_w^{-1/2}G_w H_w^{-1/2}) = \|H_w^{-1}\|_2\text{Tr}(G_w H_w^{-1})$.
To summarize, for any $\delta \in (0,1)$ and any $n \ge c\max\{N^\star\log(d/\delta), N(\delta), N'(\delta)\}$, with probability at least $1 - 3\delta$, we have

$$R_M(\beta_{\text{MWLE}}) \le c\frac{\text{Tr}(G_w H_w^{-1})\log\frac{d}{\delta}}{n}.$$

$\square$

In the following, we prove Lemma C.2 and C.3.

**Proof of Lemma C.2**

*Proof of Lemma C.2.* For notation simplicity, we denote $g := \nabla\ell_n^w(\beta^\star) - \mathbb{E}_{\mathbb{P}_S}[\nabla\ell_n^w(\beta^\star)]$. Note that
$$\begin{aligned}V &= n\cdot\mathbb{E}[\|A(\nabla\ell_n^w(\beta^\star) - \mathbb{E}[\nabla\ell_n^w(\beta^\star)])\|_2^2]\\ &= n\cdot\mathbb{E}[\nabla\ell_n^w(\beta^\star)^T A^T A\nabla\ell_n^w(\beta^\star)]\\ &= n\cdot\mathbb{E}[\text{Tr}(A\nabla\ell_n^w(\beta^\star)\nabla\ell_n^w(\beta^\star)^T A^T)]\\ &= \text{Tr}(AG_w A^T).\end{aligned}$$

By taking $A = H_w^{-1}$ in Assumption C.2, for any $\delta$ and any $n > N(\delta)$, we have with probability at least $1 - \delta$:

$$\|H_w^{-1}g\|_2 \le c\sqrt{\frac{\text{Tr}(G_w H_w^{-2})\log\frac{d}{\delta}}{n}} + WB_1\|H_w^{-1}\|_2\log^\gamma\left(\frac{WB_1\|H_w^{-1}\|_2}{\sqrt{\text{Tr}(G_w H_w^{-2})}}\right)\frac{\log\frac{d}{\delta}}{n}$$

$$= c\sqrt{\frac{\text{Tr}(G_w H_w^{-2})\log\frac{d}{\delta}}{n}} + WB_1\|H_w^{-1}\|_2\log^\gamma(W\lambda^{-1/2}\tilde\alpha_1)\frac{\log\frac{d}{\delta}}{n} \quad (50)$$

$$\left\|\nabla^2\ell_n^w(\beta^\star) - \mathbb{E}[\nabla^2\ell_n^w(\beta^\star)]\right\|_2 \le WB_2\sqrt{\frac{\log\frac{d}{\delta}}{n}}. \quad (51)$$

Let event $\tilde{A} := \{(50), (51) \text{ holds}\}$ and $\tilde{A}' := \{\ell_n^w(\cdot) \text{ has a unique local minimum, which is also global minimum}\}$. By Assumption C.2 and Assumption C.4, it then holds for any $\delta$ and any $n \geq \max\{N(\delta), N'(\delta)\}$ that $\mathbb{P}(\tilde{A} \cap \tilde{A}') \geq 1 - 2\delta$. Under the event $\tilde{A} \cap \tilde{A}'$, we have the following Taylor expansion:

$$\ell_n^w(\beta) - \ell_n^w(\beta^\star) \overset{\text{by Assumption C.1, C.3}}{\leq} (\beta - \beta^\star)^T \nabla \ell_n^w(\beta^\star) + \frac{1}{2}(\beta - \beta^\star)^T \nabla^2 \ell_n^w(\beta^\star)(\beta - \beta^\star) + \frac{WB_3}{6}\|\beta - \beta^\star\|_2^3$$

$$\overset{\mathbb{E}_{\mathbb{P}_S}[\nabla \ell_n^w(\beta^\star)]=0}{=} (\beta - \beta^\star)^T g + \frac{1}{2}(\beta - \beta^\star)^T \nabla^2 \ell_n^w(\beta^\star)(\beta - \beta^\star) + \frac{WB_3}{6}\|\beta - \beta^\star\|_2^3$$

$$\overset{\text{by (51)}}{\leq} (\beta - \beta^\star)^T g + \frac{1}{2}(\beta - \beta^\star)^T H_w(\beta - \beta^\star) + WB_2\sqrt{\frac{\log\frac{d}{\delta}}{n}}\|\beta - \beta^\star\|_2^2 + \frac{WB_3}{6}\|\beta - \beta^\star\|_2^3$$

$$\overset{\Delta_\beta := \beta - \beta^\star}{=} \Delta_\beta^T g + \frac{1}{2}\Delta_\beta^T H_w \Delta_\beta + WB_2\sqrt{\frac{\log\frac{d}{\delta}}{n}}\|\Delta_\beta\|_2^2 + \frac{WB_3}{6}\|\Delta_\beta\|_2^3$$

$$= \frac{1}{2}(\Delta_\beta - z)^T H_w(\Delta_\beta - z) - \frac{1}{2}z^T H_w z + WB_2\sqrt{\frac{\log\frac{d}{\delta}}{n}}\|\Delta_\beta\|_2^2 + \frac{WB_3}{6}\|\Delta_\beta\|_2^3 \tag{52}$$

where $z := -H_w^{-1} g$. Similarly

$$\ell_n^w(\beta) - \ell_n^w(\beta^\star) \geq \frac{1}{2}(\Delta_\beta - z)^T H_w(\Delta_\beta - z) - \frac{1}{2}z^T H_w z - WB_2\sqrt{\frac{\log\frac{d}{\delta}}{n}}\|\Delta_\beta\|_2^2 - \frac{WB_3}{6}\|\Delta_\beta\|_2^3. \tag{53}$$

Notice that $\Delta_{\beta^\star+z} = z$, by (50) and (52), we have
$\ell_n^w(\beta^\star + z) - \ell_n^w(\beta^\star)$

$$\leq -\frac{1}{2}z^T H_w z + WB_2\sqrt{\frac{\log\frac{d}{\delta}}{n}}\left(c\sqrt{\frac{\mathsf{Tr}(G_w H_w^{-2})\log\frac{d}{\delta}}{n}} + WB_1\|H_w^{-1}\|_2 \log^\gamma(W\lambda^{-1/2}\tilde{\alpha}_1)\frac{\log\frac{d}{\delta}}{n}\right)^2$$

$$+ \frac{WB_3}{6}\left(c\sqrt{\frac{\mathsf{Tr}(G_w H_w^{-2})\log\frac{d}{\delta}}{n}} + WB_1\|H_w^{-1}\|_2 \log^\gamma(W\lambda^{-1/2}\tilde{\alpha}_1)\frac{\log\frac{d}{\delta}}{n}\right)^3$$

$$\leq -\frac{1}{2}z^T H_w z + 2c^2 WB_2\mathsf{Tr}(G_w H_w^{-2})(\frac{\log\frac{d}{\delta}}{n})^{1.5} + 2W^3 B_1^2 B_2\|H_w^{-1}\|_2^2 \log^{2\gamma}(W\lambda^{-1/2}\tilde{\alpha}_1)(\frac{\log\frac{d}{\delta}}{n})^{2.5}$$

$$+ \frac{2}{3}c^3 WB_3\mathsf{Tr}(G_w H_w^{-2})^{1.5}(\frac{\log\frac{d}{\delta}}{n})^{1.5} + \frac{2}{3}W^4 B_1^3 B_3\|H_w^{-1}\|_2^3 \log^{3\gamma}(W\lambda^{-1/2}\tilde{\alpha}_1)(\frac{\log\frac{d}{\delta}}{n})^3. \tag{54}$$

For any $\beta \in \mathbb{B}_{\beta^\star}(3c\sqrt{\frac{\mathsf{Tr}(G_w H_w^{-2})\log\frac{d}{\delta}}{n}})$, by (53), we have

$$\ell_n^w(\beta) - \ell_n^w(\beta^\star) \geq \frac{1}{2}(\Delta_\beta - z)^T H_w(\Delta_\beta - z) - \frac{1}{2}z^T H_w z$$

$$- 9c^2 WB_2\mathsf{Tr}(G_w H_w^{-2})(\frac{\log\frac{d}{\delta}}{n})^{1.5} - \frac{9}{2}c^3 WB_3\mathsf{Tr}(G_w H_w^{-2})^{1.5}(\frac{\log\frac{d}{\delta}}{n})^{1.5}. \tag{55}$$

(55) - (54) gives
$\ell_n^w(\beta) - \ell_n^w(\beta^\star + z)$

$$\geq \frac{1}{2}(\Delta_\beta - z)^T H_w(\Delta_\beta - z)$$

$$- \left(11c^2 WB_2\mathsf{Tr}(G_w H_w^{-2})(\frac{\log\frac{d}{\delta}}{n})^{1.5} + \frac{31}{6}c^3 WB_3\mathsf{Tr}(G_w H_w^{-2})^{1.5}(\frac{\log\frac{d}{\delta}}{n})^{1.5}\right.$$

$$\left. + 2W^3 B_1^2 B_2\|H_w^{-1}\|_2^2 \log^{2\gamma}(W\lambda^{-1/2}\tilde{\alpha}_1)(\frac{\log\frac{d}{\delta}}{n})^{2.5} + \frac{2}{3}W^4 B_1^3 B_3\|H_w^{-1}\|_2^3 \log^{3\gamma}(W\lambda^{-1/2}\tilde{\alpha}_1)(\frac{\log\frac{d}{\delta}}{n})^3\right) \tag{56}$$

Consider the ellipsoid

$$
\mathcal{D} := \left\{ \beta \in \mathbb{R}^d \,\middle|\, \frac{1}{2}(\Delta_\beta - z)^T H_w (\Delta_\beta - z) \right.
$$

$$
\leq 11 c^2 W B_2 \mathsf{Tr}(G_w H_w^{-2})(\frac{\log \frac{d}{\delta}}{n})^{1.5} + \frac{31}{6} c^3 W B_3 \mathsf{Tr}(G_w H_w^{-2})^{1.5}(\frac{\log \frac{d}{\delta}}{n})^{1.5}
$$

$$
+ 2 W^3 B_1^2 B_2 \| H_w^{-1} \|_2^2 \log^{2\gamma}(W\lambda^{-1/2}\tilde{\alpha}_1)(\frac{\log \frac{d}{\delta}}{n})^{2.5}
$$

$$
\left. + \frac{2}{3} W^4 B_1^3 B_3 \| H_w^{-1} \|_2^3 \log^{3\gamma}(W\lambda^{-1/2}\tilde{\alpha}_1)(\frac{\log \frac{d}{\delta}}{n})^3 \right\}
$$

Then by (56), for any $\beta \in \mathbb{B}_{\beta^\star}(3c\sqrt{\frac{\mathsf{Tr}(G_w H_w^{-2})\log \frac{d}{\delta}}{n}}) \cap \mathcal{D}^C$, we have

$$
\ell_n^w(\beta) - \ell_n^w(\beta^\star + z) > 0. \tag{57}
$$

Notice that by the definition of $\mathcal{D}$, using $\lambda_{\min}^{-1}(H_w) = \| H_w^{-1} \|_2$, we have for any $\beta \in \mathcal{D}$,

$$
\| \Delta_\beta - z \|_2^2 \leq 22 c^2 \| H_w^{-1} \|_2 W B_2 \mathsf{Tr}(G_w H_w^{-2})(\frac{\log \frac{d}{\delta}}{n})^{1.5} + \frac{31}{3} c^3 \| H_w^{-1} \|_2 W B_3 \mathsf{Tr}(G_w H_w^{-2})^{1.5}(\frac{\log \frac{d}{\delta}}{n})^{1.5}
$$

$$
+ 4 \| H_w^{-1} \|_2 W^3 B_1^2 B_2 \| H_w^{-1} \|_2^2 \log^{2\gamma}(W\lambda^{-1/2}\tilde{\alpha}_1)(\frac{\log \frac{d}{\delta}}{n})^{2.5}
$$

$$
+ \frac{4}{3} \| H_w^{-1} \|_2 W^4 B_1^3 B_3 \| H_w^{-1} \|_2^3 \log^{3\gamma}(W\lambda^{-1/2}\tilde{\alpha}_1)(\frac{\log \frac{d}{\delta}}{n})^3.
$$

Thus for any $\beta \in \mathcal{D}$,

$$
\| \Delta_\beta \|_2^2 \leq 2(\| \Delta_\beta - z \|_2^2 + \| z \|_2^2)
$$

$$
\overset{\text{by}(50)}{\leq} 44 c^2 \| H_w^{-1} \|_2 W B_2 \mathsf{Tr}(G_w H_w^{-2})(\frac{\log \frac{d}{\delta}}{n})^{1.5} + \frac{62}{3} c^3 \| H_w^{-1} \|_2 W B_3 \mathsf{Tr}(G_w H_w^{-2})^{1.5}(\frac{\log \frac{d}{\delta}}{n})^{1.5}
$$

$$
+ 8 \| H_w^{-1} \|_2 W^3 B_1^2 B_2 \| H_w^{-1} \|_2^2 \log^{2\gamma}(W\lambda^{-1/2}\tilde{\alpha}_1)(\frac{\log \frac{d}{\delta}}{n})^{2.5}
$$

$$
+ \frac{8}{3} \| H_w^{-1} \|_2 W^4 B_1^3 B_3 \| H_w^{-1} \|_2^3 \log^{3\gamma}(W\lambda^{-1/2}\tilde{\alpha}_1)(\frac{\log \frac{d}{\delta}}{n})^3
$$

$$
+ 4 c^2 \mathsf{Tr}(G_w H_w^{-2})\frac{\log \frac{d}{\delta}}{n} + 4 W^2 B_1^2 \| H_w^{-1} \|_2^2 \log^{2\gamma}(W\lambda^{-1/2}\tilde{\alpha}_1)(\frac{\log \frac{d}{\delta}}{n})^2.
$$

To guarantee $\mathsf{Tr}(G_w H_w^{-2})\frac{\log \frac{d}{\delta}}{n}$ is the leading term, we only need $\mathsf{Tr}(G_w H_w^{-2})\frac{\log \frac{d}{\delta}}{n}$ to dominate the rest of the terms. Hence, if we further have $n \geq c N_1 \log(d/\delta)$, it then holds that

$$
\| \Delta_\beta \|_2^2 \leq 9 c^2 \mathsf{Tr}(G_w H_w^{-2})\frac{\log \frac{d}{\delta}}{n},
$$

i.e., $\beta \in \mathbb{B}_{\beta^\star}(3c\sqrt{\frac{\mathsf{Tr}(G_w H_w^{-2})\log \frac{d}{\delta}}{n}})$. Here

$$
N_1 := \max \left\{ W^2 B_2^2 \| H_w^{-1} \|_2^2, W^2 B_3^2 \mathsf{Tr}(G_w H_w^{-2}) \| H_w^{-1} \|_2^2, \left( \frac{W^3 B_1^2 B_2 \| H_w^{-1} \|_2^3 \log^{2\gamma}(W\lambda^{-1/2}\tilde{\alpha}_1)}{\mathsf{Tr}(G_w H_w^{-2})} \right)^{\frac{2}{3}}, \right.
$$

$$
\left. \left( \frac{W^4 B_1^3 B_3 \| H_w^{-1} \|_2^4 \log^{3\gamma}(W\lambda^{-1/2}\tilde{\alpha}_1)}{\mathsf{Tr}(G_w H_w^{-2})} \right)^{\frac{1}{2}}, \frac{W^2 B_1^2 \| H_w^{-1} \|_2^2 \log^{2\gamma}(W\lambda^{-1/2}\tilde{\alpha}_1)}{\mathsf{Tr}(G_w H_w^{-2})} \right\}.
$$

In other words, we show that $\mathcal{D} \subset \mathbb{B}_{\beta^\star}(3c\sqrt{\frac{\mathsf{Tr}(G_w H_w^{-2})\log \frac{d}{\delta}}{n}})$. Recall that by (57), we know that for any $\beta \in \mathbb{B}_{\beta^\star}(3c\sqrt{\frac{\mathsf{Tr}(G_w H_w^{-2})\log \frac{d}{\delta}}{n}}) \cap \mathcal{D}^C$,

$$
\ell_n^w(\beta) - \ell_n^w(\beta^\star + z) > 0.
$$

Note that $\beta^\star + z \in \mathcal{D}$. Hence there is a local minimum of $\ell_n^w(\beta)$ in $\mathcal{D}$. Under the event $\tilde{A}'$, we know that the global minimum of $\ell_n^w(\beta)$ is in $\mathcal{D}$, i.e.,

$$\beta_{\mathsf{MWLE}} \in \mathcal{D} \subset \mathbb{B}_{\beta^\star}(3c\sqrt{\frac{\mathsf{Tr}(G_w H_w^{-2})\log\frac{d}{\delta}}{n}}).$$

$\square$

**Proof of Lemma C.3**

*Proof of Lemma C.3.* Let $\tilde{E} := \{\beta_{\mathsf{MWLE}} \in \mathcal{D} \subset \mathbb{B}_{\beta^\star}(\sqrt{\frac{\mathsf{Tr}(G_w H_w^{-2})\log\frac{d}{\delta}}{n}})\}$. Then by the proof of Lemma C.2, for any $\delta \in (0,1)$ and any $n \geq c\max\{N_1\log(d/\delta), N(\delta), N'(\delta)\}$, we have $\mathbb{P}(\tilde{E}) \geq 1 - 2\delta$.

By taking $A = H_w^{-\frac{1}{2}}$ in Assumption C.2, for any $\delta \in (0,1)$ and any $n \geq N(\delta)$, with probability at least $1 - \delta$, we have:

$$\|H_w^{-\frac{1}{2}}g\|_2 \leq c\sqrt{\frac{\mathsf{Tr}(G_w H_w^{-1})\log\frac{d}{\delta}}{n}} + WB_1\|H_w^{-\frac{1}{2}}\|_2 \log^\gamma\left(\frac{WB_1\|H_w^{-\frac{1}{2}}\|_2}{\sqrt{\mathsf{Tr}(G_w H_w^{-1})}}\right)\frac{\log\frac{d}{\delta}}{n}$$

$$\leq c\sqrt{\frac{\mathsf{Tr}(G_w H_w^{-1})\log\frac{d}{\delta}}{n}} + WB_1\|H_w^{-\frac{1}{2}}\|_2 \log^\gamma\left(\frac{WB_1\|H_w^{-\frac{1}{2}}\|_2}{\sqrt{\mathsf{Tr}(G_w H_w^{-2})\|H_w^{-1}\|_2^{-1}}}\right)\frac{\log\frac{d}{\delta}}{n}$$

$$= c\sqrt{\frac{\mathsf{Tr}(G_w H_w^{-1})\log\frac{d}{\delta}}{n}} + WB_1\|H_w^{-\frac{1}{2}}\|_2 \log^\gamma(W\lambda^{-1/2}\tilde{\alpha}_1)\frac{\log\frac{d}{\delta}}{n} \tag{58}$$

We denote $\tilde{E}' := \{(58)\text{ holds}\}$. Then for any $\delta$ and any $n \geq c\max\{N_1(M)\log(d/\delta), N(\delta), N'(\delta)\}$, we have $\mathbb{P}(\tilde{E} \cap \tilde{E}') \geq 1 - 3\delta$.

Under $\tilde{E} \cap \tilde{E}'$, $\beta_{\mathsf{MWLE}} \in \mathcal{D}$, i.e.,

$$\frac{1}{2}(\Delta_{\beta_{\mathsf{MWLE}}} - z)^T H_w(\Delta_{\beta_{\mathsf{MWLE}}} - z)$$

$$\leq 11c^2 WB_2\mathsf{Tr}(G_w H_w^{-2})(\frac{\log\frac{d}{\delta}}{n})^{1.5} + \frac{31}{6}c^3 WB_3\mathsf{Tr}(G_w H_w^{-2})^{1.5}(\frac{\log\frac{d}{\delta}}{n})^{1.5}$$

$$+ 2W^3 B_1^2 B_2\|H_w^{-1}\|_2^2 \log^{2\gamma}(W\lambda^{-1/2}\tilde{\alpha}_1)(\frac{\log\frac{d}{\delta}}{n})^{2.5} + \frac{2}{3}W^4 B_1^3 B_3\|H_w^{-1}\|_2^3 \log^{3\gamma}(W\lambda^{-1/2}\tilde{\alpha}_1)(\frac{\log\frac{d}{\delta}}{n})^3.$$

In other words,

$$\|H_w^{\frac{1}{2}}(\Delta_{\beta_{\mathsf{MWLE}}} - z)\|_2^2$$

$$\leq 22c^2 WB_2\mathsf{Tr}(G_w H_w^{-2})(\frac{\log\frac{d}{\delta}}{n})^{1.5} + \frac{31}{3}c^3 WB_3\mathsf{Tr}(G_w H_w^{-2})^{1.5}(\frac{\log\frac{d}{\delta}}{n})^{1.5}$$

$$+ 4W^3 B_1^2 B_2\|H_w^{-1}\|_2^2 \log^{2\gamma}(W\lambda^{-1/2}\tilde{\alpha}_1)(\frac{\log\frac{d}{\delta}}{n})^{2.5} + \frac{4}{3}W^4 B_1^3 B_3\|H_w^{-1}\|_2^3 \log^{3\gamma}(W\lambda^{-1/2}\tilde{\alpha}_1)(\frac{\log\frac{d}{\delta}}{n})^3. \tag{59}$$

Thus we have

$$\|H_w^{\frac{1}{2}}(\beta_{\mathsf{MWLE}} - \beta^\star)\|_2^2$$

$$= \|H_w^{\frac{1}{2}}\Delta_{\beta_{\mathsf{MWLE}}}\|_2^2$$

$$= \|H_w^{\frac{1}{2}}(\Delta_{\beta_{\mathsf{MWLE}}} - z) + H_w^{\frac{1}{2}}z\|_2^2$$

$$\leq 2\|H_w^{\frac{1}{2}}(\Delta_{\beta_{\mathsf{MWLE}}} - z)\|_2^2 + 2\|H_w^{\frac{1}{2}}z\|_2^2$$

$$= 2\|H_w^{\frac{1}{2}}(\Delta_{\beta_{\mathsf{MWLE}}} - z))\|_2^2 + 2\|H_w^{-\frac{1}{2}}g\|_2^2$$

$$\overset{\text{by}(59)\text{and}(58)}{\leq} 4c^2 \frac{\mathsf{Tr}(G_w H_w^{-1})\log\frac{d}{\delta}}{n}$$

$$+ 44c^2 W B_2 \mathsf{Tr}(G_w H_w^{-2})(\frac{\log\frac{d}{\delta}}{n})^{1.5} + \frac{62}{3}c^3 W B_3 \mathsf{Tr}(G_w H_w^{-2})^{1.5}(\frac{\log\frac{d}{\delta}}{n})^{1.5}$$

$$+ 8W^3 B_1^2 B_2\|H_w^{-1}\|_2^2 \log^{2\gamma}(W\lambda^{-1/2}\tilde{\alpha}_1)(\frac{\log\frac{d}{\delta}}{n})^{2.5} + \frac{8}{3}W^4 B_1^3 B_3\|H_w^{-1}\|_2^3 \log^{3\gamma}(W\lambda^{-1/2}\tilde{\alpha}_1)(\frac{\log\frac{d}{\delta}}{n})^3$$

$$+ 4W^2 B_1^2\|H_w^{-1}\|_2 \log^{2\gamma}(W\lambda^{-1/2}\tilde{\alpha}_1)(\frac{\log\frac{d}{\delta}}{n})^2 \tag{60}$$

To guarantee $\frac{\mathsf{Tr}(G_w H_w^{-1})\log\frac{d}{\delta}}{n}$ is the leading term, we only need $\frac{\mathsf{Tr}(G_w H_w^{-1})\log\frac{d}{\delta}}{n}$ to dominate the rest of the terms. Hence, if we further have $n \geq cN_2 \log(d/\delta)$, we have

$$\|H_w^{\frac{1}{2}}(\beta_{\mathsf{MWLE}} - \beta^\star)\|_2^2 \leq 9c^2 \frac{\mathsf{Tr}(G_w H_w^{-1})\log\frac{d}{\delta}}{n}.$$

Here

$$N_2 := \max\left\{ \left(\frac{W B_2 \mathsf{Tr}(G_w H_w^{-2})}{\mathsf{Tr}(G_w H_w^{-1})}\right)^2, \left(\frac{W B_3 \mathsf{Tr}(G_w H_w^{-2})^{1.5}}{\mathsf{Tr}(G_w H_w^{-1})}\right)^2, \left(\frac{W^3 B_1^2 B_2\|H_w^{-1}\|_2^2 \log^{2\gamma}(W\lambda^{-1/2}\tilde{\alpha}_1)}{\mathsf{Tr}(G_w H_w^{-1})}\right)^{\frac{2}{3}}, \right.$$

$$\left. \left(\frac{W^4 B_1^3 B_3\|H_w^{-1}\|_2^3 \log^{3\gamma}(W\lambda^{-1/2}\tilde{\alpha}_1)}{\mathsf{Tr}(G_w H_w^{-1})}\right)^{\frac{1}{2}}, \frac{W^2 B_1^2\|H_w^{-1}\|_2 \log^{2\gamma}(W\lambda^{-1/2}\tilde{\alpha}_1)}{\mathsf{Tr}(G_w H_w^{-1})}\right\}.$$

To summarize, we show that for any $\delta$ and any $n \geq c\max\{N_1 \log(d/\delta), N_2 \log(d/\delta), N(\delta), N'(\delta)\}$, with probability at least $1 - 3\delta$, we have

$$\|H_w^{\frac{1}{2}}(\beta_{\mathsf{MWLE}} - \beta^\star)\|_2^2 \leq 9c^2 \frac{\mathsf{Tr}(G_w H_w^{-1})\log\frac{d}{\delta}}{n}.$$

$\square$

### C.3  PROOFS FOR THEOREM 5.3

*Proof of Theorem 5.3.* For any $W > 1$, we construct $\mathbb{P}_S(X)$, $\mathbb{P}_T(X)$, $\mathcal{M}$ and $\mathcal{F}$ as follows. We define $\mathbb{P}_T(X) := \mathsf{Uniform}(\mathbb{B}(1))$ and $\mathbb{P}_S(X) := \mathsf{Uniform}(\mathbb{B}(W^{\frac{1}{d}}))$, where $\mathbb{B}(1)$ and $\mathbb{B}(W^{\frac{1}{d}})$ are $d$-dimensional balls centered around the original with radius $1$ and $W^{\frac{1}{d}}$, respectively. For notation simplicity, we denote $Q := \mathbb{B}(1)$ and $P := \mathbb{B}(W^{\frac{1}{d}})$ in the following. The density ratios is then given by

$$w(x) := \frac{d\mathbb{P}_T(x)}{d\mathbb{P}_S(x)} = \begin{cases} W & x \in Q \\ 0 & x \notin Q \end{cases},$$

which is upper bounded by $W$. We further have

$$\mathcal{I}_S(\beta) = \mathbb{E}_{x\sim\mathbb{P}_S(X)}[xx^T] = \frac{W^{\frac{2}{d}}}{3d}I_d \succ 0, \; \mathcal{I}_T(\beta) = \mathbb{E}_{x\sim\mathbb{P}_T(X)}[xx^T] = \frac{1}{3d}I_d \succ 0.$$

Let $\mathcal{F} := \{f(y\,|\,x;\beta)\,|\,\beta \in \mathbb{R}^d\}$ be the linear regression class, i.e., $-\log f(y\,|\,x;\beta) = (\log 2\pi)/2 + (y - x^T\beta)^2/2$. We assume the true conditional distribution belongs to a class $\mathcal{M}$ that is defined as

$$\mathcal{M} := \left\{Y\,|\,X \text{ s.t } p(y\,|\,x) = f(y\,|\,x;\beta_1^\star)\mathbf{1}_{\{x\in Q\}} + f(y\,|\,x;\beta_2^\star)\mathbf{1}_{\{x\in P\setminus Q\}}, \beta_1^\star, \beta_2^\star \in \mathbb{B}_{\beta_0}(B)\right\}$$

for some $\beta_0 \in \mathbb{R}^d$ and $B > 0$. We utilize the function class $\mathcal{F}$ to approximate the true conditional density function, which subsequently results in model mis-specification. In the sequel, we will show the lower bound of excess risk for any estimators under this model class $\mathcal{M}$.

Fix any ground truth model $M \in \mathcal{M}$, that is, we are assuming the true conditional distribution follows the form:

$$p(y \mid x) = f(y \mid x; \beta_1^\star) \mathbf{1}_{\{x \in Q\}} + f(y \mid x; \beta_2^\star) \mathbf{1}_{\{x \in P \setminus Q\}},$$

where $\beta_1^\star$ and $\beta_2^\star$ are arbitrarily chosen fixed points from $\mathbb{B}_{\beta_0}(B)$. Note that the model is actually well-specified on the target domain. Hence the optimal fit on the target is given by

$$\beta^\star(M) = \arg\min_\beta \mathbb{E}_{(x,y)\sim \mathbb{P}_T(X,Y)}[\ell(x,y,\beta)] = \beta_1^\star.$$

For linear regression, it is easy to verify that Assumption B.2, B.3 and B.4 hold. Let $R_0$ and $R_1$ be the parameters chosen by Lemma A.5. Then similar to the proofs of Theorem 3.2, we have

$$
\begin{aligned}
&\inf_{\hat{\beta}} \sup_{M \in \mathcal{M}} \mathbb{E}_{(x_i,y_i)\sim \mathbb{P}_S(X,Y)} \left[ R_M(\hat{\beta}) \right] \\
&= \inf_{\hat{\beta}} \sup_{\beta_1^\star, \beta_2^\star \in \mathbb{B}_{\beta_0}(B)} \mathbb{E}_{(x_i,y_i)\sim \mathbb{P}_S(X,Y)} \left[ R_{\beta_1^\star}(\hat{\beta}) \right] \\
&\geq \inf_{\hat{\beta}} \sup_{\beta_1^\star, \beta_2^\star \in \mathbb{B}_{\beta_0}(R_1)} \mathbb{E}_{(x_i,y_i)\sim \mathbb{P}_S(X,Y)} \left[ R_{\beta_1^\star}(\hat{\beta}) \right] \\
&\geq \inf_{\hat{\beta} \in \mathbb{B}_{\beta_0}(R_0)} \sup_{\beta_1^\star, \beta_2^\star \in \mathbb{B}_{\beta_0}(R_1)} \mathbb{E}_{(x_i,y_i)\sim \mathbb{P}_S(X,Y)} \left[ R_{\beta_1^\star}(\hat{\beta}) \right] \\
&\geq \frac{1}{4} \inf_{\hat{\beta} \in \mathbb{B}_{\beta_0}(R_0)} \sup_{\beta_1^\star, \beta_2^\star \in \mathbb{B}_{\beta_0}(R_1)} \mathbb{E}_{(x_i,y_i)\sim \mathbb{P}_S(X,Y)} \left[ (\hat{\beta} - \beta_1^\star)^T \mathcal{I}_T(\beta_0)(\hat{\beta} - \beta_1^\star) \right] \\
&\geq \frac{1}{4} \inf_{\hat{\beta} \in \mathbb{B}_{\beta_0}(R_0)} \sup_{\beta_1^\star, \beta_2^\star \in C_{\beta_0}(\frac{R_1}{\sqrt{d}})} \mathbb{E}_{(x_i,y_i)\sim \mathbb{P}_S(X,Y)} \left[ (\hat{\beta} - \beta_1^\star)^T \mathcal{I}_T(\beta_0)(\hat{\beta} - \beta_1^\star) \right] \\
&= \frac{1}{4} \inf_{\hat{\beta} \in \mathbb{B}_{\beta_0}(R_0)} \sup_{[\beta_1^{\star T}, \beta_2^{\star T}] \in C_{[\beta_0^T, \beta_0^T]}(\frac{R_1}{\sqrt{d}})} \mathbb{E}_{(x_i,y_i)\sim \mathbb{P}_S(X,Y)} \left[ (\hat{\beta} - \beta_1^\star)^T \mathcal{I}_T(\beta_0)(\hat{\beta} - \beta_1^\star) \right] \quad (61)
\end{aligned}
$$

By Theorem 1 in Gill & Levit (1995) (multivariate van Trees inequality) with $\psi(\beta_1^\star, \beta_2^\star) = \beta_1^\star$, $C(\beta_1^\star, \beta_2^\star) \equiv C := [WI_d, 0] \in \mathbb{R}^{d \times 2d}$ and $B(\beta_1^\star, \beta_2^\star) \equiv B := \mathcal{I}_T^{-1}(\beta_0)$, we have for any estimator $\hat{\beta}$ and good prior density $\lambda$ that supported on $C_{[\beta_0^T, \beta_0^T]}(\frac{R_1}{\sqrt{d}})$,

$$\mathbb{E}_{[\beta_1^{\star T}, \beta_2^{\star T}] \sim \lambda} \mathbb{E}_{(x_i,y_i)\sim \mathbb{P}_S(X,Y)} \left[ (\hat{\beta} - \beta_1^\star)^T \mathcal{I}_T(\beta_0)(\hat{\beta} - \beta_1^\star) \right] \geq \frac{(Wd)^2}{2nWd + \tilde{\mathcal{I}}(\lambda)},$$

where

$$\tilde{\mathcal{I}}(\lambda) = \int_{C_{[\beta_0^T, \beta_0^T]}(\frac{R_1}{\sqrt{d}})} \left( \sum_{i,j,k,\ell} B_{ij} C_{ik} C_{j\ell} \frac{\partial}{\partial \tilde{\beta}_k} \lambda(\tilde{\beta}) \frac{\partial}{\partial \tilde{\beta}_\ell} \lambda(\tilde{\beta}) \right) \frac{1}{\lambda(\tilde{\beta})} d\tilde{\beta}.$$

Let $\tilde{\beta}_0 = [\beta_{0,1}, \ldots, \beta_{0,d}, \beta_{0,1}, \ldots, \beta_{0,d}]^T$, $\tilde{\beta} = [\beta_1, \ldots, \beta_{2d}]^T$ and

$$f_i(x) := \frac{\pi\sqrt{d}}{4R_1} \cos\left( \frac{\pi\sqrt{d}}{2R_1}(x - \tilde{\beta}_{0,i}) \right), \ i = 1, \ldots, 2d.$$

We define the prior density as

$$\lambda(\tilde{\beta}) := \begin{cases} \Pi_{i=1}^{2d} f_i(\beta_i) & \tilde{\beta} \in C_{[\beta_0^T, \beta_0^T]}(\frac{R_1}{\sqrt{d}}) \\ 0 & \tilde{\beta} \notin C_{[\beta_0^T, \beta_0^T]}(\frac{R_1}{\sqrt{d}}) \end{cases}.$$

Then following the same argument as in the proof of Lemma A.6, we have

$$\tilde{\mathcal{I}}(\lambda) = \frac{\pi^2 d}{R_1^2} \mathsf{Tr}(BCC^T) = \frac{\pi^2 W^2 d}{R_1^2} \mathsf{Tr}(\mathcal{I}_T^{-1}(\beta_0)).$$

As a result, for any estimator $\hat{\beta}$, we have

$$\mathbb{E}_{[\beta_1^{\star T}, \beta_2^{\star T}] \sim \lambda} \mathbb{E}_{(x_i, y_i) \sim \mathbb{P}_S(X,Y)} \left[ (\hat{\beta} - \beta_1^{\star})^T \mathcal{I}_T(\beta_0)(\hat{\beta} - \beta_1^{\star}) \right]$$

$$\geq \frac{(Wd)^2}{2nWd + \frac{\pi^2 W^2 d}{R_1^2} \mathsf{Tr}(\mathcal{I}_T^{-1}(\beta_0))},$$

which implies

$$\sup_{[\beta_1^{\star T}, \beta_2^{\star T}] \in C_{[\beta_0^T, \beta_0^T]}(\frac{R_1}{\sqrt{d}})} \mathbb{E}_{(x_i, y_i) \sim \mathbb{P}_S(X,Y)} \left[ (\hat{\beta} - \beta_1^{\star})^T \mathcal{I}_T(\beta_0)(\hat{\beta} - \beta_1^{\star}) \right]$$

$$\geq \mathbb{E}_{[\beta_1^{\star T}, \beta_2^{\star T}] \sim \lambda} \mathbb{E}_{(x_i, y_i) \sim \mathbb{P}_S(X,Y)} \left[ (\hat{\beta} - \beta^{\star})^T \mathcal{I}_T(\beta_0)(\hat{\beta} - \beta^{\star}) \right]$$

$$\geq \frac{(Wd)^2}{2nWd + \frac{\pi^2 W^2 d}{R_1} \mathsf{Tr}(\mathcal{I}_T^{-1}(\beta_0))}. \tag{62}$$

Combine (61) and (62), we have

$$\inf_{\hat{\beta}} \sup_{M \in \mathcal{M}} \mathbb{E}_{(x_i, y_i) \sim \mathbb{P}_S(X,Y)} \left[ R_M(\hat{\beta}) \right] \geq \frac{1}{4} \cdot \frac{(Wd)^2}{2nWd + \frac{\pi^2 W^2 d}{R_1} \mathsf{Tr}(\mathcal{I}_T^{-1}(\beta_0))} \gtrsim \frac{Wd}{n}$$

when $n$ is sufficiently large.

Recall that

$$H_w(M) = \mathbb{E}_{(x,y) \sim \mathbb{P}_T(X,Y)} \left[ \nabla^2 \ell(x, y, \beta^{\star}(M)) \right] = \mathbb{E}_{(x,y) \sim \mathbb{P}_T(X,Y)} \left[ \nabla^2 \ell(x, y, \beta_1^{\star}) \right] = \mathcal{I}_T(\beta_1^{\star}).$$

and by the definition of $w(x)$, we further have

$$\begin{aligned} G_w(M) &= \mathbb{E}_{(x,y) \sim \mathbb{P}_S(X,Y)} \left[ w(x)^2 \nabla \ell(x, y, \beta^{\star}(M)) \nabla \ell(x, y, \beta^{\star}(M))^T \right] \\ &= \mathbb{E}_{(x,y) \sim \mathbb{P}_T(X,Y)} \left[ w(x) \nabla \ell(x, y, \beta^{\star}(M)) \nabla \ell(x, y, \beta^{\star}(M))^T \right] \\ &= W \mathbb{E}_{(x,y) \sim \mathbb{P}_T(X,Y)} \left[ \nabla \ell(x, y, \beta^{\star}(M)) \nabla \ell(x, y, \beta^{\star}(M))^T \right] \\ &= W \mathbb{E}_{(x,y) \sim \mathbb{P}_T(X,Y)} \left[ \nabla \ell(x, y, \beta_1^{\star}) \nabla \ell(x, y, \beta_1^{\star})^T \right] \\ &= W \mathcal{I}_T(\beta_1^{\star}). \end{aligned}$$

Therefore $\mathsf{Tr}(G_w(M) H_w(M)^{-1}) = Wd$, which gives the desired result. What remains is to verify that $\mathcal{M}$ satisfies Assumption C.1, C.2, C.3 and C.4. Assumption C.1 is trivially satisfied. For Assumption C.2 and C.3, notice that

$$\begin{aligned} \nabla \ell(x, y, \beta) &= -x(y - x^T \beta), \\ \nabla^2 \ell(x, y, \beta) &= xx^T, \\ \nabla^3 \ell(x, y, \beta) &= 0. \end{aligned}$$

and

$$w(x) := \frac{d\mathbb{P}_T(x)}{d\mathbb{P}_S(x)} = \begin{cases} W & x \in Q \\ 0 & x \notin Q \end{cases},$$

By the definition of $\mathcal{M}$, we can write the distribution of $y$ as

$$y_i = \begin{cases} x_i^T \beta_1^{\star} + \epsilon_i & x_i \in Q \\ x_i^T \beta_2^{\star} + \epsilon_i & x_i \notin Q \end{cases},$$

where $\epsilon_i$ is a $\mathcal{N}(0,1)$ noise independent of all $x_i$'s. Therefore let $u_i := Aw(x_i) \nabla \ell(x_i, y_i, \beta^{\star}(M))$, we have

$$u_i = \begin{cases} -WA x_i \epsilon_i & x_i \in Q \\ 0 & x_i \notin Q \end{cases},$$

which indicates that $\|u_i\|$ is $\|A\|W$-subgaussian. Therefore by Lemma D.1, the vector concentration in Assumption C.2 is satisfied with $\gamma = 0.5$, $B_1 = 1$. For the matrix concentration, notice that

$$w(x_i)\nabla^2\ell(x_i, y_i, \beta^\star(M)) = \begin{cases} W x_i x_i^T & x_i \in Q \\ 0 & x_i \notin Q \end{cases},$$

therefore my matrix Hoeffding, $\|w(x_i)\nabla^2\ell(x_i, y_i, \beta^\star(M))\|_2 \leq W$, thus the matrix concentration in Assumption C.2 is satisfied with $B_2 = 1$. Further more, $N(\delta) = 0$ is enough for satisfying Assumption C.2.

Assumption C.3 is satisfied with $B_3 = 0$ since $\nabla^3\ell(x, y, \beta) = 0$.

For Assumption C.4, we can prove that it is satisfied with $N'(\delta) = \max\{8W\log\frac{1}{\delta}, 2dW\}$. This is because,

$$\begin{aligned}
\mathbb{P}(\nabla^2\ell_n^w(\beta) \succ 0 \text{ for all } \beta) &= \mathbb{P}(\frac{W}{n}\sum_{i=1}^n x_i x_i^T \mathbb{I}_{x_i \in Q} \succ 0) \\
&\geq \mathbb{P}(\#\{x_i \in Q\} > d) \\
&= 1 - \mathbb{P}(\#\{x_i \in Q\} \leq d) \\
&\overset{\text{by Chernoff bound}}{\geq} 1 - \exp(-\frac{\mu}{2}(1 - \frac{d}{\mu})^2) \\
&\geq 1 - \delta,
\end{aligned}$$

where $\mu := \frac{n}{W}$, and the last inequality hold when $n \geq N'(\delta)$. Therefore when $n \geq N'(\delta)$, with probability at least $1 - \delta$, $\ell_n^w$ is strictly convex, therefore has a unique local minimum which is also the global minimum. □

# D  AUXILIARIES

In this section, we present several auxiliary lemmas and propositions.

## D.1  CONCENTRATION FOR GRADIENT AND HESSIAN

The following lemma gives a generic version of Bernstein inequality for vectors.

**Lemma D.1.** *Let $u, u_1, \cdots, u_n$ be i.i.d. mean-zero random vectors. We denote $V = \mathbb{E}[\|u\|_2^2]$ and*

$$B_u^{(\alpha)} := \inf\{t > 0 : \mathbb{E}[\exp(\|u\|^\alpha/t^\alpha)] \leq 2\}, \quad \alpha \geq 1.$$

*Suppose $B_u^{(\alpha)} < \infty$ for some $\alpha \geq 1$. Then there exists an absolute constant $c > 0$ such that for all $\delta \in (0, 1)$, with probability at least $1 - \delta$:*

$$\left\|\frac{1}{n}\sum_{i=1}^n u_i\right\|_2 \leq c\left(\sqrt{\frac{V\log\frac{d}{\delta}}{n}} + B_u^{(\alpha)}\left(\log\frac{B_u^{(\alpha)}}{\sqrt{V}}\right)^{1/\alpha}\frac{\log\frac{d}{\delta}}{n}\right).$$

*Proof.* See Proposition 2 in Koltchinskii et al. (2011) for the proof. □

The following proposition shows that when gradient and Hessian are bounded or sub-Gaussian (sub-exponential), Assumption A.1 is naturally satisfied.

**Proposition D.2.** *If $\|\nabla\ell(x_i, y_i, \beta^\star)\|_2 \leq b_1$ for all $i \in [n]$, then the vector concentration (5) is satisfied with $B_1 = b_1$ and $\gamma = 0$. Alternatively, if $\|\nabla\ell(x_i, y_i, \beta^\star)\|_2$ is $b_1$-subgaussian, then (5) is satisfied with $B_1 = b_1$ and $\gamma = 1/2$. When $\|\nabla\ell(x_i, y_i, \beta^\star)\|_2$ is $b_1$-subexponential, then (5) is satisfied with $B_1 = b_1$ and $\gamma = 1$. For the Hessian concenntration, if $\|\nabla^2\ell(x_i, y_i, \beta^\star)\|_2 \leq b_2$ for all $i \in [n]$, then (6) is satisfied with $B_2 = b_2$.*

*Proof.* The vector concentration (5) is a direct proposition of Lemma D.1. The Hessian concentration (6) is a direct consequence of matrix Hoeffding inequality. □