# OpenReview forum: "Maximum Likelihood Estimation is All You Need for Well-Specified Covariate Shift"
_ICLR.cc/2024/Conference — ICLR 2024 poster_

### Official Review · Reviewer_jKTp · 2023-10-19

**Soundness:** 3 good
**Presentation:** 2 fair
**Contribution:** 2 fair
**Rating:** 6
**Confidence:** 3

**Summary:**

This paper studies the problem of covariate shift, where a model is trained on a dataset sampled from a source distribution and aims to achieve minimal generalization error on a target distribution. It is presumed that both the source and target distributions share the same conditional distribution of label $Y$ given covariates $X$ (i.e., the so-called well-specified case) but differ in their distribution over the covariates. Under specific assumptions about the model class and the source and target distributions, the authors demonstrate that the minimax generalization error on the target distribution is achieved by the MLE (Maximum Likelihood Estimation) using only data sampled from the source. The authors further apply their findings to several (generalized) linear models. In conclusion, the authors show that there are instances in the miss-specified case where the MLE is not consistent, whereas a weighted version of MLE achieves tight sample complexity.

**Strengths:**

This is a technically solid paper, which demonstrates, for certain natural models, that the MLE achieves the minimax optimal sample complexity for covariate shift in the well-specified case. The authors also apply their findings to several (generalized) linear models, which may have potential real-world applications. The examples distinguishing between the well-specified and misspecified cases are also intriguing.

**Weaknesses:**

My primary reservation about this paper is that it might overly exaggerate its contributions. While the title suggests that MLE is "All You Need" for well-specified covariate shift, the authors introduce a series of quite non-trivial assumptions in Assumption A. It seems to me that Theorem 3.1 is essentially a direct corollary of these assumptions. Technically, these assumptions appear to essentially state that the MLE converges to the optimal parameter — the very result that Theorem 3.1 claims to prove. Additionally, regarding applications, the paper limits its focus to only (generalized) linear models. This narrow scope hardly convinces that the MLE is indeed "All You Need".

I would urge the authors to represent their results more accurately. Specifically, I recommend changing the title to "Minimax Optimal Sample Complexity for Well-Specified Covariate Shift via MLE" or something along those lines.

I would like to outline the following further comments:

- Assumption A and C apply to both the model class and the source and target distributions, not only to the model class as claimed by the authors.
- In Section 4, regarding the applications, it would be beneficial to also mention the lower bounds for each example.
- The current length of the paper, spanning 50 pages, is excessive. It would be beneficial to provide an outline of the primary proof techniques. This would help readers in distinguishing between essential steps and routine procedures.

**Questions:**

See above.

---

> ### Author Response · Authors · 2023-11-15
> **Author Response**
>
> We thank the reviewer for the comments.
>
> **Regarding Assumptions:**
> We highlight that we not only established the consistency of MLE (which is known before), but also critically establish that statistical rate of MLE among all algorithms. The assumptions used in this paper are essentially standard assumptions that are used in the literature to ensure that MLE is consistent in the basic setting without any distribution shift. This paper goes much further to prove that MLE is not only consistent, but also minimax-optimal even under covariate shift, with almost no additional assumption. In this sense, we consider our assumptions not strong.
>
> **Regarding Applications:**
> Our main results hold for general model classes beyond generalized linear models. The applications section is primarily intended to illustrate the versatility of our framework. Notably, our third example deviates from the typical GLM framework. Unlike GLMs, which generally feature a convex loss function and a unique minimum, phase retrieval does not conform to these characteristics, thereby extending the scope of our theory beyond GLMs.
>
> **Regarding further comments:**
>
> Regarding Assumption A and C:
> We first fix a pair of source and target distributions and then make assumptions on the model class. Our minimax optimality results are instance-specific (which is stronger than standard instance independent minimax result that allows the choice of worse case instance), i.e., given a pair of source and target distributions, when the model class satisfies the assumptions, MLE is minimax optimal.
>
> Regarding comments 2&3: Thanks for your valuable comments. We will address these issues in our revision.

---

### Official Review · Reviewer_Vzp3 · 2023-10-22

**Soundness:** 4 excellent
**Presentation:** 4 excellent
**Contribution:** 4 excellent
**Rating:** 8
**Confidence:** 4

**Summary:**

This paper working under the classic covariate shift seating where the marginal distribution of X is different between source and target data but the conditional distributions of Y|X are the same, and proved the surprising yet elegant result that classical Maximum Likelihood Estimation (MLE) purely using source data achieves the minimax optimality  under the well-specified setting. This result holds for a large family of parametric models and the authors illustrated this in linear regression, logistic regression, and phase retrieval, where no boundedness condition on the density ratio is required. They further proved that for the misspecified setting, MLE can perform poorly, and the Maximum Weighted Likelihood Estimator (MWLE) emerges as minimax optimal in specific scenarios, outperforming MLE.

**Strengths:**

The presentation of the covariate shift setting and the author's result is very clean and well-written, which is also reflected in the authors' choice of using linear, logistic models to illustrate their main result. The upper and lower matching bounds on the MLE estimator is nicely presented with necessary conditions along with the MWLE estimator under the misspecified model, which imposes stronger conditions.

**Weaknesses:**

It would be great it the authors could present more intuition on why simple MLE works so well in well-specified model, where the estimator uses purely source data. In addition, perhaps the authors could cite and compare with some growingly popular and relevant literature in nonparametric settings of covariate shift that leverages conformal prediction, such as https://arxiv.org/abs/1904.06019 and https://arxiv.org/abs/2203.01761;
Some comparisons between prediction and estimation problem under covariate shift could be very beneficial to the community.

**Questions:**

In the well-specified case where the MLE estimator is minimax optimal, is there anything to gain if there is also access to X and Y in the target population?

---

> ### Author Response · Authors · 2023-11-15
> **Author Response**
>
> We thank the reviewer for the positive feedback.
>
> **Response to Weaknesses:**
>
> To gain some intuition, from the lower bound we can see that $\\mathcal{I}_S^{-1}$ captures the variance of the parameter estimation, and $\\mathcal{I}_T$ measures how the excess risk on the target depends on the estimation accuracy of the parameter. Therefore what really affects the excess risk (on target) is the accuracy of estimating the parameter, and vanilla MLE is naturally the most efficient choice.
>
> Conformal prediction is a popular topic in the field of uncertainty quantification. We thank the reviewer for the reference and will add more discussion in our related work regarding conformal prediction.
>
> **Response to Questions:**
> If we also have some samples $(X,Y)$ from the target, then we can simply apply MLE on the combination of source samples and these target samples. MLE (on augmented samples) will still be optimal under this scenario.

---

> > ### Comment · Reviewer_Vzp3 · 2023-11-22
> > **Reviewer response**
> >
> > The authors' response is appreciated and quite satisfactory.

---

### Official Review · Reviewer_xCis · 2023-10-31

**Soundness:** 3 good
**Presentation:** 3 good
**Contribution:** 3 good
**Rating:** 8
**Confidence:** 3

**Summary:**

The authors consider the problem of generalization for parametric models in the presence of covariate shift. Specificallly, we are given $(X,Y)$ pairs where the distribution of $X$ might be different during training and testing, but the conditional $Y|X$ is the same. They first consider the well specified regime, where the conditional distribution $Y|X$ lies in the parametric family considered. In that regime, they prove that for general parametric families under minimal assumptions (identifiability of the model, convexity of the likelihood), the Maximum Likelihood Estimator (MLE) is consistent and achieves the minimax-optimal sample complexity, without any assumptions on the density ratio between source and target distribution. They also instantiate their bounds for some concrete settings: linear regression, logistic regression and phase retrieval. They then focus on the misspecified setting, where the conditional $Y|X$ doesn't necessarily lie in the parametric class. They prove that MLE is no longer consistent and that a modification called Weighted Maximum Likelihood Estimation (WMLE) is indeed consistent. Finally, they provide a sample complexity lower bound for a specific parametric class which matches the upper bound for WMLE.

**Strengths:**

This paper tackles a well-motivated problem which has not been studied in prior work. I believe it is a nice step towards understanding the sample complexity of learning parametric models under covariate shift. It offers a comprehensive set of results, first tackling the well-specified case and then the mis-specified one, thus providing a clear picture of the state of the art in that area. A nice feature of the result in the well-specified case is that it gives a tight answer about the sample complexity, identifying exactly how the mismatch in training and test distribution will impact the performance. The paper is also written clearly, with claims that are explained adequately and there is a nice discussion about prior work.

**Weaknesses:**

Perhaps this is not much of a weakness, but some of the main results do not involve significant novelty on a technical level. For example, the proof of the upper bound follows using standard arguments in parametric estimation. First they prove that the parameter $\beta$ can be learned with accuracy $\sqrt{\frac{Tr(\mathcal{I}_S^{-1})}{n}}$(Lemma A.2), which is expected, as $\mathcal{I}_S$ is the Fischer information evaluated on the source distribution. This is shown using the standard Taylor expansion up to third order and utilizing the bounds on the first three derivatives. Then they compute how this guarantee on $\beta^*$ translates to a guarantee about the target distribution, which is how the final factor of $Tr(\mathcal{I}_T \mathcal{I}_S^{-1})$ is obtained. Overall, this is similar to the standard consistency arguments in parametric statistics.

**Questions:**

-The finite sample guarantees that are presented hold when the number of samples $n$ is greater than some threshold, which depends on many parameters of the problem, including for example $\|\mathcal{I}_T^{1/2}\mathcal{I}_S^{-1}\mathcal{I}_T^{1/2}\|_2$, which also measures how close the source and target distributions are in some sense. Since the sample complexity bounds only hold above that threshold, it seems that an important question is what is the dependence of the optimal sample complexity on quantities like $\|\mathcal{I}_T^{1/2}\mathcal{I}_S^{-1}\mathcal{I}_T^{1/2}\|_2$. Have the authors considered this question?

-The authors instantiate their general bound for the well-specified setting in the case of linear regression. It would be interesting to compare the results that they get with the guarantees of prior work on this problem, such as Lei et al., if such a comparison can be made.

-Since there are two different distributions of source and target, I think it would help with exposition if the authors had some notation about taking expectation wrt each of the two distributions instead of hiding this dependence. For example, in the proof of Lemma A.2 the expectation is wrt the source distribution.

-In page 14, in the derivation of inequality (13) in the second line, shouldn't there be an expectation above the equality sign? ($\mathbb{E}[\nabla l(\beta^*)] = 0$)

---

> ### Author Response · Authors · 2023-11-15
> **Author Response**
>
> We thank the reviewer for the positive feedback.
>
> **Q1**:
>
> Thanks for this interesting question. Our main results state that the optimal rate is achieved when the sample size exceeds some threshold. In some cases, the threshold we obtained is optimal. For example, in linear regression (Section 4.1), when the mean shift is small, the threshold is $O(d)$, which aligns with the result of usual linear regression. However, for some other cases, e.g., logistic regression, we admit that the threshold may not be tight, as we lean on a general framework designed for a variety of models rather than a specific one. Conducting a more detailed analysis to determine the optimal threshold presents an intriguing direction for future research.
>
> **Q2**:
>
> Regarding Lei et al, they give an exact minimax linear estimator under fixed design. The estimator they proposed is not MLE and is much more complicated in certain regimes. Their results rely on the strong assumption that **the covariance matrix on source and on target commute** with each other. In contrast, our results suggest that MLE is optimal among all estimators up to some constants under mild assumptions. We will add more discussions after the linear regression example (Section 4.1).
>
> **Q3 & Q4**:
>
> Thanks for your valuable suggestions and pointing out the typo regarding the expectation. We will edit our paper accordingly.

---

### Official Review · Reviewer_qbMA · 2023-11-09

**Soundness:** 2 fair
**Presentation:** 2 fair
**Contribution:** 2 fair
**Rating:** 3
**Confidence:** 4

**Summary:**

This study investigates the effectiveness of maximum likelihood estimation (MLE) in the out-of-distribution optimization problem. The authors prove the effectiveness by showing the minimax optimality.

**Strengths:**

This study enlightens the use of the MLE under correctly specified models. The reported results are practically important because we can avoid using advanced methods for addressing the covariate shift problem. The main strength of this study lies in the finite-sample minimax optimality of MLE under a covariate shift.

**Weaknesses:**

Although the points raised by the authors are practically important, I cannot find novel findings.

**OOD under the correctly specified models**
Firstly, as the authors mention, it is known that the covariate shift problem is not serious when models are correctly specified (Shimodaira, 2000). Although it seems that recent studies tend to omit discussing model misspecification when discussing the covariate shift problem, classical studies motivate the use of covariate shift adaptation by considering the model misspecification setting. Those studies and classical statistics agree that correctly specified models can address the OOD generation. Therefore, when models are correctly specified, "MLE is all we need" has been known to researchers, though recent studies often omit this point.

**Finte sample optimality**
Then, my question for this study is its contribution. The authors insist that the contributions lie in the finite-sample minimax optimality of MLE under correctly specified parametric models. However, if we consider parametric models and minimax optimality, I think that such a result has already been shown by existing studies with more general forms or is trivial, though I do not raise some specific related work. This is because when considering parametric models $f_\theta(x)$ parametrized by $\theta$, it is enough to consider the minimax optimality of estimation of $\theta$. Once we establish the optimality, we can extend the result to show the minimax optimality of estimation of $f_\theta(x)$ using the Taylor expansion for each $x$ or uniformly over $x$. There are various results to discuss the optimality of $\theta$ estimation, and we can just employ them.

**Uniform convergence**
Several existing studies discuss the uniform or point-wise minimax optimality of estimation of $f_\theta(x)$, which directly implies the minimax optimality under a covariate shift. I believe that we can easily obtain such a result if we only consider parametric models, and the results shown by the authors can be more strengthened.

**Questions:**

1. Why do authors consider the minimax optimality of parametric models under a covariate shift? Is it difficult to derive corresponding results for nonparametric models? Or can the authors show uniform optimality?
2. To the best of my knowledge, it has been known that correctly specified parametric models can adapt the OOD, at least in statistics. Therefore, the contributions of this study mainly lie in the finite-sample minimax optimality. Is my understanding correct?


**Title**
By the way, I think that the title may not be appropriate. As I discussed and the authors mention in the draft, "MLE is all we need" has been known. The true contribution of this study is the finite-sample minimax optimality of MLE for correctly specified parametric models. If so, the title should express the true contributions. For example, if I were the authors, I name the study "Minimax Optimality of Maximum Likelihood Estimation for Covariate Shifted Norm." Anyway, the authors should focus on their contributions more to clarify the claims of this study.

**Details Of Ethics Concerns:**

None.

---

> ### Author Response · Authors · 2023-11-15
> **Author Response**
>
> **1. Response to Weaknesses**:
>
> We thank the reviewer for the comments. We respectfully disagree that “MLE is all we need” is known. We will first defend our title and re-highlight our novelty and main contribution. We will then answer the remaining questions individually.
>
> **Title, novelty of this paper.** We believe that in order to claim that MLE is *all you need* for well-specified covariate shift, we need to provably show two points:
>
> (a) MLE can find the correct solution under well-specified covariate shifts.
>
> (b) More importantly, no algorithm is better/more efficient than MLE in this setting (therefore MLE is truly all that you need). That is, the statistical rate of MLE is optimal among *all* algorithms.
>
> As written in our related work section, we agree that Shimodaira (2000) provides (asymptotic only) analysis for (a), which makes (a) partially known as the reviewer pointed out. However, Shimodaira (2000) did not prove (b) even in the asymptotic sense. Specifically, Shimodaira (2000) only proves that the asymptotic rate of MLE is the best among a family of “weighted likelihood estimators”. Their work/techniques do not extend to prove the optimality of MLE among a rich variety of algorithms beyond that family. In contrast, this paper provides the first general non-asymptotic result for point (a), and gives the first proof for point (b), where lies the novelty and contribution of this paper.
>
> **It’s known OOD is not a serious problem under the correctly specified models.** We agree it’s known asymptotically. However, as we highlighted in the first bullet point, this only partially contributes to point (a), which is a fraction of our claim. The equally important (if not more important) point (b) and the non-asymptotic analysis of point (a) are not known, which are first addressed by this paper.
>
> **Finite sample optimality.** The reviewer raised the comment: “However, if we consider parametric models and minimax optimality, I think that such a result has already been shown by existing studies with more general forms or is trivial, though I do not raise some specific related work. ” To our best knowledge, we are not aware of any existing minimax optimality results of MLE under general well-specified covariate shift, either with more general forms, or with trivial derivations. We are also not quite able to verify the correctness of the short proof sketch provided by the reviewer. In particular, we are not sure why “it is enough to consider the minimax optimality of estimation of theta” especially in the lower bound perspective. In general, parameter estimation being hard does not imply prediction (the focus of this paper) being hard. It would be highly appreciated if the reviewer can provide concrete references or more detailed checkable proof sketch to facilitate discussion.
>
> **Uniform convergence.** We agree with the reviewer that if one can provide a point-wise optimal estimator, then optimality under covariate shift is guaranteed. However, to the best of our knowledge, we haven’t seen such studies in our context. The closest one we know is the paper “Local convergence rates of the least squares estimator with applications to transfer learning”. The paper studies the local convergence rate when the regression function is Lipschitz. It would be helpful if the reviewer can provide more related work on this thread.
>
> **2. Response to Questions**:
>
> **Q1**: Thanks for raising this interesting and important question. In fact, nonparametric models behave differently under covariate shifts. For instance, nonparametric models over RKHS under covariate shift have been studied by [1]. It was shown that MLE (ERM in their language) is provably suboptimal  for addressing covariate shift under nonparametric RKHS assumptions. In contrast, for well-specified parametric models, we show that this wouldn’t be the case as MLE is minimax optimal under covariate shift for well-specified parametric models.
>
> **Q2**: Hopefully our response in the first bullet point answers this question.
>
> [1] Cong Ma, Reese Pathak, and Martin J Wainwright. Optimally tackling covariate shift in rkhs-based nonparametric regression. The Annals of Statistics, 51(2):738–761, 2023.

---

### Meta-Review · Area_Chair_HMBp · 2023-12-14

**Metareview:**

The paper shows that MLE is still relevant and achieves minimax optimality even under covariate shift for certain well-specified settings. In particular, they show the result for some families of parametric models. The paper also shows that in the non well-specified setting other methods are required. Overall, the reviews were positive.

**Justification For Why Not Higher Score:**

NA

**Justification For Why Not Lower Score:**

NA

---

### Decision · Program_Chairs · 2024-01-16

Accept (poster)